# On Feasible Rewards in
# Multi-agent Inverse Reinforcement Learning

**Till Freihaut**
Department of Computer Science
University of Zurich
`freihaut@ifi.uzh.ch`

**Giorgia Ramponi**
Department of Computer Science
University of Zurich
`ramponi@ifi.uzh.ch`

## Abstract

Multi-agent Inverse Reinforcement Learning (MAIRL) aims to recover agent reward functions from expert demonstrations. We characterize the feasible reward set in Markov games, identifying all reward functions that rationalize a given equilibrium. However, equilibrium-based observations are often ambiguous: a single Nash equilibrium can correspond to many reward structures, potentially changing the game's nature in multi-agent systems. We address this by introducing entropy-regularized Markov games, which yield a unique equilibrium while preserving strategic incentives. For this setting, we provide a sample complexity analysis detailing how errors affect learned policy performance. Our work establishes theoretical foundations and practical insights for MAIRL.

## 1 Introduction

Multi-agent Reinforcement Learning (MARL) has garnered substantial attention in recent years due to its capacity to model scenarios involving interacting agents. Notable successes have been achieved across diverse domains, including autonomous driving [Shalev-Shwartz et al., 2016, Zhou et al., 2020], internet marketing [Jin et al., 2018], multi-robot control [Dawood et al., 2023], traffic control [Wang et al., 2019], and multi-player games [Baker et al., 2019, Samvelyan et al., 2019]. A critical prerequisite for these applications is the careful design of reward functions, a task that proves challenging even in single-agent settings [Amodei et al., 2016, Hadfield-Menell et al., 2017] and becomes significantly more complex in multi-agent environments where each agent's reward function must be tailored to their specific, potentially conflicting, objectives.

In numerous real-world scenarios, expert demonstrations of optimal behavior may be observable, while the underlying reward function driving these actions remains unknown. This is precisely the domain of Inverse Reinforcement Learning (IRL) [Ng and Russell, 2000]. The objective of IRL is to recover plausible reward functions that can rationalize the observed behavior as optimal. However, early research in IRL highlighted a fundamental challenge: the problem is inherently ill-posed, as multiple reward functions can potentially explain the same observed behavior. Subsequent research has therefore focused on reformulating the IRL problem to enhance its practicality and applicability in real-world contexts [Abbeel and Ng, 2004, Ziebart et al., 2008, Ramachandran and Amir, 2007, Ratliff et al., 2006a].

The extension of IRL to the multi-agent setting introduces novel complexities, particularly concerning the definition of optimality and the multiplicity of Nash equilibria, given that each agent's optimal strategy is dependent on the strategies of all other agents. This necessitates the adoption of game-theoretic solution concepts, with the Nash equilibrium being the most prevalent [Goktas et al., 2024, Song et al., 2018, Ramponi et al., 2023, Fu et al., 2021]. In contrast to the substantial progress in understanding the theoretical underpinnings of single-agent IRL, the theoretical foundations of Multi-agent Inverse Reinforcement Learning remain comparatively underexplored. In single-

39th Conference on Neural Information Processing Systems (NeurIPS 2025).

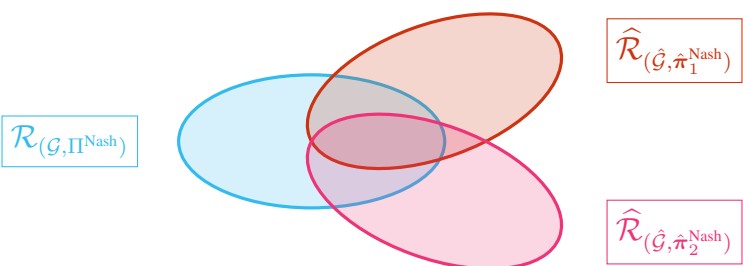

Figure 1: Feasible Reward Sets of true sets of Nash equilibria and the recovered feasible reward sets for two different observed Nash equilibria.

agent IRL Metelli et al. [2021] established explicit conditions for feasible reward functions and developed efficient algorithms for unknown transition models and expert policies, assuming access to a generative model. This work has been extended to settings without a generative model [Lindner et al., 2022], stricter optimality metrics [Zhao et al., 2024, Metelli et al., 2023], and offline settings Lazzati et al. [2024b], Zhao et al. [2024]. However, these studies are confined to single-agent scenarios and evaluate performance based on criteria that are not directly transferable to general-sum Markov games. In Appendix B we provide an extensive discussion on related works.

This paper aims to bridge the existing gap between the theoretical understanding of IRL in single-agent systems and its application to multi-agent systems. Specifically, we first address the research question:

(Q1) *What constitutes a rigorous definition of Multi-agent Inverse Reinforcement Learning?*

To address this question independently of a specific MAIRL algorithm, we derive properties of the feasible reward set, drawing inspiration from the initial work in single-agent settings by Metelli et al. [2021]. First, we define a straight-forward extension from the single-agent feasible reward set to the multi-agent setting as all the rewards under which a *single* Nash equilibrium expert is optimal, meaning it is indeed a Nash equilibrium. Then, we demonstrate that a single observed equilibrium is insufficient for identifying expressive reward sets, as distinct observed equilibria can induce different feasible reward sets (see Fig. 1 for an illustration). This can result in a Nash Gap of order $(1 - \gamma)^{-1}$ due to the multiplicity of the Nash equilibria. To mitigate the equilibrium selection problem, we introduce entropy-regularized Multi-agent IRL. Then, we formally characterize the inherent increase in complexity associated with the multi-agent setting. Within this framework, we characterize feasible rewards and establish sample complexity bounds that account for errors in transition dynamics and policy estimation, assuming access to a generative model.

In single-agent settings, the introduction of entropy-regularized experts has facilitated the derivation of conditions under which the reward function is identifiable [Cao et al., 2021, Rolland et al., 2022]. This motivates our second research question:

(Q2) *Is reward identifiability achievable in Multi-agent settings?*

We provide a partial positive answer to this question. We show that in general-sum Markov games without additional structural assumptions, reward identifiability is only possible in the average reward sense. However, we prove that if the underlying reward structure is linearly separable, meaning that the reward can be decomposed into a reward for player 1 and player 2, $R(s, a, b) = R_A(s, a) + R_B(s, b)$, then reward identification (up to additive constants) is possible.

## 2 Preliminaries

We present the essential background and notation used throughout this paper, also summarized in Appendix A.

**Mathematical background.** Let $\mathcal{X}$ be a finite set, then we denote by $\mathbb{R}^{\mathcal{X}}$ all functions mapping from $\mathcal{X}$ to $\mathbb{R}$. Additionally, we denote by $\Delta^{\mathcal{X}}$ the set of probability measures over $\mathcal{X}$. For $n \in \mathbb{N}$ we use $[n] := \{1, \dots, n\}$. We introduce for a (pre)metric space $(\mathcal{X}, d)$ with $\mathcal{Y}, \mathcal{Y}' \subseteq \mathcal{X}$

two non-empty sets the *Hausdorff (pre)metric* $\mathcal{H}_d : 2^{\mathcal{X}} \times 2^{\mathcal{X}} \to [0, +\infty)$ as $\mathcal{H}_d(\mathcal{Y}, \mathcal{Y}') :=$ $\max\left\{\sup_{y \in \mathcal{Y}} \inf_{y' \in \mathcal{Y}'} d(y, y'), \sup_{y' \in \mathcal{Y}'} \inf_{y \in \mathcal{Y}} d(y, y')\right\}$. Additionally, for two probability distributions $\mu, \mu'$ over a finite set $\mathcal{X}$, we denote the total variation as $\mathrm{TV}(\mu, \mu') := \sum_{x \in \mathcal{X}} |\mu(x) - \mu'(x)|$.

**Markov games.** An infinite time, discounted n-person general-sum Markov game [Shapley, 1953, Takahashi, 1964, Fink, 1964] without reward function (MG $\setminus$ R) is characterized by a tuple $\mathcal{G} = (n, \mathcal{S}, \mathcal{A}, P, \gamma, \rho)$, where $n \in \mathbb{N}$ denotes the finite number of players; $\mathcal{S}$ the finite state space; $\mathcal{A} := \mathcal{A}^1 \times \ldots \times \mathcal{A}^n$ the joint action space of the individual action spaces $\mathcal{A}^i$; $P : \mathcal{S} \times \mathcal{A} \to \Delta^{\mathcal{S}}$ the transition model; $\gamma$ is the discount factor and $\rho$ is the initial state distribution. We will make use of the words persons, agents and players interchangeably. The strategy of a single agent, also called the policy, we denote by $\pi^i : S \to \Delta^{\mathcal{A}^i}$. A joint strategy is given by $\boldsymbol{\pi} = (\pi^1, \ldots, \pi^n) = (\pi^i, \pi^{-i})$, where $\pi^{-i}$ refers to the (joint-) policy of all players except player $i$. A joint action is denoted by $\mathbf{a} = (a^1, \ldots, a^n) \in \mathcal{A}$. Therefore, the probability of a joint strategy is given by $\boldsymbol{\pi}(\mathbf{a} \mid s) := \prod_{j=1}^n \pi^j(a^j \mid s)$. $\Pi^i$ denotes the set of all policies for agent $i$. The discounted probability of visiting a state-(joint-)action pair, given that the starting state is drawn from $\rho$, is defined as $\overline{w}_{s,\mathbf{a}}^{\boldsymbol{\pi}, \rho} = \sum_{t=0}^{\infty} \gamma^t \mathbb{P}^t(s, \mathbf{a}, \rho)$, where $\mathbb{P}^t$ denotes the probability of visiting the joint state action pair when drawing the initial state from $\rho$ and following the joint policy $\boldsymbol{\pi}$ for $t$ steps. If the starting distribution is deterministic for a state $s$, we omit the dependence on $\rho$ and simply write $\overline{w}_{s,\mathbf{a}}^{\boldsymbol{\pi}}$.

**Reward function.** The reward function for an agent, $R^i : \mathcal{S} \times \mathcal{A} \to [-R_{\max}^i, R_{\max}^i]$, takes a state and a joint action as inputs, mapping them to a bounded real number. The joint reward is represented as $R = (R^1, \ldots, R^n)$. The uniform reward bound across all agents is defined by $R_{\max} := \max_{i \in [n]} R_{\max}^i$. A Markov game without reward $\mathcal{G}$ combined with a joint reward results in a standard Markov game denoted as $\mathcal{G} \cup R$.

**Value functions and equilibrium concepts.** For a Markov game $\mathcal{G} \cup R$ with a policy $\boldsymbol{\pi}$ we define the *Q-function* and the *value-function* of an agent $i$ for a given state and action as $Q_{\mathcal{G} \cup R}^{i, \boldsymbol{\pi}}(S, \mathbf{A}) = \mathbb{E}^{\boldsymbol{\pi}}[\sum_{t=0}^{\infty} \gamma^t R^i(S_t, \mathbf{A}_t) \mid S_t = S, \mathbf{A}_t = \mathbf{A}]$ and $V_{\mathcal{G} \cup R}^{i, \boldsymbol{\pi}}(s) = \sum_{\mathbf{a}} \boldsymbol{\pi}(\mathbf{a} \mid s) Q_{\mathcal{G} \cup R}^{i, \boldsymbol{\pi}}(s, \mathbf{a})$. If it is clear from the context what the underlying Markov game is, we omit the subscript.

**Nash Equilibrium.** Various types of equilibrium solutions have been proposed to model optimal strategies in Markov games. In this work, we focus on the NE similarly to previous works on MAIRL [Goktas et al., 2024, Song et al., 2018]. A strategy profile is a Nash equilibrium if no agent can improve their expected return by unilaterally deviating from their strategy, assuming the strategies of the other agents remain unchanged. Formally, a policy $\boldsymbol{\pi}^{\mathrm{Nash}}$ is a (perfect) Nash equilibrium strategy, if for every state $s$ and every agent $i$ $V_{\mathcal{G} \cup R}^{i, \boldsymbol{\pi}^{\mathrm{Nash}}}(s) \geq V_{\mathcal{G} \cup R}^{i, (\pi^i, \pi^{-i, \mathrm{Nash}})}(s) \quad \forall \pi^i \in \Pi^i$. To simplify the notation used in the remainder of this work, we will denote this as $V^i(\boldsymbol{\pi}^{\mathrm{Nash}}) \geq V^i(\pi^i, \pi^{-i, \mathrm{Nash}})$. The set of Nash equilibria, depending on the underlying reward, we will denote as $\Pi^{\mathrm{Nash}}(R)$.

**Entropy Regularized Markov games.** For better readability, let us consider the case $\boldsymbol{\pi} = (\mu, \nu)$ and $\mathcal{A}^2 := \mathcal{B}$. Then, the value function of player 1 in a $\lambda$ entropy regularized Markov game is defined as $V_{\lambda}^{1, (\mu, \nu)}(s) = \mathbb{E}^{(\mu, \nu)}[\sum_{t=0}^{\infty} \gamma^t \left(R^1(S_t, A_t, B_t) - \lambda \log(\mu(A_t \mid S_t))\right) \mid S_0 = s]$. Additionally, we have $V_{\lambda}^{1, (\mu, \nu)}(\rho) = \mathbb{E}_{S_0 \sim \rho}[V_{\lambda}^{1, (\mu, \nu)}(S_0)]$. The value function for player 2 is defined analogously. Additionally, the Q-function is defined as $Q_{\lambda}^{1, (\mu, \nu)}(s, a, b) = R^1(s, a, b) + \sum_{s'} P(s' \mid s, a, b) V_{\lambda}^{1, (\mu, \nu)}(s')$.

**Multi-agent Inverse Reinforcement Learning.** Given a Markov game without a reward function $\mathcal{G}$ and a Nash equilibrium expert, the MAIRL problem is defined as the tuple $(\mathcal{G}, \boldsymbol{\pi}^{\mathrm{Nash}})$. If we only have access to an estimated version of $(\mathcal{G}, \boldsymbol{\pi}^{\mathrm{Nash}})$, we call it recovered MAIRL problem and denote it as $(\hat{\mathcal{G}}, \hat{\boldsymbol{\pi}}^{\mathrm{Nash}})$ and analogously for entropy equilibria by replacing $\hat{\boldsymbol{\pi}}^{\mathrm{Nash}}$.

# 3 Feasible reward set in multi-agent systems

This section addresses research question (Q1). First, we will define MAIRL for a single NE observation. Then, we will show, that this can be ill-suited for multi-agent systems due to the multiplicity of equilibria. This motivates to consider entropy regularized Markov games to guarantee a unique equilibrium.

## 3.1 Nash equilibrium observations

In this section, we begin by revisiting single-agent Inverse Reinforcement Learning. The feasible reward set in single-agent IRL, first defined by Metelli et al. [2021] and later refined for different settings as e.g. the offline case in [Zhao et al., 2024, Metelli et al., 2023, Lazzati et al., 2024b], lacks a multi-agent counterpart for observed expert equilibria. We thus translate the single-agent definition, formalizing feasible rewards as in prior works [Lin et al., 2014, 2018].

**Definition 3.1.** Let a MAIRL problem $(\mathcal{G}, \boldsymbol{\pi}^{\mathrm{Nash}})$ with a single (observed) Nash equilibrium policy be given. Then, the feasible reward set for general-sum Markov games is given by

$$\mathcal{R}_{(\mathcal{G}, \boldsymbol{\pi}^{\mathrm{Nash}})} = \left\{ R \in \mathcal{R} \mid \forall i \in [n], \forall s \in \mathcal{S}, \forall \pi_i \in \Pi^i : V_{\mathcal{G} \cup R}^{i, (\pi_i^{\mathrm{Nash}}, \pi_{-i}^{\mathrm{Nash}})}(s) \geq V_{\mathcal{G} \cup R}^{i, (\pi_i, \pi_{-i}^{\mathrm{Nash}})}(s) \right\}.$$

Here, $\boldsymbol{\pi}^{\mathrm{Nash}} \in \Pi^{\mathrm{Nash}}(R)$ is any Nash equilibrium, analogous to an optimal policy in single-agent IRL. A key difference in MAIRL is that different NEs can yield varying values, and we impose no restriction on the observed NE (pure or mixed) from $\Pi^{\mathrm{Nash}}(R)$. This is the first fundamental difference between IRL and MAIRL.

Since varying NE values make single-agent value-based gap objectives [Metelli et al., 2021, 2023, Zhao et al., 2024, Lazzati et al., 2024b] unsuitable for MAIRL, we adapt the Nash Gap, recently used in multi-agent imitation learning [Ramponi et al., 2023, Tang et al., 2024], as our objective.

**Definition 3.2** (Nash Imitation Gap for MAIRL). Let $\mathcal{G} \cup R$ be the underlying $n$-person general-sum Markov game. Furthermore, let $\hat{\boldsymbol{\pi}}$ be the policy recovered from the corresponding MAIRL problem. Then we define the Nash Imitation Gap of $\hat{\boldsymbol{\pi}}$ as

$$\mathcal{E}(\hat{\boldsymbol{\pi}}, R) := \max_{i \in [n]} \max_{\pi^i \in \Pi^i} V_{\mathcal{G} \cup R}^i(\pi^i, \hat{\pi}^{-i}) - V_{\mathcal{G} \cup R}^i(\hat{\boldsymbol{\pi}}).$$

The definition possesses the desirable property that it equals $0$ if $\hat{\boldsymbol{\pi}}$ is an NE, and, it is $> 0$ if $\hat{\boldsymbol{\pi}}$ is not an NE in the underlying Markov game.

Normally, we cannot assume to know the expert equilibrium nor the transition function. Therefore, to analyze how estimation errors in the transition probability and expert policy affect the recovered feasible reward set, we relate them to our proposed optimality criterion.

**Definition 3.3** (Optimality Criterion). Let $\mathcal{R} := \mathcal{R}_{(\mathcal{G}, \boldsymbol{\pi}^{\mathrm{Nash}})}$ be the exact feasible set and $\hat{\mathcal{R}} := \mathcal{R}_{(\hat{\mathcal{G}}, \hat{\boldsymbol{\pi}})}$ the recovered feasible set after observing $N \geq 0$ samples from the underlying MAIRL problem $(\mathcal{G}, \boldsymbol{\pi}^{\mathrm{Nash}})$. An algorithm is $(\varepsilon, \delta, N)$-correct after $N$ samples if, with probability at least $1 - \delta$, it holds that:

$$\sup_{R \in \mathcal{R}} \inf_{\hat{R} \in \hat{\mathcal{R}}} \sup_{\hat{\boldsymbol{\pi}} \in \Pi^{\mathrm{Nash}}(\hat{R})} \mathcal{E}(\hat{\boldsymbol{\pi}}, R) \leq \varepsilon, \quad \sup_{\hat{R} \in \hat{\mathcal{R}}} \inf_{R \in \mathcal{R}} \sup_{\hat{\boldsymbol{\pi}} \in \Pi^{\mathrm{Nash}}(\hat{R})} \mathcal{E}(\hat{\boldsymbol{\pi}}, R) \leq \varepsilon,$$

where $\Pi^{\mathrm{Nash}}(\hat{R}) := \{ \boldsymbol{\pi} \mid V_{\hat{\mathcal{G}} \cup \hat{R}}^{\boldsymbol{\pi}}(s) > V_{\hat{\mathcal{G}} \cup \hat{R}}^{\tilde{\pi}^i, \pi^{-i}}(s) \, \forall \tilde{\pi}^i \in \Pi^i, \forall s \in \mathcal{S}, \forall i \in [n] \}$.

The optimality criterion is in the Hausdorff metric style, see 2. The first condition ensures that the recovered feasible set captures a reward function that makes sure that the recovered policy is at most an $\varepsilon$-NE in the true Markov game. However, this would support choosing a set that captures all possible reward functions $\mathbb{R}^{\mathcal{S} \times \mathcal{A}}$. Consequently, the second condition ensures that this is not possible by requiring every recovered reward to also have a true reward function that captures the desired behavior. Additionally, note that the optimality criterion depends on $\hat{R}$ as $\hat{\boldsymbol{\pi}}$ is the Nash equilibrium of the recovered Markov game $\hat{\mathcal{G}} \cup \hat{R}$.

The shortcomings of observing only a single equilibrium are discussed next. For completeness, this framework is further analyzed in Appendix C.

**Feasible reward set and equilibrium ambiguity.** In this section, we show that observing only a single equilibrium solution leads to a non-expressive feasible reward set. Fu et al. [2021] noted that relying on a single Nash equilibrium for inverse learning introduces inherent limitations. Here, we extend this finding to the feasible reward set, rather than focusing solely on specific reward functions.

Tang et al. [2024] showed that minimizing the regret gap is hard in Imitation Learning problems. However, in this work we do not consider the setting of mimicking the expert policy instead we examine the feasible reward set. In general, (MA)IRL can be more powerful as it allows to transfer the reward function to new environments. In Appendix G we provide an experiment in a simple Grid World game that emphasizes this. Unfortunately, the next theorem shows, that learning under a recovered reward function from the feasible set stemming from a MAIRL problem with a single equilibrium observation can lead to a NE that has a Nash Gap of the order $(1-\gamma)^{-1}$ in the original Markov game.

**Proposition 3.4.** *Let us consider any MAIRL algorithm* $\mathrm{Alg}_{\mathrm{MAIRL}}$ *that chooses* $\hat{R} \in \mathcal{R}_{(\hat{\mathcal{G}}, \hat{\boldsymbol{\pi}}^{\mathrm{Nash}})}$ *that is not a constant reward, i.e.* $\hat{R} \neq C$ *for* $C \in [-R_{\max}, R_{\max}]$. *Furthermore, consider a MARL algorithm* $\mathrm{Alg}_{\mathrm{MARL}}$ *that guarantees learning a policy* $\tilde{\boldsymbol{\pi}} \in \Pi^{\mathrm{Nash}}(\hat{R})$. *Then, there exists a Markov game, such that even if* $\hat{\boldsymbol{\pi}} \in \Pi_{\mathrm{Nash}}$ *and* $\hat{R} \in \mathcal{R}_{(\mathcal{G}, \boldsymbol{\pi}^{\mathrm{Nash}})}$ *it holds true that* $\mathcal{E}(\tilde{\boldsymbol{\pi}})$ *is of order* $(1-\gamma)^{-1}$.

The construction of the underlying general-sum Markov game can be found in Fig. 2. The idea of the proof is that Definition 3.1 only ensures that the recovered expert $\hat{\boldsymbol{\pi}}^{\mathrm{Nash}}$ is a NE under $\hat{R}$, an MAIRL algorithm cannot capture a meaningful reward for other equilibria that potentially have different values. Therefore, the constraints on the rewards inside the feasible reward set only gives a relation for a fixed strategy of the opponent. Consider a simple example with two players where player 2 plays action $b$ with probability one. Then, for player one the constraints only tell us something about $R^1(s, a^{\mathrm{Nash}}, b) \geq R^1(s, a, b) \, \forall a \in \mathcal{A}^1$, but nothing on rewards $R^1(s, a, b') \, \forall (a, b') \in \mathcal{A}^1 \times \mathcal{A}^2$. This allows that the recovered reward functions allow for new equilibria, i.e. $\Pi^{\mathrm{Nash}}(\hat{R}) \supset \Pi^{\mathrm{Nash}}(R)$ that can be exploited in the original Markov game. We empirically investigate this for the considered Game in Proposition 3.4. Summarized, the reward set captures too many reward functions and this flexibility in reward specification allows for undesirable scenarios, such as **changing the nature of the game**. This means the game can transform the set of all Nash equilibria e.g. from a coordination game into an anti-coordination variant. For further intuition we provided additional examples in Example D.1.

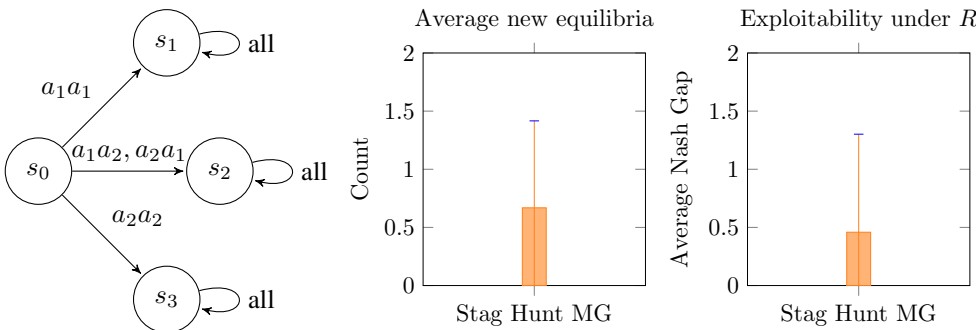

Figure 2: Failure of single equilibrium observation.

To address this, we propose the following definition of the feasible reward set.

**Definition 3.5.** Let $\Pi^{\mathrm{Nash}}(R)$ denote the set of Nash equilibrium in the underlying Markov game $\mathcal{G} \cup R$. Then, the feasible reward set for general-sum Markov games is defined as:

$$\mathcal{R}_{(\mathcal{G}, \Pi^{\mathrm{Nash}})} = \left\{ R \in \mathcal{R} \mid \forall \boldsymbol{\pi}^{\mathrm{Nash}} \in \Pi^{\mathrm{Nash}}, \, i \in [n], \, s \in \mathcal{S}, \, \pi_i \in \Pi^i : V_{\mathcal{G} \cup R}^{i, (\pi_i^{\mathrm{Nash}}, \pi_{-i}^{\mathrm{Nash}})}(s) > V_{\mathcal{G} \cup R}^{i, (\pi_i, \pi_{-i}^{\mathrm{Nash}})}(s) \right\}.$$

Note that, the subscript is now on the set of equilibria instead of a single one. This definition ensures that the feasible reward set aligns with all equilibrium solutions of the original game, rather than a single observed equilibrium. While this improves the interpretability of the feasible reward set, calculating even one Nash equilibrium is computationally intractable in general-sum Markov games. Hence, this definition does not fully resolve the tractability issue in MAIRL.

The multiplicity of equilibria and the absence of a unique value pose a significant challenge in Inverse Reinforcement Learning, in particular which equilibria should be chosen, commonly referred to as the *Equilibrium Selection problem*. This ambiguity complicates reward inference, as different equilibria can lead to inconsistent or unreliable outcomes.

Leonardos et al. [2021] address this by proposing game structure modifications, such as *regularized Markov games*. Techniques like entropy regularization refine the equilibrium set to a unique one, eliminating the selection problem. We will transfer this concept to MAIRL and investigate its implications in subsequent chapters.

## 3.2 Feasible rewards for entropy regularized games

This section examines *Entropy-Regularized Markov games* and their unique *Quantal Response Equilibrium (QRE)*, which reflects bounded rationality. We show that QREs help avoid problematic reward configurations (cf. Fig. 2) and yield recovered rewards more similar to the true game rewards. We then formally define the feasible reward set for QRE experts and provide a sample complexity analysis for MAIRL to achieve the entropy version of Definition 3.3. Technically, this is not the Nash Gap anymore, instead it only measures the exploitability of a policy in the entropy regularized game.

**Avoiding the ambiguity with QRE expert.** We begin by demonstrating that entropy regularization, yielding a unique fully mixed QRE, provides sufficient structure for reward recovery to avoid the degenerate cases highlighted in Fig. 2. Recalling our empirical validations (Fig. 2), single, potentially pure, equilibrium observations permitted exploitable new equilibria in the original game. We now present analogous experiments using QRE observations (see Fig. 3).

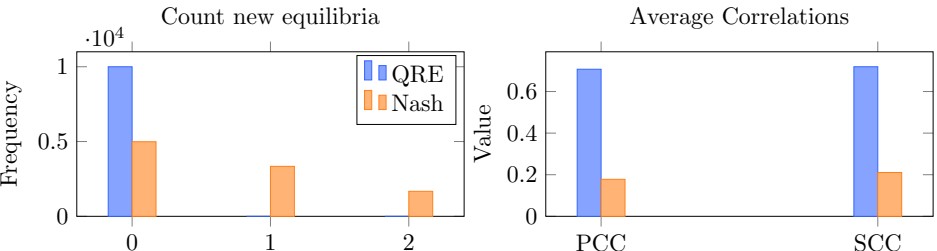

Figure 3: Recovered rewards under QRE equilibrium observations.

We can observe that in simple games the QRE ensures that no new pure equilibria arise and additionally the correlation of the recovered reward function and the true reward function measured by the Pearson Correlation Coefficient (PCC) and Spearman Correlation Coefficient (SCC) are significantly higher. The reason for this is that the QRE enforces the equilibrium observation to explore the environment better and gets rid of the equilibrium selection problem. This can be further motivated by assumptions needed also in the Multi-agent Imitation Learning setting, where Tang et al. [2024] showed that under coverage assumption equilibria can successfully be recovered. This is automatically fulfilled if the observed expert is a QRE expert.

**Characterization of feasible rewards.** In the single-agent IRL setting, an explicit characterization of the reward function in entropy-regularized Markov Decision Processes (MDPs) was first derived by Cao et al. [2021]. The authors introduced entropy regularization to tackle the ill-posedness of the IRL problem. In particular, the authors established conditions under which the reward function is identifiable up to a constant. These conditions were subsequently simplified by Rolland et al. [2022], providing further insights into the structure of the reward function. This naturally motivates our second research question (Q2), which we will answer in Section 4. In this section we will first focus on the feasible reward set.

We begin our analysis in a manner similar to the single-agent case. For better readability we from now on assume to only have two players, with policies $\mu, \nu$. We can give an explicit characterization of the optimal policy for the agents. Next, we give this definition for player 1, it analogously holds also for player 2

$$\mu^*(a \mid s) = \frac{\exp\left(\frac{1}{\lambda} \sum_{b' \in \mathcal{B}} \nu^*(b' \mid s) Q_\lambda^{*,1}(s, a, b')\right)}{\sum_{a' \in \mathcal{A}} \exp\left(\frac{1}{\lambda} \sum_{b' \in \mathcal{B}} \nu^*(b' \mid s) Q_\lambda^{*,1}(s, a', b')\right)}, \tag{1}$$

where $\mu^*$ denotes the optimal policy, i.e. the quantal response equilibrium policy of player 1 and $Q^{*,1}$ the corresponding Q-function for player 1. Several remarks are in order to clarify this equation.

First, the optimal strategy depends not only on the $Q$-function but also on the strategy of the opposing agent. This can be interpreted as fixing one agent and considering the induced MDP (see, for example, Definition 4.1 in Fu et al. [2021]).

From now on we will state everything from the perspective of player one. The results for player 2 follow analogously. Therefore, we will also omit the index of the reward and value functions. Next, using the definition of the Q-function and the value function, we can rewrite (1) to get an explicit formulation of the *average* reward that agent 1 receives for playing a specific action $a \in \mathcal{A}$:

$$\sum_{b' \in \mathcal{B}} \nu^*(b' \mid s) R(s, a, b') = \lambda \log (\mu^*(a \mid s)) + V_\lambda^*(s) - \gamma \sum_{s'} \sum_{b' \in \mathcal{B}} \nu^*(b' \mid s) P(s' \mid s, a, b') V_\lambda^*(s'). \tag{2}$$

Regarding the feasible reward set, our focus is to find an explicit reward formulation to characterize the reward functions in the feasible reward set. Rewriting equation (2) in terms of a specific reward, we get the following characterization.

**Lemma 3.6.** *Let $(\mu^*, \nu^*)$ be two equilibrium policies for the 2 Person $\lambda$ Entropy-Regularized Markov game $\mathcal{G}$. Then for the MAIRL problem $(\mathcal{G}, (\mu^*, \nu^*))$ a reward $R$ is feasible if and only if there exists a function $V \in \mathbb{R}^{\mathcal{S}}$ and $|\mathcal{B}| - 1$ functions $R : \mathbb{R}^{\mathcal{S} \times \mathcal{A} \times \mathcal{B}}$, such that for all $(s, a, b) \in \mathcal{S} \times \mathcal{A} \times \mathcal{B}$*

$$R(s, a, b) = \tfrac{1}{\nu^*(b|s)} \left( \lambda \log(\mu^*(a \mid s)) + V(s) - \gamma \sum_{s'} \sum_{b' \in \mathcal{B}} \nu^*(b' \mid s) P(s' \mid s, a, b') V(s') - \sum_{b' \neq b} \nu^*(b' \mid s) R(s, a, b') \right).$$

As in practice, we do not have access to $P, \nu^*$ and $\mu^*$, our goal is to analyze the impact of estimating the transition probability and the expert's policy, and how this affects the existence of a recovered feasible reward. Therefore, we now want to analyze how a recovered MAIRL problem $(\hat{\mathcal{G}}, (\hat{\mu}^*, \hat{\nu}^*))$ translates to the original MAIRL problem. Care is required for this analysis, as we must consider the estimation of the expert itself, the induced transition model, which incorporates the estimation of the other expert's policy and their deviations for alternative actions.

**Theorem 3.7** (Error propagation). *Let the MAIRL problem be given by $(\mathcal{G}, (\mu^*, \nu^*))$ and another MAIRL problem by $(\hat{\mathcal{G}}, (\hat{\mu}^*, \hat{\nu}^*))$. Then, we have that*

$$|R(s, a, b) - \hat{R}(s, a, b)| \leq \frac{1}{\nu^*(b \mid s)\hat{\nu}^*(b \mid s)} \bigg( \lambda |\log \mu^*(a \mid s) - \log \hat{\mu}^*(a \mid s)|$$

$$+ \gamma \max_b | \sum_{s'} V(s') P(s' \mid s, a, b) - \hat{P}(s' \mid s, a, b)| + R_{\max} \mathrm{TV}(\nu, \hat{\nu}) \bigg).$$

The introduced theorem highlights the additional complexity of the problem. Specifically, it reveals two key challenges that arise in error propagation for multi-agent IRL. First, the maximum of the joint transition probabilities plays a critical role, amplifying the sensitivity of the system to inaccuracies in transition estimation. Second, any deviation in estimating the other expert's policy, quantified by the total variation distance, directly contributes to errors in the recovered reward.

**Recovering feasible rewards.** The previous section revealed, that also in the Multi-agent case, the explicit feasible reward can be decomposed into parts that depend on the policy of both agents and the transition model. Therefore, we will now analyze the amount of samples required to obtain a meaningful reward function. Let us first introduce the assumption, also common in single-agent IRL, that the lowest probability of an action taken from the experts is bounded away from zero by some constant (see e.g. Assumption D.1. in Metelli et al. [2023]).

**Assumption 3.8.** *Let $\mu^*, \nu^*$ be the QRE equilibrium expert policies. Then we assume that*

$$\min_{a \in \mathcal{A}, b \in \mathcal{B}} (\mu^*(a \mid s), \nu^*(b \mid s)) \geq \Delta_{\min} \ \forall s \in \mathcal{S}.$$

For both estimation tasks, the expert policies and the transition probability, we employ empirical estimators. For each iteration $k \in [K]$, let $n_k(s, a, b, s') = \sum_{t=1}^{k} \mathbf{1}_{(s_t, a_t, b_t, s_t') = (s, a, b, s')}$ denote the count of visits to the triplet $(s, a, b, s') \in \mathcal{S} \times (\mathcal{A} \times \mathcal{B}) \times \mathcal{S}$, and let $n_k(s, a, b) = \sum_{s' \in \mathcal{S}} n_k(s, a, b, s')$ denote the count of visits to the state-action pair $(s, a)$. Additionally, we introduce $n_k(s, a) = \sum_{t=1}^{k} \mathbf{1}_{(s_t, a_t) = (s, a)}$ and $n_k(s, b) = \sum_{t=1}^{k} \mathbf{1}_{(s_t, b_t) = (s, b)}$ as the count of times action $a$ and respectively

$b$ was sampled in state $s \in \mathcal{S}$ for each agent $i$, and $n_k(s) = \sum_{a \in \mathcal{A}} n_k(s, a)$ as the count of visits to state $s$ for any agent.

It is important to note the distinction here: the count of actions must be done separately for each agent, whereas the count of state visits needs to be done for both of the agents.

In the following theorem we will assume to have access to a generative model, an assumption that is common in initial theoretical works on IRL [Metelli et al., 2021, Lindner et al., 2022, Metelli et al., 2023]. We will discuss potential directions that loosen this assumption in Section 5.

**Theorem 3.9.** *Let Assumption 3.8 hold true. Then, allocating the samples uniformly over $\mathcal{S} \times \mathcal{A} \times \mathcal{B}$ and using the empirical estimators introduced in Eq. (12) and Eq. (13), we can stop the sampling procedure with a probability of at least $1 - \delta$ after iteration $\tau$ and satisfy the optimality criterion Definition E.2, where the sample complexity is of order $\tilde{\mathcal{O}} \left( \frac{\gamma^2 R_{\max}^2 |\mathcal{S}||\mathcal{A}||\mathcal{B}|}{(1-\gamma)^4 \varepsilon^2 \Delta_{\min}^4} \right).$*

Some remarks are in order for this complexity bound. We observe that the sample complexity bound depends on the product of the action space of both players. Translating this to the n-player setting would result in an exponential dependency in the number of players. Although this might seem unfavorable, it is generally known that learning an NE in the worst case has an exponential bound. Zhang et al. [2023] show that even in model-based two player zero-sum games with access to a generative model, where the reward knowledge can not be used during learning the sample complexity depends on $|\mathcal{A}||\mathcal{B}|$. Therefore, the derived bound for the MAIRL setting aligns with the bounds derived for learning NE in the MARL setting. Additionally, we can see that the sample complexity bound is related to estimating the expert policies. This requires to estimate the log probabilities as well as the inverse probabilities of specific action taken by one player. To do so, we need the assumptions that the probabilities are bounded away from zero (Assumption 3.8).

# 4 Identifiability in multi-agent games?

At the beginning of Section 3.2, we noted that entropy regularization has been introduced in the single-agent setting to derive conditions to identify the reward (up to constants). Unlike the single-agent case, multi-agent systems introduce additional challenges due to the interplay between agents' strategies and the underlying reward structure. These challenges make identifiability like in single-agent settings not possible, unless further assumptions on the underlying Markov game are posed. We split this section into two parts. First, we consider average reward characterization. Then, we introduce linear separable Markov games and investigate identification in this setting.

**Average reward identification.** First, let us revisit the derivations from the last section. In particular, note that the left-hand side of Eq. (2) shows, that agent 1's average reward depends on agent 2's policy $\nu(b' \mid s)$ which averages over agent 2's action. Additionally, note that the reward function is in the multi-agent setting, has dimensionality $|\mathcal{S}||\mathcal{A}||\mathcal{B}|$, while Eq. (1) only gives condition for $|\mathcal{S}||\mathcal{A}|$ and $|\mathcal{S}||\mathcal{B}|$ respectively. This shows immediately that the resulting system of equations is under-determined.

However, the left hand side can be interpreted in terms of the induced MDP. In particular we derive an explicit reward function for the average reward $R^\nu(s, a) = \sum_{b' \in \mathcal{B}} \nu(b' \mid s) R(s, a, b')$. Next, let us additionally define the induced transition function, keeping the strategy of agent 2 fixed. Then, we have $P^\nu(\cdot \mid s, a) := \sum_{b' \in \mathcal{B}} \nu(b' \mid s) P(s' \mid s, a, b')$. Thus, agent 1's decision problem, when facing a fixed opponent policy $\nu$, is equivalent to a single-agent MDP with the average reward function $R^\nu$ and the transition function $P^\nu$.

These findings indicate that single-agent IRL theory [Cao et al., 2021, Rolland et al., 2022] can be applied. However, note that for identifiability in single-agent settings at least two environments with different transition dynamics and discount factors that induce experts with the same reward functions are necessary. In multi-agent systems, one obtains a different transition model by varying the policy of the opponent and observing the best responds to this change of dynamics. This can be less restrictive than requiring a new environment as in the single-agent case. Next, we state the result for the average reward case for multi-agents, this is similar to Theorem 3 by [Rolland et al., 2022].

**Theorem 4.1.** *Let a Markov game be given with two different opponents $\nu_1, \nu_2$ that induce different dynamics $P^{\nu_1}, P^{\nu_2}$ and discount factors $\gamma_1, \gamma_2$. Suppose that in both Games we observe QRE equilibrium policy pairs $(\mu_1, \nu_1)$ and a different $\nu_2$ with a best responding policy $\mu_2$ such that they*

*have same average reward functions $R^{\nu_1} = R^{\nu_2}$. Additionally, define $P_a^{\nu_i} \in \mathbb{R}^{\mathcal{S} \times \mathcal{S}}$ the induced transition matrix of expert $i \in \{1,2\}$. Then, the* average *reward player 1 receives can be recovered up to a constant if and only if*

$$
rank \begin{pmatrix} I - \gamma_1 P_{a_1}^{\nu_1} & I - \gamma_2 P_{a_1}^{\nu_2} \\ \vdots & \vdots \\ I - \gamma_1 P_{a_{|\mathcal{A}|}}^{\nu_1} & I - \gamma_2 P_{a_{|\mathcal{A}|}}^{\nu_2} \end{pmatrix} = 2|\mathcal{S}| - 1. \tag{3}
$$

*Analogously this holds for player 2.*

Therefore, for the average case this closely resembles the single-agent case. However, if we want to estimate the underlying problem, which is the more realistic setting, things change. In particular the error in the estimated induced transition $\|P^{\nu} - P^{\tilde{\nu}}\|$ is bounded by terms dependent on the policy estimation error and the underlying transition model error. Using the $L1$ norm we receive for a given $(s, a) : \|P^{\nu}(\cdot \mid s, a) - P^{\tilde{\nu}}(\cdot \mid s, a)\| \leq \|\nu(\cdot \mid s) - \hat{\nu}(\cdot \mid s)\|_1 + \max_{b'} \|P(s, a, b') - \hat{P}(\cdot \mid s, a, b')\|_1$. We can derive the following sample complexity for this.

**Theorem 4.2** (Sample Complexity for Induced Transitions). *To estimate the induced transition model $P^{\nu}$ for Player 1 such that the maximum $L_1$ error over all $(s, a)$ rows is bounded by $\varepsilon$ with probability at least $1 - \delta$, the total number of samples $N_{total}$ is in the order of $\mathcal{O}\left(\frac{|\mathcal{S}||\mathcal{A}||\mathcal{B}|}{\varepsilon^2}\right)$.*

With this theorem we can recover the same result of the single agent case. In particular, we get for every player that if the estimated transition model satisfies Eq. (3), then also the true transition matrices satisfies this condition given that a condition on the second smallest eigenvalue holds true. We state the complete theorem in Appendix F.

**Reward identification in linearly separable Markov games.** To get an identifiable reward not only in the average sense one needs to disentangle the reward from player 2's strategy. This shows that the multi-agent case is fundamentally harder than then the single-agent case. Intuitively, the reason is that the definition of the NE only ensures optimality against a fixed opponent strategy. Additionally, note that the reward function is a matrix of size $|\mathcal{S}||\mathcal{A}||\mathcal{B}|$ while the optimal policy is only defined on $|\mathcal{S}||\mathcal{A}|$ and $|\mathcal{S}||\mathcal{B}|$ respectively.

A potential assumption that disentangles the joint dependency of the reward is fulfilled in *linearly separable reward Markov games*, first introduced in the seminal work of Parthasarathy et al. [1984]. This framework has also been leveraged in more recent studies (e.g., Pérolat et al. [2021]). A reward function is said to be *linearly separable* if it can be decomposed into two independent terms, each depending solely on one player's action $R^1(s, a, b) = R_A^1(s, a) + R_B^1(s, b)$. This formulation immediately mitigates the complexity introduced by the product space dependency, as the reward function now operates over individual action spaces instead. Now, we can rewrite the condition on a reward for a specific state action pair $(s, a)$. We give the formulation directly for two observed environments, meaning that for every $(s, a) \in \mathcal{S} \times \mathcal{A}$

$$
R_A(s, a) = \lambda \log(\mu_1^*(a \mid s)) + V_1(s) - \gamma \sum_{s'} P^{\nu_1^*}(s' \mid, s, a) V_1(s') - \sum_{b \in \mathcal{B}} \nu_1^*(b \mid s) R_B(s, b)
$$
$$
= \lambda \log(\mu_2^*(a \mid s)) + V_2(s) - \gamma \sum_{s'} P^{\nu_2^*}(s' \mid, s, a) V_2(s') - \sum_{b \in \mathcal{B}} \nu_2^*(b \mid s) R_B(s, b) \quad (4)
$$

We can observe the following two cases. If we have two environments for which the reward $R_A(s, a)$ is the same, and player 1 faces the same opponent strategy $\nu^*$, but different transition dynamics $P_1$ and $P_2$, then we notice that $\sum_{b \in \mathcal{B}} \nu^*(b \mid s) R_B(s, b)$ cancels out. Instead if we have different opponent policies $\nu_1^* \neq \nu_2^*$, then we get a new rank requirement for the resulting system of equation. We give a detailed discussion in Appendix F and summarize the findings in the following proposition.

**Proposition 4.3** (Identifiability with Linearly Separable Rewards). *Let two Markov games with linearly separable rewards be given. Then, the resulting system of equations from Eq. (4) is solvable, i.e. the reward is identifiable up to a constant, if the matrix has rank $2|\mathcal{S}| - 1$ and $\nu_1^* = \nu_2^*$ or rank $2|\mathcal{S}|(|\mathcal{B}| + 1) - 1$ in case $\nu_1^* \neq \nu_2^*$ and for one action $b_0 \in \mathcal{B}$ we have $R_B(s, b_0) = 0$.*

Summarized, we have shown that identifiability is not possible in general in the multi-agent setting for a particular $R(s, a, b)$. Additionally, we have given scenarios and conditions that make identifiability possible.

# 5 Conclusion and future work

We formalized Multi-agent Inverse Reinforcement Learning (MAIRL), highlighting its unique challenges over single-agent IRL, particularly the insufficiency of single equilibrium observations for meaningful reward construction due to equilibrium multiplicity and selection ambiguity. To address this, we focused on regularized Markov games to ensure equilibrium uniqueness. In this setting, we extended single-agent IRL bounds Metelli et al. [2021] to analyze error propagation from these estimations to the recovered reward set, resulting in a sample complexity bound for the Uniform Sampling algorithm for Quantal Response Equilibria. Additionally, we addressed the question of reward identifiability in multi-agent systems. This work gives theoretical foundation of MAIRL and opens many potential directions for future work. We outline a few of them next.

**Removing assumption of generative model.**    In the provided sample complexity analysis we have assumed having to a generative model and sampled uniformly from this. While this is a common assumption in initial works also in single-agent IRL [Metelli et al., 2021, Lindner et al., 2022, Metelli et al., 2023], this can be restrictive in general. Therefore, it could be of interest to explore the offline setting as done in [Lazzati et al., 2024b, Zhao et al., 2024] for multi-agents. However, this might not be easy as in offline learning for multi-agents different coverage assumptions are needed compared to the single-agent setting even in case of reward knowledge [Cui and Du, 2022, Zhong et al., 2022]. Another direction would be to consider the active exploration direction, meaning how to construct policies that actively the environment as done in Lindner et al. [2022] for the single-agent case.

**Designing algorithms.**    Our analysis showed that single Nash equilibrium observations allow for an uninformative feasible reward set. One potential algorithmic implication of this finding could be to design an algorithm that guarantees that the observed NE is the only equilibrium under this reward function. This algorithm could be in a similar spirit as the Max-Gap IRL algorithm for single-agents [Ratliff et al., 2006b], but now on an equilibrium level instead of a value-based design.

**Observing multiple equilibria.**    As pointed out in the discussion around Definition 3.5 and in Fu et al. [2021], if one would be able to observe multiple or all equilibria from the set of Nash equilibria of the underlying game, the characterized feasible reward set is more meaningful. As this might be restrictive in practice it could give important theoretical insights.

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

# Contents of Appendix

This appendix provides supplementary material to support the main findings of the paper. First, we give an overview of the used notations, including some additional notations needed for the proofs in the appendix. Then, we present the omitted analysis for the feasible reward set under a single equilibrium observation in Appendix C. We then present the complete proofs for key results for the regularized Markov game setting in Appendix E. Afterwards, in Appendix F we give the missing proofs for the identifiability section. Then, we give the details for our presented numerical validations and some additional experiments that show that MAIRL can be superior to BC. Finally, the appendix compiles a list of technical results, along with their proofs, that are referenced throughout this work. For a better overview we provide a table of contents.

# A    Notation and Symbols

In this part of the appendix, we include notation used in the main paper and some additional notation used for the proofs in the appendix.

| Notation | Description |
|---|---|
| $\mathcal{X}$ | Finite set |
| $\mathbb{R}^{\mathcal{X}}$ | Set of all functions mapping from $\mathcal{X}$ to $\mathbb{R}$ |
| $\Delta^{\mathcal{X}}$ | Set of probability measures over $\mathcal{X}$ |
| $[n]$ | Set $\{1, ..., n\}$ |
| $(\mathcal{X}, d)$ | (Pre)metric space |
| $\mathcal{H}_d$ | Hausdorff (pre)metric |
| $\mathcal{G}$ | Markov game without reward function $(n, \mathcal{S}, \mathcal{A}, P, \gamma, \rho)$ |
| $n$ | Number of players |
| $\mathcal{S}$ | Finite state space |
| $\mathcal{A}^i$ | Action space for player $i$ |
| $\mathcal{A}$ | Joint action space $\mathcal{A}^1 \times ... \times \mathcal{A}^n$ |
| $P(s'\|s, a)$ | Transition model (probability of next state $s'$ given state $s$ and joint action $a$) |
| $\gamma$ | Discount factor |
| $\rho$ | Initial state distribution |
| $\pi^i$ | Policy (strategy) for player $i : \mathcal{S} \to \Delta^{\mathcal{A}^i}$ |
| $\boldsymbol{\pi}$ | Joint policy $(\pi^1, ..., \pi^n) = (\pi^i, \pi^{-i})$ |
| $\boldsymbol{\pi}(\mathbf{a}\|s)$ | Probability of joint action $a$ under joint policy $\pi$ in state $s$, $\prod_{j=1}^n \pi^j(a^j\|s)$ |
| $\Pi^i$ | Set of all policies for agent $i$ |
| $\Pi$ | Set of all joint policies |
| $\bar{w}_{s,a}^{\boldsymbol{\pi},\rho}$ | Discounted probability of visiting state-action pair $(s, a)$ starting from $\rho$ under $\boldsymbol{\pi}$ |
| $R^i(s, a)$ | Reward function for agent $i : \mathcal{S} \times \mathcal{A} \to [-R_{max}^i, R_{\max}^i]$ |
| $R$ | Joint reward function $(R^1, ..., R^n)$ |
| $R_{\max}$ | Maximum absolute reward across all agents, $\max_{i\in[n]} R_{max}^i$ |
| $\mathcal{G} \cup R$ | Standard Markov game with reward |
| $Q_{\mathcal{G}\cup R}^{i,\pi}(s, a)$ | Q-function for agent $i$ under policy $\pi$ |
| $V_{\mathcal{G}\cup R}^{i,\pi}(s)$ | Value function for agent $i$ under policy $\pi$ |
| $\boldsymbol{\pi}^{\text{Nash}}$ | Nash Equilibrium policy |
| $\Pi^{Nash}(R)$ | Set of Nash Equilibria for reward $R$ |
| $V_{\lambda}^{i,(\mu,\nu)}(s)$ | Value function for player $i$ in $\lambda_i$-entropy regularized MG |
| $Q_{\lambda}^{i,(\mu,\nu)}(s, a, b)$ | Q-function for player $i$ in $\lambda_i$-entropy regularized MG |
| $(\mathcal{G}, \boldsymbol{\pi}^{\text{Nash}})$ | MAIRL problem definition |
| $(\hat{\mathcal{G}}, \hat{\boldsymbol{\pi}}^{\text{Nash}})$ | Recovered MAIRL problem |
| $\mathcal{R}_{(\mathcal{G},\pi^{Nash})}$ | Feasible reward set for a single observed NE $\pi^{Nash}$ |
| $\hat{\pi}$ | Policy recovered from MAIRL problem |
| $\mathcal{E}(\hat{\pi})$ | Nash Imitation Gap for MAIRL |
| $\mathcal{R}_{(\mathcal{G},\Pi^{Nash})}$ | Feasible reward set for the set of all NE $\Pi^{Nash}$ |
| $\mu^*, \nu^*$ | QRE equilibrium policies |
| $R^{\nu}(s, a)$ | Average reward for player 1 when player 2 plays $\nu$: $\sum_{b'\in\mathcal{B}} \nu(b'\|s) R(s, a, b')$ |
| $P^{\nu}(s'\|s, a)$ | Induced transition for player 1 when player 2 plays $\nu$: $\sum_{b'\in\mathcal{B}} \nu(b'\|s) P(s'\|s, a, b')$ |
| $R_A^1(s, a) + R_B^1(s, b)$ | Linearly separable reward for player 1 |
| $\Delta_{min}$ | Minimum probability bound for QRE policies |
| $N_k(s, a, b, s')$ | Count of visits to $(s, a, b, s')$ up to iteration $k$ |
| $N_k(s, a, b)$ | Count of visits to $(s, a, b)$ up to iteration $k$ |
| $N_k(s, a)$ | Count of player 1 taking action $a$ in state $s$ up to iteration $k$ |
| $N_k(s, b)$ | Count of player 2 taking action $b$ in state $s$ up to iteration $k$ |
| $N_k(s)$ | Count of visits to state $s$ up to iteration $k$ |
| $\hat{P}_k(s'\|s, a, b)$ | Empirical estimate of transition probability at iteration $k$ |
| $\hat{\mu}_k(a\|s)$ | Empirical estimate of policy $\mu$ at iteration $k$ |
| $\hat{\nu}_k(b\|s)$ | Empirical estimate of policy $\nu$ at iteration $k$ |

In this section, we introduce the additional notation needed for the matrix expression of the Q-function, the value function, and in particular, for an additional implicit condition for the feasible reward function (Theorem C.1) similar to the one derived in Lin et al. [2018]. To achieve this we use a similar notation from Lin et al. [2018], adjusted to this work. First, we introduce for every agent $i \in [n]$ the stacked reward $\boldsymbol{R}^i$. For every state $s \in \mathcal{S}$ the reward can be seen as a matrix of dimension $|\mathcal{A}^i| \times |\prod_{j \neq i}^n |\mathcal{A}^j|$. Doing this for every state and stacking them, results in a vector $\boldsymbol{R}^i \in \mathbb{R}^{|\mathcal{S}||\mathcal{A}|}$, $|\mathcal{A}|$ is the dimension of the joint action space. We additionally introduce the operator $\boldsymbol{\pi}$, which can be written as a $|\mathcal{S}| \times |\mathcal{S}||\mathcal{A}|$ matrix, structured in the following way. First, we need to fix an arbitrary order on the joint action space $[|\mathcal{A}|]$ in the same way as already done for stacking the Reward for every agent. Given the order, we have that for $k \in [|S|]$, the $k$-th row is given by

$$\Phi_1^\pi(k), \ldots, \Phi_{|\mathcal{A}|}^\pi(k),$$

where for $j \in [|\mathcal{A}|]$ we have

$$\Phi_j^\pi(k) = \left[ \underbrace{0, \ldots, 0}_{k-1}, \prod_{i=1}^n \pi^i(a_j^i \mid k), \underbrace{0, \ldots, 0}_{|\mathcal{S}|-k} \right].$$

Therefore, the resulting matrix has in its first $|\mathcal{S}|$ columns a diagonal matrix of size $|\mathcal{S}| \times |\mathcal{S}|$ with the corresponding probabilities of playing the first joint action in all possible states.

$$\begin{pmatrix} \prod_{i=1}^n \pi^i(a_1 \mid 1) & 0 & 0 & \cdots & 0 & \cdots \\ 0 & \prod_{i=1}^n \pi^i(a_1 \mid 2) & 0 & \cdots & 0 & \cdots \\ 0 & 0 & \prod_{i=1}^n \pi^i(a_1 \mid 3) & \cdots & 0 & \cdots \\ \vdots & \vdots & \vdots & \ddots & \vdots & \cdots \\ 0 & 0 & 0 & \cdots & \prod_{i=1}^n \pi^i(a_1 \mid S) & \cdots \end{pmatrix}$$

The transition matrix $\boldsymbol{P}$ of a Markov game also depends on the joint actions, making the resulting transition matrix of dimension $|\mathcal{S}||\mathcal{A}| \times \mathcal{S}$. This allows us to write the value function as a column vector of dimension $\mathbb{R}^{|\mathcal{S}|}$ and the Q-value function as a vector, identically as the reward vector, of dimension $|\mathcal{S}||\mathcal{A}| \times 1$. Therefore, we can write:

$$\boldsymbol{Q}^{i,\boldsymbol{\pi}} = \boldsymbol{R}^i + \gamma \boldsymbol{P} \boldsymbol{V}^{i,\boldsymbol{\pi}}, \quad \boldsymbol{V}^{i,\boldsymbol{\pi}} = \boldsymbol{\pi} \boldsymbol{Q}^{i,\boldsymbol{\pi}}.$$

# B  Related Work

This work intersects with several fields of research, particularly **Inverse Reinforcement Learning**, **Multi-agent Inverse Reinforcement Learning** , and **Inverse (Algorithmic) Game Theory**.

**Theoretical Understanding of IRL.** IRL was first introduced by Ng and Russell [2000], emphasizing its ill-posed nature. Subsequent work tackled ambiguity via reformulations [Abbeel and Ng, 2004, Ziebart et al., 2008, Ramachandran and Amir, 2007, Ratliff et al., 2006a, Levine et al., 2011]. To avoid the ambiguity in IRL, recent research has addressed to characterize the set of feasible rewards instead of picking a single reward function. Theoretical efforts to characterize the feasible reward set were pioneered by Metelli et al. [2021]. The authors provide an explicit reward formulation, that shows that the reward depends on the expert policy and the transition dynamics. As in realistic scenarios it is not common to know the transition model and the expert policy, the authors provide an error propagation analysis on how estimation errors in these quantities transfer to the recovered reward. Additionally, they provide a uniform sampling algorithm with access to a generative model combined with a sample complexity analysis on how many samples are required to find a suitable reward function from the set of feasible rewards, that is also transferable to new environments. In Lindner et al. [2022] the authors extend this to the finite horizon setting. Additionally, the authors provide the first algorithm that removes the assumption of a generative model and instead create exploration policies to mitigate the reward uncertainty dubbed active inverse reinforcement learning. Additionally, they provide a sample complexity analysis for the most general case and a problem-dependent variant. Further insights on the theoretical insights of IRL have been provided by [Metelli et al., 2023]. The authors investigate different metrics for the IRL problem, leading to a more nuanced analysis and the requirement of refined concentration inequalities. Additionally, they provide the first lower bound for IRL, addressing an open question , that IRL is not harder than forward RL. The first offline algorithm for IRL combined with a sample complexity analysis has been provided in [Zhao et al., 2024]. The authors note limitations of so far introduced metrics in settings without a generative model. Additionally, the authors show that IRL is not harder than standard RL in the offline setting. The offline setting has also been considered in Lazzati et al. [2024b]. The authors provide a new formulation of the feasible reward set, more suitable for the offline setting. They introduce two new efficient algorithms designed for the offline setting, overcoming the new introduced challenges as the data coverage cannot be controlled anymore. An investigation how IRL translates to large state spaces has been obtained in Lazzati et al. [2024a]. The authors provide the negative result, that the feasible reward set cannot be learned efficiently in large state spaces without additional assumptions. Instead of the feasible reward set, they provide a new framework, rewards compatibility and an efficient algorithm for this setting.

While these works analyze distances to value functions and expert policies, applying them to Markov games remains challenging due to the need for equilibrium-based solutions. This implies that the standard objectives for IRL, namely value based gaps, cannot be applied in multi-agent settings. Additionally, we show that MAIRL introduces new challenges due to the multiplicity of equilibria.

Another line of works, considers the case of reward identifiability. In Cao et al. [2021] the authors give an explicit reward characterization in the setting of regularized MDPs. In particular, the authors note that the value function can be chosen arbitrarily for a given optimal policy and therefore identifiability is not possible by a single expert policy. However, if one considers two MDPS with different transition dynamics and discount factors and the same optimal reward function, then identifiablity is possible if the MDPs are value-distinguishable. Based on these observations, Rolland et al. [2022] derived explicit conditions what value-distinguishable MDPs are. In particular, they rewrote the reward identifiability problem as system of equations and derived rank conditions that need to be fulfilled to identify the rewards up to constants. Additionally, they provide insights for the case with unknown transition functions and transferability to new environments.

In [Kim et al., 2021] the authors study the type of MDPs under which reward identifiability is possible. Considering a deterministic MDP with an entropy regularized objective, the authors provide necessary and sufficient conditions whether and MDP is identifiable. Additionally, building on these findings they provide efficient algorithms to check if an MDP is identifiable.

In this work we address the question of identifiability in the context of multi-agents. We show that it is not possible to identify the reward function up to constants without additional assumptions. Instead

it is only possible to obtain an average reward function or one poses additional structural assumptions on the underlying Markov game as e.g. linear separable rewards.

**Multi-agent Inverse Reinforcement Learning.** The first extension of IRL to multi-agent settings was introduced by Natarajan et al. [2010], focusing on a centralized controller in an average reward RL framework, but without addressing competitive settings requiring game-theoretic solutions. Lin et al. [2014] extended this to Zero-Sum Markov games, introducing a Bayesian MAIRL framework based on observed Nash equilibria, later expanded by Lin et al. [2018] to incorporate various solution concepts, though without sample complexity bounds. Yu et al. [2019] extended Maximum Entropy IRL to multi-agent settings via the logistic best response equilibrium, focusing on recovering a single reward function rather than analyzing the feasible reward set. More recently, Goktas et al. [2024] explored Inverse Multi-agent Learning with parameter-dependent payoffs, simplifying the problem by assuming access to samples from the reward function. Fu et al. [2021] approached MAIRL by decomposing it into multiple single-agent IRL tasks, applying utility-matching IRL algorithms on the induced MDPs. Additionally, this work is the first that notes that single Nash equilibrium observations can be limited. Instead of considering a single reward function, we formalize that that single equilibrium observations lead to an uninformative feasible reward set. Additionally, their approach does not address equilibrium multiplicity and lacks a sample complexity analysis. In a concurrent work, Liao et al. [2025] address reward-function recovery in two-player zero-sum games with entropy regularization. They first analyze a static (matrix) game under a linear parametrization of the payoff matrix and derive a rank-condition on the feature kernel under which the reward parameters are uniquely identifiable. They then extend their framework to linear Markov games (entropy-regularized zero-sum Markov decision processes) and propose algorithms with high-probability finite-sample guarantees for recovering the feasible set of reward (and transition) parameters. Finally, Tang et al. [2024] highlighted the inadequacy of value-based gaps in Multi-agent Imitation Learning, proposing regret as a more suitable objective when observing Correlated Equilibrium experts. In this work, we consider the Multi-agent Inverse Reinforcement Learning framework and consider the Nash equilibrium as well as the QRE.

**Inverse (Algorithmic) Game Theory.** There is significant overlap between *Multi-agent Inverse Reinforcement Learning* and *inverse algorithmic game theory*. Many works in this area apply game-theoretic solution concepts to rationalize the behavior of observed players in specific types of games [Kalyanaraman and Umans, 2008, 2009]. A related work is by Kuleshov and Schrijvers [2015], who developed polynomial-time algorithms for coarse correlated equilibria in succinct games, where the structure of the game is known and noted that in cases where the game structure is unknown, the problem is NP-hard. Their theorems indicate that without additional assumptions or more specific settings, polynomial-time algorithms cannot be expected for inversely solving Nash equilibria. In a more recent work, [Wu et al., 2022] introduce bounded rationality, i.e. considering QRE as the observed behavior, in the context of Stackelberg Games. In particular, the authors show that QRE observations are more informative than Nash equilibrium observations. This means that bounded rationality helps to be more robust against an irrational opponent. Additionally, this makes it possible to construct algorithms that have exponential dependencies on neither the number of leader actions nor the number of follower actions.

## C   Omitted analysis for Section 3

The first theorem serves as an extension of the two player version theorem by Lin et al. [2018] (see section 4.6 in Lin et al. [2018]) to $n$-person games and general Nash equilibria. It makes use of the notation introduced in Appendix A.

**Theorem C.1.** *Let $\mathcal{G} \cup R$ be a $n$-person general-sum Markov game. A policy $\boldsymbol{\pi}$ is an NE strategy if and only if*

$$(\boldsymbol{\pi}^{\mathrm{Nash}} - \tilde{\boldsymbol{\pi}})(I - \gamma \boldsymbol{P} \boldsymbol{\pi}^{\mathrm{Nash}})^{-1} \boldsymbol{R}^i \geq 0.$$

*with the meaning that without $(s, a)$ symbols a matrix notation and $\tilde{\boldsymbol{\pi}}$ is the policy with $\pi^{-i} = \pi^{-i,\mathrm{Nash}}$ and $\pi^i$ plays action $a$ with probability 1.*

*Proof.* In the first step of the proof we state the theorem for the case where $n = 2$ with the use of the definition of an NE. We only write the condition for agent 1 to understand the structure. For every

action $a^1 \in \mathcal{A}^1$ and every state $s \in \mathcal{S}$ it must hold true that:

$$\sum_{a^2 \in \mathcal{A}^2} \pi^{2,\text{Nash}}(a^2 \mid s) R^1(s, a^1 a^2) + \gamma \sum_{a^2 \in \mathcal{A}^2} \pi^{2,\text{Nash}}(a^2 \mid s) \sum_{s'} P(s' \mid s, a^1 a^2) V^{\boldsymbol{\pi}^{\text{Nash}}}(s) \leq V^{\boldsymbol{\pi}^{\text{Nash}}}(s)$$

If we now want to generalize this to a $n$-person Markov game, we get that for every player $i \in [n]$, every action $a^i$ and every state $s \in \mathcal{S}$ it must hold true that:

$$\sum_{a^{-i} \in \mathcal{A}^{-i}} \pi^{-i,\text{Nash}}(a^{-i} \mid s) R^1(s, a^i a^{-i})$$

$$+ \gamma \sum_{a^{-i} \in \mathcal{A}^{-i}} \pi^{-i,\text{Nash}}(a^{-i} \mid s) \sum_{s'} P(s' \mid s, a^i a^{-i}) V^{\boldsymbol{\pi}^{\text{Nash}}}(s) \leq V^{\boldsymbol{\pi}^{\text{Nash}}}(s)$$

We can rewrite this equation in terms of the Q-function and get

$$\sum_{a^{-i} \in \mathcal{A}^{-i}} \pi^{-i}(a^{-i} \mid s) Q^{\boldsymbol{\pi}^{\text{Nash}}}(s, \mathbf{a}) \leq V^{\boldsymbol{\pi}^{\text{Nash}}}(s). \tag{5}$$

Now we want to rewrite the equation for all states simultaneously. Therefore we recall the notation introduced in Appendix A. We have that

$$\boldsymbol{Q}^{i,\boldsymbol{\pi}} = \boldsymbol{R}^i + \gamma \boldsymbol{P} \boldsymbol{V}^{i,\boldsymbol{\pi}}, \quad \boldsymbol{V}^{i,\boldsymbol{\pi}} = \boldsymbol{\pi} \boldsymbol{Q}^{i,\boldsymbol{\pi}}.$$

Rewriting this equation for the Nash Policy $\boldsymbol{\pi}^{\text{Nash}}$ gives us

$$\boldsymbol{Q}^{i,\boldsymbol{\pi}^{\text{Nash}}} = (I - \gamma \boldsymbol{P} \boldsymbol{\pi}^{\text{Nash}})^{-1} \boldsymbol{R}^i.$$

Plugging in the derived equations in (5) using matrix notation for all states $s \in \mathcal{S}$ simultaneously and additionally denote the joint policy, where agent $i$ plays action $a^i$ with probability 1 and the other agents execute their Nash strategy $\pi^{-i,\text{Nash}}$ as $\tilde{\boldsymbol{\pi}}$, we get

$$(\tilde{\boldsymbol{\pi}} - \boldsymbol{\pi}^{\text{Nash}})(I - \gamma \boldsymbol{P} \boldsymbol{\pi}^{\text{Nash}})^{-1} \boldsymbol{R}^i \leq 0.$$

$\square$

The next lemma restates the condition by directly using the expectation of the advantage function with respect to the policy.

**Lemma C.2** (Feasible Reward Set Implicit). *A reward function $R = (R^1, \ldots, R^n)$ is feasible if and only if for a Nash policy $\boldsymbol{\pi}^{\text{Nash}}$, for every agent $i \in [n]$ and all $(s, a^i) \in \mathcal{S} \times \mathcal{A}^i$, it holds true that:*

$$\sum_{a^{-i} \in \mathcal{A}^{-i}} \pi^{-i,\text{Nash}}(a^{-i} \mid s) A^{i,\boldsymbol{\pi}^{\text{Nash}}}_{\mathcal{G} \cup R}(s, a^i, a^{-i}) = 0, \text{if } \pi^{i,\text{Nash}}(a^i \mid s) > 0, a^{-i} \in \text{supp}(\pi^{-i,\text{Nash}}(\cdot \mid s)).$$

$$\sum_{a^{-i} \in \mathcal{A}^{-i}} \pi^{-i,\text{Nash}}(a^{-i} \mid s) A^{i,\boldsymbol{\pi}^{\text{Nash}}}_{\mathcal{G} \cup R}(s, a^i, a^{-i}) \leq 0, \text{if } \pi^{i,\text{Nash}}(a^i \mid s) = 0, a^{-i} \in \text{supp}(\pi^{-i,\text{Nash}}(\cdot \mid s)).$$

*Proof.* As we know that $a^{-i} \in \text{supp}(\pi^{-i,\text{Nash}}(\cdot \mid s))$ for both cases, we get for all agents $i \in [n]$ and all actions $a^{i,Nash} \in \mathcal{A}^i$ that fulfill $\pi^{i,\text{Nash}}(a^{i,Nash} \mid s) > 0$, that $\sum_{a^{-i} \in \mathcal{A}^{-i}} \pi^{-i,\text{Nash}}(a^{-i} \mid s) Q^{i,\boldsymbol{\pi}^{\text{Nash}}}(s, a^{i,\text{Nash}} a^{-i}) > \sum_{a^{-i} \in \mathcal{A}^{-i}} \pi^{-i,\text{Nash}}(a^{-i} \mid s) Q^{i,\boldsymbol{\pi}^{\text{Nash}}}(s, a^i a^{-i})$. Additionally, we have that for all $a^{i,\text{Nash}}$ with $\pi^{i,\text{Nash}}(a^{i,\text{Nash}} \mid s) > 0$ that $\sum_{a^{-i} \in \mathcal{A}^{-i}} \pi^{-i,\text{Nash}}(a^{-i} \mid s) Q^{i,\boldsymbol{\pi}^{\text{Nash}}}(s, a^{i,\text{Nash}} a^{-i}) = V^{i,\boldsymbol{\pi}^{\text{Nash}}}(s)$. $\square$

In the following we will constrain to the case of pure NE, therefore it holds true, that for some $a^{-i} \in \mathcal{A}^{-i}$, we have $\pi(a^{-i} \mid s) = 1 \, \forall s \in \mathcal{S}$.

**Lemma C.3.** *Let $i \in [n]$ be an arbitrary agent. Then the Q-function of player $i$ satisfies the optimality conditions of Lemma C.2 for a pure Nash equilibrium if and only if for every $(s, \boldsymbol{a}) \in \mathcal{S} \times \mathcal{A}$ there exists a function $A^i \in \mathbb{R}^{\mathcal{S} \times \mathcal{A}}_{\geq 0}$ and $V^i \in \mathbb{R}^{\mathcal{S}}$ such that:*

$$Q^{i,\boldsymbol{\pi}^{\text{Nash}}}_{\mathcal{G} \cup R}(s, \boldsymbol{a}) = -A^i(s, \boldsymbol{a}) \mathbf{1}_{\{\pi^{i,\text{Nash}}(a^i \mid s) = 0\}} \mathbf{1}_{\{\pi^{-i,\text{Nash}}(a^{-i} \mid s) = 1\}} + V^i(s)$$

*Proof.* First we assume that the Q-function can be expressed as

$$Q_{\mathcal{G}\cup R}^{i,\boldsymbol{\pi}^{\text{Nash}}}(s,\mathbf{a}) = -A^i(s,\mathbf{a})\mathbf{1}_{\{\pi^{i,\text{Nash}}(a^i|s)=0\}}\mathbf{1}_{\{\pi^{-i,\text{Nash}}(a^{-i}|s)=1\}} + V^i(s).$$

We note that

$$\begin{aligned}
V_{\mathcal{G}\cup R}^{i,\boldsymbol{\pi}^{\text{Nash}}}(s) &= \sum_{\mathbf{a}\in\mathcal{A}} \boldsymbol{\pi}^{\text{Nash}}(\mathbf{a}\mid s)Q_{\mathcal{G}\cup R}^{i,\boldsymbol{\pi}^{\text{Nash}}}(s,\mathbf{a}) \\
&= \sum_{\mathbf{a}\in\mathcal{A}} \boldsymbol{\pi}^{\text{Nash}}(\mathbf{a}\mid s)\left(-A^i(s,\mathbf{a})\mathbf{1}_{\{\pi^{i,\text{Nash}}(a^i|s)=0\}}\mathbf{1}_{\{\pi^{-i,\text{Nash}}(a^{-i}|s)=1\}} + V^i(s)\right) \\
&= V^i(s),
\end{aligned}$$

where the last equality follows from the fact, that if $\boldsymbol{\pi}^{\text{Nash}}(\mathbf{a}\mid s) > 0$, then $\mathbf{1}_{\{\pi^{i,\text{Nash}}(a^i|s)=0\}} = 0$ and vice versa. Additionally, $V^i(s)$ is independent of $\mathbf{a}$ and as the sum is over the joint action space it holds true that $\sum_{\mathbf{a}\in\mathcal{A}}\boldsymbol{\pi}^{\text{Nash}}(\mathbf{a}\mid s) = 1$. We now have to consider two cases. The first one is if $\mathbf{1}_{\{\pi^{i,\text{Nash}}(a^i|s)=0\}} = 0$ and $\mathbf{1}_{\{\pi^{-i,\text{Nash}}(a^{-i}|s)=1\}} = 1$. Then it holds true that

$$\sum_{\mathbf{a}\in\mathcal{A}}\boldsymbol{\pi}^{\text{Nash}}(\mathbf{a}\mid s)Q_{\mathcal{G}\cup R}^{i,\boldsymbol{\pi}^{\text{Nash}}}(s,\mathbf{a}) = V^i(s) = V_{\mathcal{G}\cup R}^{i,\boldsymbol{\pi}^{\text{Nash}}}.$$

The second case is if $\mathbf{1}_{\{\pi^{i,\text{Nash}}(a^i|s)=0\}} = 1$ and $\mathbf{1}_{\{\pi^{-i,\text{Nash}}(a^{-i}|s)=1\}} = 1$ for one action $\tilde{a}^{-i}$ with $\pi^{-i}(\tilde{a}^{-i}\mid s)$ as we assumed it is a pure NE. Then it holds true that

$$\begin{aligned}
\sum_{\mathbf{a}\in\mathcal{A}}&\pi^{-i,\text{Nash}}(a^{-i}\mid s)Q_{\mathcal{G}\cup R}^{i,\boldsymbol{\pi}^{\text{Nash}}}(s,a^i a^{-i}) \\
&= Q_{\mathcal{G}\cup R}^{i,\boldsymbol{\pi}^{\text{Nash}}}(s,a^i\tilde{a}^{-i}) \\
&= -A^i(s,\mathbf{a})\mathbf{1}_{\{\pi^{i,\text{Nash}}(a^i|s)=0\}}\mathbf{1}_{\{\pi^{-i,\text{Nash}}(a^{-i}|s)=1\}} + V^i(s) \\
&\leq V^i(s) = V_{\mathcal{G}\cup R}^{i,\boldsymbol{\pi}^{\text{Nash}}},
\end{aligned}$$

where we used the fact that $-A^i(s,\mathbf{a}) \leq 0$.
If we now assume that the conditions of Lemma C.2 hold, we can set for every $(s,\mathbf{a}) \in \mathcal{S}\times\mathcal{A}$ $V^i(s) = V_{\mathcal{G}\cup R}^{i,\boldsymbol{\pi}^{\text{Nash}}}$ and $A^i(s,\mathbf{a}) = V_{\mathcal{G}\cup R}^{i,\boldsymbol{\pi}^{\text{Nash}}}(s) - Q_{\mathcal{G}\cup R}^{i,\boldsymbol{\pi}^{\text{Nash}}}(s,\mathbf{a})$. $\qquad\square$

**Lemma C.4** (Feasible Reward Set Explicit). *A reward function $R$ is feasible if and only if, for each agent $i \in [n]$, there exist a function $A^i \in \mathbb{R}_{\geq 0}^{\mathcal{S}\times\mathcal{A}}$ and a function $V^i \in \mathbb{R}^{\mathcal{S}}$ such that for all $(s,\boldsymbol{a}) \in \mathcal{S}\times\mathcal{A}$, the following holds:*

$$R^i(s,\boldsymbol{a}) = -A^i(s,\boldsymbol{a})\mathbf{1}_{\{\pi^{i,\text{Nash}}(a^i|s)=0\}}\mathbf{1}_{\{\pi^{-i,\text{Nash}}(a^{-i}|s)=1\}} + V^i(s) - \gamma\sum_{s'}P(s'\mid s,\boldsymbol{a})V^i(s').$$

*Proof.* Remembering that we can express the $Q$-function as $Q_{\mathcal{G}\cup R}^{i,\boldsymbol{\pi}^{\text{Nash}}}(s,\mathbf{a}) = R^i(s,\mathbf{a}) + \gamma\sum_{s'}P(s'\mid s,\mathbf{a})V_{\mathcal{G}\cup R}^{i,\boldsymbol{\pi}^{\text{Nash}}}(s')$ and applying Lemma C.3 to express the $Q$-function for an NE policy, we can conclude

$$\begin{aligned}
R^i(s,\mathbf{a}) &= Q_{\mathcal{G}\cup R}^{i,\boldsymbol{\pi}^{\text{Nash}}}(s,\mathbf{a}) - \gamma\sum_{s'}P(s'\mid s,a)V_{\mathcal{G}\cup R}^{i,\boldsymbol{\pi}^{\text{Nash}}}(s') \\
&= -A^i(s,\mathbf{a})\mathbf{1}_{\{\pi^{i,\text{Nash}}(a^i|s)=0\}}\mathbf{1}_{\{\pi^{-i,\text{Nash}}(a^{-i}|s)=1\}} + V^i(s) - \gamma\sum_{s'}P(s'\mid s,\mathbf{a})V^i(s').
\end{aligned}$$

$\qquad\square$

**Theorem C.5** (Error Propagation). *Let $(\mathcal{G},\boldsymbol{\pi}^{\text{Nash}})$ and $(\hat{\mathcal{G}},\hat{\boldsymbol{\pi}}^{\text{Nash}})$ be the true and the recovered MAIRL problem. Then, for every agent $i \in [n]$ and any $R_i \in \mathcal{R}_{\mathcal{B}}$ there exists $\hat{R}_i \in \mathcal{R}_{\hat{\mathcal{B}}}$ such that:*

$$|R_i(s,\boldsymbol{a}) - \hat{R}_i(s,\boldsymbol{a})| \leq A^i(s,\boldsymbol{a})|\mathbf{1}_E - \mathbf{1}_{\hat{E}}| + \gamma\sum_{s'}V^i(s')|P(s'\mid s,\boldsymbol{a}) - \hat{P}(s'\mid s,\boldsymbol{a})|,$$

*where $E := \{\{\pi^{i,\text{Nash}}(a^i\mid s) = 0\}\cap\{\pi^{-i,\text{Nash}}(a^{-i}\mid s) > 0\}\}$ and $\hat{E} := \{\{\hat{\pi}^{i,\text{Nash}}(a^i\mid s) = 0\}\cap\{\hat{\pi}^{-i,\text{Nash}}(a^{-i}\mid s) = 1\}\}$.*

*Proof.* From the explicit expression of a feasible reward C.4, we know that we can write the reward function of any agent $i \in [n]$ as

$$R^i(s, \mathbf{a}) = -A^i(s, \mathbf{a})\mathbf{1}_{\{\pi^{i,\text{Nash}}(a^i|s)=0\}}\mathbf{1}_{\{\pi^{-i,\text{Nash}}(a^{-i}|s)=1\}} + V^i(s) - \gamma \sum_{s'} P(s' \mid s, \mathbf{a})V^i(s')$$
(6)

$$\hat{R}^i(s, \mathbf{a}) = -\hat{A}^i(s, \mathbf{a})\mathbf{1}_{\{\hat{\pi}^{i,\text{Nash}}(a^i|s)=0\}}\mathbf{1}_{\{\hat{\pi}^{-i,\text{Nash}}(a^{-i}|s)=1\}} + \hat{V}^i(s) - \gamma \sum_{s'} \hat{P}(s' \mid s, \mathbf{a})\hat{V}^i(s')$$
(7)

As pointed out in Metelli et al. [2023], the rewards $\hat{R}^i(s, \mathbf{a})$ do not have to be bounded by the same $R^i(s, \mathbf{a})$ and therefore also not by the same $R_{\max}$. To fix this issue the authors point out, that the reward needs to be rescaled such that the recovered feasible reward set is bounded by the same value. In our case we have to be a bit more careful with the choice of the scaling, as we did not assume that the reward is bounded by 1. As we proof the existence of such reward function, we can choose $\tilde{V}^i(s) = V^i(s)$ for every $s \in \mathcal{S}$ and $\tilde{A}^i(s, \mathbf{a}) = A^i(s, \mathbf{a})$ for every $(s, \mathbf{a}) \in \mathcal{S} \times \mathcal{A}$, which results in a reward

$$\tilde{R}^i(s, \mathbf{a}) = -A^i(s, \mathbf{a})\mathbf{1}_{\{\hat{\pi}^{i,\text{Nash}}(a^i|s)=0\}}\mathbf{1}_{\{\hat{\pi}^{-i,\text{Nash}}(a^{-i}|s)=1\}} + V^i(s) + \gamma \sum_{s'} \hat{P}(s' \mid s, \mathbf{a})V^i(s').$$

Now we need to rescale the reward with $R_{\max} + |\varepsilon^i(s, \mathbf{a})|$, where

$$\varepsilon^i(s, \mathbf{a}) = -A^i(s, \mathbf{a})\left(\mathbf{1}_{\{\pi^{i,\text{Nash}}(a^i|s)=0\}}\mathbf{1}_{\{\pi^{-i,\text{Nash}}(a^{-i}|s)=1\}} - \mathbf{1}_{\{\hat{\pi}^{i,\text{Nash}}(a^i|s)=0\}}\mathbf{1}_{\{\hat{\pi}^{-i,\text{Nash}}(a^{-i}|s)=1\}}\right)$$
$$+ \gamma \sum_{s'}(P(s' \mid s, \mathbf{a}) - \hat{P}(s, \mathbf{a}))V^i(s'),$$

such that it remains bounded by $R_{\max}$, we receive

$$\hat{R}^i(s, \mathbf{a}) = \tilde{R}^i(s, \mathbf{a})\frac{R_{\max}}{R_{\max} + |\varepsilon^i(s, \mathbf{a})|}$$

$$= -A^i(s, \mathbf{a})\frac{R_{\max}}{R_{\max} + |\varepsilon^i(s, \mathbf{a})|}\mathbf{1}_{\{\hat{\pi}^{i,\text{Nash}}(a^i|s)=0\}}\mathbf{1}_{\{\hat{\pi}^{-i,\text{Nash}}(a^{-i}|s)=1\}}$$

$$+ \frac{R_{\max}V^i(s)}{R_{\max} + |\varepsilon^i(s, \mathbf{a})|} + \gamma \sum_{s'} \hat{P}(s' \mid s, \mathbf{a})\frac{R_{\max}V^i(s')}{R_{\max} + |\varepsilon^i(s, \mathbf{a})|}$$

It then follows that:

$$|R^i(s, \mathbf{a}) - \hat{R}^i(s, \mathbf{a})| = |R^i(s, \mathbf{a}) - \frac{R_{\max}\tilde{R}^i(s, \mathbf{a})}{R_{\max} + |\varepsilon^i(s, \mathbf{a})|}|$$

$$\leq \frac{R_{\max}}{R_{\max} + |\varepsilon^i(s, \mathbf{a})|}\left|\left(\frac{R_{\max} + |\varepsilon^i(s, \mathbf{a})|}{R_{\max}}\right)R^i(s, \mathbf{a}) - \tilde{R}^i(s, \mathbf{a})\right|$$

$$\leq \frac{R_{\max}}{R_{\max} + |\varepsilon^i(s, \mathbf{a})|}\left(|R^i(s, \mathbf{a}) - \tilde{R}^i(s, \mathbf{a})| + |\frac{\varepsilon^i(s, \mathbf{a})}{R_{\max}}R^i(s, \mathbf{a})|\right)$$

$$\leq \frac{R_{\max}}{R_{\max} + |\varepsilon^i(s, \mathbf{a})|}\left(|\varepsilon^i(s, \mathbf{a})| + |\varepsilon^i(s, \mathbf{a})|\right) \leq \frac{R_{\max}}{R_{\max}}\left(|\varepsilon^i(s, \mathbf{a})| + |\varepsilon^i(s, \mathbf{a})|\right)$$

$$= 2\left(A^i(s, \mathbf{a})|\left(\mathbf{1}_E - \mathbf{1}_{\hat{E}}\right)| + \gamma\left|\sum_{s'}(P(s' \mid s, \mathbf{a}) - \hat{P}(s' \mid s, \mathbf{a}))V^i(s')\right|\right)$$

$\square$

**Lemma C.6.** *Let* $\mathcal{G} \cup R$ *be a n-person general-sum Markov game,* $P, \hat{P}$ *two transition probabilities and* $R, \hat{R}$ *two reward functions, such that* $\hat{\boldsymbol{\pi}}$ *is a Nash equilibrium strategy in* $\hat{\mathcal{G}} \cup \hat{R}$. *Then, it holds true that:*

$$V^i(\pi^i, \hat{\pi}^{-i}) - V^i(\hat{\pi}_i, \hat{\pi}^{-i})$$

$$\leq \sum_{s, \boldsymbol{a}} \overline{w}_{s,a}^{\hat{\boldsymbol{\pi}}}(R^i(s, \boldsymbol{a}) - \hat{R}^i(s, \boldsymbol{a}) + \gamma \sum_{s'}(\hat{P}(s' \mid s, \boldsymbol{a}) - P(s' \mid s, a)V^{i,\hat{\pi}}(s')))$$

$$+ \sum_{s, \boldsymbol{a}} \overline{w}_{s,a}^{\tilde{\boldsymbol{\pi}}}(R^i(s, \boldsymbol{a}) - \hat{R}^i(s, \boldsymbol{a}) + \gamma \sum_{s'}(\hat{P}(s' \mid s, \boldsymbol{a}) - P(s' \mid s, \boldsymbol{a})V^{i,\tilde{\pi}}(s'))),$$

**Algorithm 1** MAIRL Uniform Sampling Algorithm with Generative Model

---

**Require:** Significance $\delta \in (0,1)$, target accuracy $\varepsilon$
1: Initialize $k \leftarrow 0$, $\varepsilon_0 \leftarrow +\infty$
2: **while** $\varepsilon_k > \varepsilon$ **do**
3:     Generate one sample for each $(s, \mathbf{a}) \in \mathcal{S} \times \mathcal{A}$
4:     Update $\hat{P}_k$ as described in (12)
5:     Update accuracy $\varepsilon_k \leftarrow \frac{1}{1-\gamma} \max_{(s,\mathbf{a})} \hat{C}_k(s, \mathbf{a})$
6: **end while**

---

*where $\tilde{\boldsymbol{\pi}} = (\pi^i, \hat{\pi}^{-i})$.*

*Proof.*

$$V^i(\pi^i, \hat{\pi}^{-i}) - V^i(\hat{\pi}^i, \hat{\pi}^{-i})$$

$$= V^i(\pi^i, \hat{\pi}^{-i}) - \hat{V}^i(\pi^i, \hat{\pi}^{-i}) + \hat{V}^i(\hat{\pi}^i, \hat{\pi}^{-i}) - V^i(\hat{\pi}^i, \hat{\pi}^{-i}) + \hat{V}^i(\pi^i, \hat{\pi}^{-i}) - \hat{V}^i(\hat{\pi}^i, \hat{\pi}^{-i})$$

$$\leq V^i(\pi^i, \hat{\pi}^{-i}) - \hat{V}^i(\pi^i, \hat{\pi}^{-i}) + \hat{V}^i(\hat{\pi}^i, \hat{\pi}^{-i}) - V^i(\hat{\pi}^i, \hat{\pi}^{-i})$$

$$= \sum_{s,\mathbf{a}} \overline{w}_{s,a}^{\hat{\boldsymbol{\pi}}} (R^i(s, \mathbf{a}) - \hat{R}^i(s, \mathbf{a}) + \gamma \sum_{s'} (\hat{P}(s' \mid s, \mathbf{a}) - P(s' \mid s, a)) V^{i,\hat{\pi}}(s'))$$

$$+ \sum_{s,\mathbf{a}} \overline{w}_{s,\mathbf{a}}^{\tilde{\boldsymbol{\pi}}} (R^i(s, \mathbf{a}) - \hat{R}^i(s, \mathbf{a}) + \gamma \sum_{s'} (\hat{P}(s' \mid s, \mathbf{a}) - P(s' \mid s, \mathbf{a})) V^{i,\tilde{\boldsymbol{\pi}}}(s')),$$

where we used that $\hat{V}^i(\hat{\pi}^i, \hat{\pi}^{-i}) - V^i(\hat{\pi}^i, \hat{\pi}^{-i}) \leq 0$ as $\hat{\boldsymbol{\pi}}$ is a NE policy and in the last equation we applied H.2. $\qquad \square$

### C.1 Sample Complexity analysis of the Uniform Sampling algorithm

In this section we give the proofs of the sample complexity for Uniform Sampling algorithm and lemmas derived in Theorem C.12 and Lemma C.8 The structure is as follows:

1. We first state the Uniform Sampling algorithm.
2. Then, we state the optimality criterion based on the Nash Imitation Gap.
3. Next, we present the Good Event Lemma bounds, using Hoeffding's inequality.
4. We define the reward uncertainty.
5. Then, we state a theorem that provides conditions—dependent on the derived confidence bounds—under which the optimality criterion holds.
6. Finally, we consolidate all results to prove the sample complexity bound for the uniform sampling algorithm.

The algorithm, that we are evaluating in this section is given in Algorithm 1.

Now, we can restate the optimality criterion of the algorithm.

**Definition C.7** (Optimality Criterion). Let $\mathcal{R}_{(\mathcal{G}, \boldsymbol{\pi}^{\mathrm{Nash}})}$ be the exact feasible set and $\mathcal{R}_{(\hat{\mathcal{G}}, \hat{\boldsymbol{\pi}}^{\mathrm{Nash}})}$ the recovered feasible set after sampling $N \geq 0$ from $(\mathcal{G}, \boldsymbol{\pi}^{\mathrm{Nash}})$. We consider an algorithm to be $(\varepsilon, \delta, N)$-correct after observing $N$ samples if it holds with a probability of at least $1 - \delta$:

$$\sup_{R \in \mathcal{R}_{(\mathcal{G}, \boldsymbol{\pi}^{\mathrm{Nash}})}} \inf_{\hat{R}^i \in \mathcal{R}_{(\hat{\mathcal{G}}, \hat{\boldsymbol{\pi}}^{\mathrm{Nash}})}} \max_{i \in [n]} \max_{\pi^i \in \pi^i} V_{\mathcal{G} \cup R}^i(\pi^i, \hat{\pi}^{-i}) - V_{\mathcal{G} \cup R}^i(\hat{\pi}^i, \hat{\pi}^{-i}) \leq \varepsilon$$

$$\sup_{\hat{R} \in \mathcal{R}_{(\mathcal{G}, \boldsymbol{\pi}^{\mathrm{Nash}})}} \inf_{R \in \mathcal{R}_{(\hat{\mathcal{G}}, \hat{\boldsymbol{\pi}}^{\mathrm{Nash}})}} \max_{i \in [n]} \max_{\pi^i \in \pi^i} V_{\mathcal{G} \cup R}^i(\pi^i, \hat{\pi}^{-i}) - V_{\mathcal{G} \cup R}^i(\hat{\pi}^i, \hat{\pi}^{-i}) \leq \varepsilon$$

We are now introducing the empirical estimator for the transition dynamic. For each iteration $k \in [K]$, let $n_k(s, \mathbf{a}, s') = \sum_{t=1}^{k} \mathbf{1}_{(s_t, \mathbf{a}_t, s_t') = (s, \mathbf{a}, s')}$ denote the count of visits to the triplet $(s, \mathbf{a}, s') \in \mathcal{S} \times \mathcal{A} \times \mathcal{S}$, and let $n_k(s, \mathbf{a}) = \sum_{s' \in \mathcal{S}} n_k(s, \mathbf{a}, s')$ denote the count of visits to the state-action pair $(s, \mathbf{a})$.

It is important to note the distinction here: the count of actions must be done separately for each agent, whereas the count of state visits needs to be done for any one of the agents. The cumulative count over all iterations $k \in [K]$ can then be written as:

$$N_k(s, \mathbf{a}, s') = \sum_{j \in [k]} n_j(s, \mathbf{a}, s'),$$

The cumulative state visit count are given by:

$$N_k^i(s, a^i) = \sum_{j \in [k]} n_j^i(s, a^i) \quad \forall i \in [n]$$

After introducing the empirical counts, we can now state the empirical estimators for the transition model:

$$\hat{P}_k(s' \mid s, \mathbf{a}) = \begin{cases} \frac{N_k(s, \mathbf{a}, s')}{N_k(s, \mathbf{a})} & \text{if} \quad N_k(s, \mathbf{a}) > 0 \\ \frac{1}{S} & \text{otherwise} \end{cases} \tag{8}$$

Next, we state the lemma that derives the good event by applying Hoeffding's inequality. As the Nash equilibrium is assumed to be a pure one, meaning it is deterministic for every $s \in \mathcal{S}$, we only have to define the good event for the transition probability and for the experts policy we require, that for each agent we have seen each state only once.

**Lemma C.8** (Good Event). *Let $k$ be the number of iterations and $\pi^{\mathrm{Nash}}$ be the stochastic expert policy. Furthermore let $\hat{\pi}^{\mathrm{Nash}}$ and $\hat{P}$ be the empirical estimates of the transition probability after $k$ iterations as defined in Eq. (8) respectively. Then for $\delta \in (0, 1)$, define the good event $\mathcal{E}$ as the event such that the following inequalities hold simultaneously for all $(s, \boldsymbol{a}) \in \mathcal{S} \times \mathcal{A}$ and $k \geq 1$:*

$$\sum_{s'} \left| (P(s' \mid s, \boldsymbol{a}) - \hat{P}_k(s' \mid s, \boldsymbol{a})) V^i(s') \right| \leq \frac{R_{\max}}{1 - \gamma} \sqrt{\frac{8 l_k(s, \boldsymbol{a})}{N_k^+(s, \boldsymbol{a})}},$$

$$\sum_{s'} \left| (P(s' \mid s, \boldsymbol{a}) - \hat{P}_k(s' \mid s, \boldsymbol{a})) \hat{V}^i(s') \right| \leq \frac{R_{\max}}{1 - \gamma} \sqrt{\frac{8 l_k(s, \boldsymbol{a})}{N_k^+(s, \boldsymbol{a})}}.$$

*where we introduced $l_k(s, \boldsymbol{a}) := \log\left( \frac{12 |\mathcal{S}| \prod_i |\mathcal{A}^i| (N_k^+(s, \boldsymbol{a}))^2}{\delta} \right).$*

*Proof.* We start with bound the two last equations. Therefore we define $l_k(s, \mathbf{a}) = \log\left( \frac{12 S \prod_i |\mathcal{A}^i| (N_k^+(s, \boldsymbol{a}))^2}{\delta} \right)$ and additionally we denote $\beta_{N_k(s, \mathbf{a})} = \frac{R_{\max}}{1 - \gamma} \sqrt{\frac{2 l_k(s, \mathbf{a})}{N_k^+(s, \mathbf{a})}}$. Now we define the set

$$\mathcal{E}^{\mathrm{trans}} := \left\{ \forall k \in \mathbb{N}, \forall (s, \mathbf{a}) \in \mathcal{S} \times \mathcal{A} : \sum_{s' \in \mathcal{S}} |(P(s' \mid s, \mathbf{a}) - \hat{P}(s' \mid s, \mathbf{a})) V^i(s')| \leq \beta_{N_k(s, \mathbf{a})} \right\}.$$

Then we get for $V^i$ with probability of $1 - \delta$:

$$\mathbb{P}\left((\mathcal{E}^{\text{trans}})^C\right)$$

$$= \mathbb{P}\left(\exists k \geq 1, \exists (s, \mathbf{a}) \in \mathcal{S} \times \mathcal{A} : \sum_{s'} \left|\left(P(s' \mid s, \mathbf{a}) - \hat{P}_k(s' \mid s, \mathbf{a})\right) V^i(s')\right| > \beta_{N_k(s, \mathbf{a})}(s, \mathbf{a})\right)$$

$$\overset{(a)}{\leq} \sum_{(s, \mathbf{a}) \in \mathcal{S} \times \mathcal{A}} \mathbb{P}\left(\exists k \geq 1 : \sum_{s'} \left|\left(P(s' \mid s, \mathbf{a}) - \hat{P}_k(s' \mid s, \mathbf{a})\right) V^i(s')\right| > \beta_{N_k(s, \mathbf{a})}(s, \mathbf{a})\right)$$

$$\overset{(b)}{\leq} \sum_{(s, \mathbf{a}) \in \mathcal{S} \times \mathcal{A}} \mathbb{P}\left(\exists m \geq 0 : \sum_{s'} \left|\left(P(s' \mid s, \mathbf{a}) - \hat{P}_k(s' \mid s, \mathbf{a})\right) V^i(s')\right| > \beta_{N_k(s, \mathbf{a})}(s, \mathbf{a})\right)$$

$$\overset{(c)}{\leq} \sum_{(s, \mathbf{a}) \in \mathcal{S} \times \mathcal{A}} \sum_{m \geq 0} 2\exp\left(\frac{\beta_{N_k(s, \mathbf{a})}^2 m^2 (1 - \gamma)^2}{4 m \gamma^2 R_{\max}^2}\right)$$

$$\leq \sum_{(s, \mathbf{a}) \in \mathcal{S} \times \mathcal{A}} \sum_{m \geq 0} \frac{\delta}{6 |\mathcal{S}| (\prod_{i=1}^n |\mathcal{A}^i|)(m^+)^2}$$

$$\leq \frac{\delta}{6}\left(1 + \frac{\pi^2}{6}\right) \leq \frac{\delta}{2},$$

where (a) uses a union bound over the state and joint-action space, (b) uses that we only consider the $m$-times, where we updated the estimated transition model and (c) uses an union bound over the update times $m$ and an application of Hoeffding's inequality combined with the fact that we can bound the value function, i.e. $V^i(s') \leq \frac{R_{\max}}{1 - \gamma}$ for every $s' \in \mathcal{S}$. $\qquad\square$

Following, we present the reward uncertainty metric, which allows us to demonstrate that the difference between the recovered reward function and the true reward function is bounded.

**Definition C.9** (Reward Uncertainty). Let $k$ be the number of iterations. Then the reward uncertainty after $k$ iterations for any $(s, \mathbf{a}) \in \mathcal{S} \times \mathcal{A}$ is defined as

$$C_k(s, \mathbf{a}) := \frac{4\gamma R_{\max}}{1 - \gamma}\left(\sqrt{\frac{8 l_k(s, \mathbf{a})}{N_k^+(s, \mathbf{a})}}\right).$$

**Theorem C.10.** *Let the reward uncertainty be defined as in C.9. Under the good event it holds for any $(s, \boldsymbol{a}) \in \mathcal{S} \times \mathcal{A}$ that:*

$$|R^i(s, \boldsymbol{a}) - \hat{R}^i(s, \boldsymbol{a})| \leq C_k(s, \boldsymbol{a}).$$

*Proof.* The theorem is an application of the error propagation Theorem C.5, followed by Lemma C.8

$$|R^i(s, \mathbf{a}) - \hat{R}^i(s, \mathbf{a})| \leq 2\left(A^i(s, \mathbf{a})|\mathbf{1}_E - \mathbf{1}_{\hat{E}}| + \gamma \sum_{s'} |(P(s' \mid s, \mathbf{a}) - \hat{P}(s' \mid s, \mathbf{a}))V^i(s')|\right)$$

$$\leq \frac{4\gamma R_{\max}}{1 - \gamma}\left(\sqrt{\frac{2 l_k(s, \mathbf{a})}{N_k^+(s, \mathbf{a})}}\right)$$

$$\leq \frac{4\gamma R_{\max}}{1 - \gamma}\left(\sqrt{\frac{8 l_k(s, \mathbf{a})}{N_k^+(s, \mathbf{a})}}\right)$$

$$= C_k(s, \mathbf{a}).$$

$\qquad\square$

**Corollary C.11.** *Let $k$ be the number of iterations for any allocation of the samples over the state-action space $\mathcal{S} \times \mathcal{A}$. Furthermore, let $\mathcal{R}_{(\mathcal{G}, \boldsymbol{\pi}^{\text{Nash}})}$ be the true feasible set and $\mathcal{R}_{(\hat{\mathcal{G}}, \hat{\boldsymbol{\pi}}^{\text{Nash}})}$ the recovered one. Then the optimality criterion 3.3 holds true, if*

$$\frac{1}{1 - \gamma} \max_{(s, \boldsymbol{a})} C_k(s, \boldsymbol{a}) \leq \frac{\varepsilon}{2}.$$

*Proof.* We complete the proof for the first case of the optimality criterion, the second one follows analogously.

$$\sup_{R \in \mathcal{R}_{(\mathcal{G}, \boldsymbol{\pi}^{\text{Nash}})}} \inf_{\hat{R}^i \in \mathcal{R}_{(\hat{\mathcal{G}}, \hat{\boldsymbol{\pi}}^{\text{Nash}})}} \max_{i \in [n]} \max_{\pi^i \in \Pi^i} \left( V_{\mathcal{G} \cup R}^i(\pi^i, \hat{\pi}^{-i}) - V_{\mathcal{G} \cup R}^i(\hat{\pi}^i, \hat{\pi}^{-i}) \right)$$

$$\overset{(a)}{\leq} \sup_{R \in \mathcal{R}_{(\mathcal{G}, \boldsymbol{\pi}^{\text{Nash}})}} \inf_{\hat{R}^i \in \mathcal{R}_{(\hat{\mathcal{G}}, \hat{\boldsymbol{\pi}}^{\text{Nash}})}} \max_{i, \pi^i} \left( \sum_{s, \mathbf{a}} \overline{w}_{s, \mathbf{a}}^{\hat{\pi}} \left( R^i(s, \mathbf{a}) - \hat{R}^i(s, \mathbf{a}) + \gamma \sum_{s'} (\hat{P} - P)(s' \mid s, \mathbf{a}) V^{i, \hat{\pi}}(s') \right) \right.$$

$$\left. + \sum_{s, \mathbf{a}} \overline{w}_{s, \mathbf{a}}^{\tilde{\pi}} \left( R^i(s, \mathbf{a}) - \hat{R}^i(s, \mathbf{a}) + \gamma \sum_{s'} (\hat{P} - P)(s' \mid s, \mathbf{a}) V^{i, \tilde{\pi}}(s') \right) \right)$$

$$\overset{(b)}{\leq} \sup_{R \in \mathcal{R}_{(\mathcal{G}, \boldsymbol{\pi}^{\text{Nash}})}} \inf_{\hat{R}^i \in \mathcal{R}_{(\hat{\mathcal{G}}, \hat{\boldsymbol{\pi}}^{\text{Nash}})}} \max_{i, \pi^i} 2 \sum_{s, \mathbf{a}} \overline{w}_{s, \mathbf{a}}^{\pi} \left( R^i(s, \mathbf{a}) - \hat{R}^i(s, \mathbf{a}) + \gamma \sum_{s'} (\hat{P} - P)(s' \mid s, \mathbf{a}) V^{i, \pi}(s') \right)$$

$$\overset{(c)}{\leq} \sup_{R \in \mathcal{R}_{(\mathcal{G}, \boldsymbol{\pi}^{\text{Nash}})}} \inf_{\hat{R}^i \in \mathcal{R}_{(\hat{\mathcal{G}}, \hat{\boldsymbol{\pi}}^{\text{Nash}})}} \max_{i, \pi^i} 2 \sum_{s, \mathbf{a}} \overline{w}_{s, \mathbf{a}}^{\pi} \left( A^i(s, \mathbf{a}) \left| \mathbf{1}_E - \mathbf{1}_{\hat{E}} \right| + 2\gamma \sum_{s'} \left| (\hat{P} - P)(s' \mid s, \mathbf{a}) V^i(s') \right| \right)$$

$$\overset{(d)}{\leq} \frac{2}{1 - \gamma} \max_{(s, \mathbf{a})} C_k(s, \mathbf{a}) \leq \varepsilon.$$

where in (a) we applied C.6; in (b) we used the fact that $a + b \leq 2 \max\{a, b\}$ and denoted the corresponding policy as $\boldsymbol{\pi}$; in (c) we used the error propagation Theorem C.5 and in (d) we used C.10. $\qquad \square$

We can combine the derived results to now state the main theorem regarding the sample complexity of allocating the samples uniformly over the state action space.

**Theorem C.12** (Sample Complexity of Uniform Sampling MAIRL). *Allocating the samples uniformly (see Algorithm 1) over the state and (joint-) action space stops with a probability of at least $1 - \delta$ after iteration $\tau$ and satisfies the optimality criterion (see 3.3), where the sample complexity is of order*

$$\tilde{\mathcal{O}} \left( \frac{\gamma^2 R_{\max}^2 |\mathcal{S}| \prod_{i=1}^n |\mathcal{A}^i|}{(1 - \gamma)^4 \varepsilon^2} \right)$$

*Proof.* We know from C.11, that we need

$$\frac{1}{1 - \gamma} \max_{(s, \mathbf{a})} C_k(s, \mathbf{a}) \leq \frac{\varepsilon_k}{2}$$

$$\Leftrightarrow \frac{2 R_{\max}}{(1 - \gamma)^2} \max_{(s, \mathbf{a})} \left( \gamma \sqrt{\frac{8 l_k(s, \mathbf{a})}{N_k^+(s, \mathbf{a})}} \right) \leq \frac{\varepsilon_k}{2}$$

This is satisfied if

$$\frac{4 \gamma R_{\max}}{(1 - \gamma)^2} \sqrt{\frac{8 l_k(s, \mathbf{a})}{N_k^+(s, \mathbf{a})}} \leq \frac{\varepsilon_k}{2}$$

To achieve the first condition, we get

$$N_k(s, \mathbf{a}) \geq \frac{R_{\max}}{(1 - \gamma)^4} \gamma^2 8 l_k(s, \mathbf{a}) \frac{8}{\varepsilon_k^2} = \frac{\gamma^2 64 R_{\max}}{(1 - \gamma)^4 \varepsilon_k^2} \log \left( \frac{12 S \prod_i |\mathcal{A}^i| (N_k^+(s, \mathbf{a}))^2}{\delta} \right)$$

Applying Lemma B.8 by Metelli et al. [2021] we get that

$$N_k(s, \mathbf{a}) \leq \frac{256 \gamma^2 R_{\max}^2}{(1 - \gamma)^4 \varepsilon_k^2} \log \left( \frac{128 \gamma^2 R_{\max}^2}{(1 - \gamma)^4 \varepsilon_k^2} \sqrt{\frac{12 |\mathcal{S}| \prod_i |\mathcal{A}^i|}{\delta}} \right).$$

At each iteration we are allocating the samples uniformly over $\mathcal{S} \times \mathcal{A}$ and recalling that $\tau_{s,a} = S \prod_i |\mathcal{A}^i| N_k(s, \mathbf{a})$ therefore we get

$$\tau \leq \frac{256|\mathcal{S}| \prod_i |\mathcal{A}^i| \gamma^2 R_{\max}^2}{(1-\gamma)^4 \varepsilon_k^2} \log \left( \frac{128\gamma^2 R_{\max}^2}{(1-\gamma)^4 \varepsilon_k^2} \sqrt{\frac{12|\mathcal{S}| \prod_i |\mathcal{A}^i|}{\delta}} \right)$$

Now we only have to achieve that we have seen each state at least once, to correctly estimate the policies for every agent. Therefore, we force that $N_k(s) \geq 1$. As we here need to allocate samples uniformly over the state space only but for every agent separately and recalling that $\tau_s = |\mathcal{S}| N_k(s)$, we get

$$\tau_s \leq n|S|$$

With $\tau = \tau_{s,a} + \tau_s$ we get in total

$$\tau \leq \frac{128|\mathcal{S}| \prod_i |\mathcal{A}^i| \gamma^2 R_{\max}^2}{(1-\gamma)^4 \varepsilon_k^2} \log \left( \frac{64a\gamma^2 R_{\max}^2}{(1-\gamma)^4 \varepsilon_k^2} \sqrt{\frac{12|\mathcal{S}| \prod_i |\mathcal{A}^i|}{\delta}} \right) + n|\mathcal{S}|.$$

This is exactly of order

$$\tilde{\mathcal{O}} \left( \frac{\gamma^2 R_{\max}^2 |\mathcal{S}| \prod_{i=1}^n |\mathcal{A}^i|}{(1-\gamma)^4 \varepsilon^2} \right)$$

$\square$

# D   Hardness result

In this section, we want to quantify the non-expressiveness of the recovered feasible reward set under a single NE observation. Therefore, we give the following hardness result. The idea is that the observed NE only covers a part of the environment and the definition of NE only ensures robustness against single agent deviations.

**Theorem D.1.** *Let us consider any IRL algorithm* $\mathrm{Alg}_{\mathrm{IRL}}$ *that chooses* $\hat{R} \in \mathcal{R}_{(\hat{\mathcal{G}}, \hat{\boldsymbol{\pi}}^{\mathrm{Nash}})}$ *that is not a constant reward, i.e.* $\hat{R} \neq C$ *for* $C \in [-R_{\max}, R_{\max}]$. *Furthermore consider a forward MARL algorithm* $\mathrm{Alg}_{\mathrm{MARL}}$ *that guarantees learning a policy* $\tilde{\boldsymbol{\pi}} \in \Pi^{\mathrm{Nash}}$. *Then, there exists a Markov game, such that even if* $\hat{\boldsymbol{\pi}} \in \Pi_{\mathrm{Nash}}$ *and* $\hat{R} \in \mathcal{R}_{(\mathcal{G}, \boldsymbol{\pi}^{\mathrm{Nash}})}$ *it holds true that* $\mathcal{E}(\boldsymbol{\pi}')$ *is of order* $(1-\gamma)^{-1}$.

*Proof.* We consider the following 2-player general-sum Markov game $\mathcal{G} \cup R$. Let $\mathcal{S} = \{s_0, s_1, s_2, s_3, s_4\}$ with $\mathcal{A}^1 = \mathcal{A}^2 = \{a_1, a_2\}$. Furthermore, let the transition dynamics be given by

$$P(\cdot \mid s_0, \mathbf{a}) = \begin{cases} s_1 & \text{if } \mathbf{a} = (a_1 a_1), \\ s_2 & \text{if } \mathbf{a} \in \{(a_1 a_2), (a_2 a_1)\}, \\ s_3 & \text{if } \mathbf{a} = (a_2, a_2), \end{cases}$$

In states $s_1, s_2, s_3$ the transition is defined to stay in the respective state with probability 1. Furthermore, let the true reward of the Markov game be given by

$$R^1(s_0, \mathbf{a}) = \begin{cases} 3 & \text{if } \mathbf{a} = (a_1 a_1), \\ 0 & \text{if } \mathbf{a} \in \{(a_1 a_2), (a_2 a_1)\}, \\ 2 & \text{if } \mathbf{a} = (a_2, a_2), \end{cases} \qquad R^2(s_0, \mathbf{a}) = \begin{cases} 2 & \text{if } \mathbf{a} = (a_1 a_1), \\ 0 & \text{if } \mathbf{a} \in \{(a_1 a_2), (a_2 a_1)\}, \\ 3 & \text{if } \mathbf{a} = (a_2, a_2), \end{cases}$$

For the other states we have that $R(s_1, \mathbf{a}) = R(s_3, \mathbf{a}) = 1 \, \forall \mathbf{a} \in \mathcal{A}$ and $R(s_2, \mathbf{a}) = 0 \, \forall \mathbf{a} \in \mathcal{A}$. This indicates that the Markov game has two pure NE strategies $\boldsymbol{\pi}^{\mathrm{Nash}}_1$ with $\pi_1(a_1 \mid s_0) = \pi_2(a_1 \mid s_0) = 1$ and $\boldsymbol{\pi}^{\mathrm{Nash}}_2$ with $\pi_1(a_2 \mid s_0) = \pi_2(a_2 \mid s_0) = 1$ and any distribution in states $s_1, s_2, s_3$. Note that this game can be seen as a Markov game extension of the Normal Form Game Battle of the Saxes that rewards the NE strategies in subsequent states. An illustration of this game can be found in Fig. 2. Let us assume that the observed Nash equilibrium is $\boldsymbol{\pi}^{\mathrm{Nash}}_1$. Next, we apply any $\mathrm{Alg}_{\mathrm{IRL}}$ that returns

$\hat{R} \in \mathcal{R}_{(\hat{\mathcal{G}}, \hat{\boldsymbol{\pi}}^{\text{Nash}})}$. Note, that the Nash equilibrium for the state $s_0$ and $s_1$ will be recovered perfectly and a potential $\hat{R}$ is given by

$$\hat{R}^1(s_0, \mathbf{a}) = \begin{cases} 2 & \text{if } \mathbf{a} = (a_1 a_1), \\ -1 & \text{if } \mathbf{a} = (a_1 a_2), \\ 2 & \text{if } \mathbf{a} = (a_2 a_1), \\ -2 & \text{if } \mathbf{a} = (a_2, a_2), \end{cases} \qquad \hat{R}^2(s_0, \mathbf{a}) = \begin{cases} 2 & \text{if } \mathbf{a} = (a_1 a_1), \\ 2 & \text{if } \mathbf{a} = (a_1 a_2), \\ -1 & \text{if } \mathbf{a} = (a_2 a_1), \\ -2 & \text{if } \mathbf{a} = (a_2, a_2), \end{cases}$$

Additionally, the rewards $\hat{R}(s_1, \mathbf{a}) = 1$, $\hat{R}(s_3, \mathbf{a}) = -1 \, \forall \mathbf{a} \in \mathcal{A}$ and $R(s_2, \mathbf{a}) = 1 \, \forall \mathbf{a} \in \mathcal{A}$. Then, we note that $\hat{\boldsymbol{\pi}}$ is indeed a NE under $\hat{R}$ and also in the true underlying environment $\mathcal{G} \cup R$.

However, in the recovered Markov game $\hat{\mathcal{G}} \cup \hat{R}$ there exists another pure equilibrium solutions $\boldsymbol{\pi}^{\text{Nash}}{}_3, \boldsymbol{\pi}^{\text{Nash}}{}_4 \notin \Pi^{\text{Nash}}_{\mathcal{G} \cup R}$, where for $\boldsymbol{\pi}^{\text{Nash}}{}_3$ we have $\pi_3^1(a_1 \mid s_0) = 1$ and $\pi_3^2(a_2 \mid s_0) = 1$ and $\boldsymbol{\pi}^{\text{Nash}}{}_4$ is given by $\pi_3^1(a_2 \mid s_0) = 1$ and $\pi_3^2(a_1 \mid s_0) = 1$.

If one now applies a forward MARL algorithm that guarantees convergence to a any (pure) NE defined by $\tilde{\boldsymbol{\pi}}$, i.e. satisfying

$$\mathcal{E}_{\hat{R}} = \max_{i \in \{1,2\}} \max_{\pi^i \in \Pi^i} V_{\hat{R}}^i(\pi^i, \tilde{\pi}^{-i}) - V_{\hat{R}}^i(\tilde{\boldsymbol{\pi}}) = 0.$$

Then, assuming that $\tilde{\boldsymbol{\pi}}$ in the true Markov game it holds true that

$$\mathbb{E}_{\text{Alg}_{\text{MARL}}}[\mathcal{E}_R(\tilde{\boldsymbol{\pi}})] \geq \frac{1}{(1 - \gamma)} \mathbb{P}(\tilde{\boldsymbol{\pi}} \neq \boldsymbol{\pi}^{\text{Nash}}{}_1).$$

This holds true because for $\tilde{\boldsymbol{\pi}} = \boldsymbol{\pi}^{\text{Nash}}{}_1$, we have that $\mathcal{E}(\boldsymbol{\pi}^{\text{Nash}}{}_1) = 0$ as this is also a NE in the original Markov game. However for $\boldsymbol{\pi}^{\text{Nash}}{}_3$ and $\boldsymbol{\pi}^{\text{Nash}}{}_4$ which both go to state $s_2$ a Best response would either be to go $s_1$ or $s_3$ resulting in a exploitability for 1 for all future states. Assuming that the algorithm returns a NE uniformly across the set of NE, we get

$$\mathbb{E}_{\text{Alg}_{\text{MARL}}}[\mathcal{E}_R(\tilde{\boldsymbol{\pi}})] \geq \frac{1}{(1 - \gamma)} \mathbb{P}(\tilde{\boldsymbol{\pi}} \neq \boldsymbol{\pi}^{\text{Nash}}{}_1) = \frac{2}{3(1 - \gamma)}.$$

This is exactly of the order $(1 - \gamma)^{-1}$ and completes the proof. $\qquad \square$

Next, we want to provide further intuition on this phenomenon by giving the Normal Form Game that is the origin of the considered Markov game instance.

*Example* D.1. We consider the general form of a coordination game as a Normal Form Game (NFG):

|  | Player 2: Stag | Player 2: Hare |
|---|---|---|
| Player 1: Stag | $(A, A)$ | $(C, B)$ |
| Player 1: Hare | $(B, C)$ | $(D, D)$ |

In general coordination games, we have that $D > B$ and $D - B < A + D - B - C$. Assume we observe the pure Nash equilibrium strategy (Stag, Stag). The feasible reward set $\mathcal{R}$ then contains all rewards that satisfy:

$$R^1(\text{Stag}, \text{Stag}) \geq R^1(\text{Hare}, \text{Stag}) \wedge R^2(\text{Stag}, \text{Stag}) \geq R^2(\text{Stag}, \text{Hare}),$$

while for all other reward values, **any** rewards are feasible, i.e., $R^1(\text{Stag}, \text{Hare}), R^1(\text{Hare}, \text{Hare}) \in \mathbb{R}^{\mathcal{A}^1 \times \mathcal{A}^2}$.

This flexibility in reward specification allows for undesirable scenarios (see Example D.1), such as:

- **Changing the nature of the game**: The game can transform into an anti-coordination variant with additional pure Nash equilibria not present in the original game.
- **Losing equilibria**: Rewards can be defined so that "Stag" becomes the unique dominant strategy for player 1.

The following are two examples of feasible rewards if observing the NE expert (Stag, Stag):

|  | Player 2: Stag | Player 2: Hare |
|---|---|---|
| Player 1: Stag | $(2,2)$ | $(0,0)$ |
| Player 1: Hare | $(0,0)$ | $(-1,-1)$ |

|  | Player 2: Stag | Player 2: Hare |
|---|---|---|
| Player 1: Stag | $(2,2)$ | $(-1,2)$ |
| Player 1: Hare | $(2,-1)$ | $(-10,-10)$ |

This example highlights that even in simple NFGs, the feasible reward set encompasses too many reward functions, including those that significantly alter the game's equilibria. This contrasts with the single-agent IRL setting, where the feasible reward set contains degenerate rewards like constant ones, but due to the fact that all equilibria obtain the same value, preserving the meaning of the environment. In the multi-agent setting, this second source of ambiguity allows for strategic behavior entirely absent in the original game, which is highly undesirable if the goal is to recover meaningful reward functions for transfer to new environments.

# E  Proofs of Section 3.2

In this section, we will give the missing proofs of Section 3.2. We start by giving again the definition of a feasible reward function for an observed pair of expert policies. In particular, if the observed expert policy is a QRE equilibrium.

**Definition E.1** (Feasible Reward Set (regularized))**.** A reward function $R$ is feasible for an MAIRL problem $(\mathcal{G}, (\mu^*, \nu*))$ if and only if the observed policy pair forms an equilibrium in $\mathcal{G} \cup R$.

Additionally, we will restate Definition 3.3 in terms of regularized games.

**Definition E.2** (Regularized Optimality Criterion)**.** Let $\mathcal{R} := \mathcal{R}_{(\mathcal{G},(\mu^*,\nu^*))}$ be the exact feasible set and $\hat{\mathcal{R}} := \mathcal{R}_{(\hat{\mathcal{G}},(\hat{\mu}^*,\hat{\nu}^*))}$ the recovered feasible set after observing $N \geq 0$ samples from the underlying MAIRL problem $(\mathcal{G}, \boldsymbol{\pi}^{\mathrm{Nash}})$. We consider an algorithm to be $(\varepsilon, \delta, N)$-correct after observing $N$ samples if with a probability of at least $1 - \delta$ it holds:

$$\sup_{R \in \mathcal{R}} \inf_{\hat{R} \in \hat{\mathcal{R}}} \max\{\max_{\mu_\lambda} V_\lambda^1(\mu, \hat{\nu}) - V_\lambda^1(\hat{\mu}, \hat{\nu}), \max_{\nu_\lambda} V_\lambda^2(\hat{\mu}, \nu) - V_\lambda^2(\hat{\mu}, \hat{\nu})\} \leq \varepsilon$$

$$\sup_{\hat{R} \in \hat{\mathcal{R}}} \inf_{R \in \mathcal{R}} \max\{\max_{\mu_\lambda} V_\lambda^1(\mu, \hat{\nu}) - V_\lambda^1(\hat{\mu}, \hat{\nu}), V_\lambda^2(\hat{\mu}, \hat{\nu}) - \min_{\nu_\lambda} V_\lambda^2(\hat{\mu}, \nu)\} \leq \varepsilon,$$

where we used $\mu_\lambda, \nu_\lambda$ to denote entropy regularized policies.

This optimality criterion can be seen as the *soft* version of the Nash Imitation Gap(Definition 3.2).

Next, note that a policy is considered optimal, i.e. a QRE equilibrium, if the policies satisfy

$$\mu^*(a \mid s) = \frac{\exp\left(\frac{1}{\lambda} \sum_{b' \in \mathcal{B}} \nu^*(b' \mid s) Q_\lambda^{*,1}(s, a, b')\right)}{\sum_{a' \in \mathcal{A}} \exp\left(\frac{1}{\lambda} \sum_{b' \in \mathcal{B}} \nu^*(b' \mid s) Q_\lambda^{*,1}(s, a', b')\right)}, \tag{9}$$

$$\nu^*(b \mid s) = \frac{\exp\left(\frac{1}{\lambda} \sum_{a' \in \mathcal{A}} \mu^*(a' \mid s) Q_\lambda^{*,2}(s, a', b)\right)}{\sum_{b' \in \mathcal{A}} \exp\left(\frac{1}{\lambda} \sum_{a' \in \mathcal{A}} \mu^*(a' \mid s) Q_\lambda^{*,2}(s, a', b')\right)}. \tag{10}$$

Similarly to the analysis done in the single-agent setting the goal is to derive an explicit characterization of the reward function [Metelli et al., 2023, Lindner et al., 2022, Metelli et al., 2023, Zhao et al., 2024, Cao et al., 2021]. The idea is to rewrite the formulation of the optimal policy in terms of the reward function by using the definition of the value function and the $Q$-function. We will present the analysis only for player 1, it holds analogously for player 2. For a better readability we drop the superscript for the player.

**Lemma E.3** (Feasible Explicit (regularized))**.** *A reward function $R$ for the regularized Markov game i feasible if and only if there exists $V \in \mathbb{R}^{\mathcal{S}}$ and $|\mathcal{B}| - 1$ many functions $R' \in [-R_{\max}, R_{\max}]^{\mathcal{S} \times \mathcal{A} \times \mathcal{B}}$ such that for all $(s, a, b)$ it holds that*

$$R(s,a,b) = \frac{1}{\nu^*(b|s)} \left( \lambda \log(\mu^*(a \mid s)) + V(s) - \gamma \sum_{s'} \sum_{b' \in \mathcal{B}} \nu^*(b' \mid s) P(s' \mid s, a, b') V(s') - \sum_{b' \neq b} \nu^*(b' \mid s) R(s, a, b') \right).$$

*Proof.* First, assume that the reward function $R$ is feasible. By definition, this implies that $\mu^*$ is an optimal policy for agent 1 under $R$ when agent 2 plays $\nu^*$. Let $V_\lambda^*(s)$ be the corresponding unique entropy-regularized optimal value function for agent 1. The optimal policy $\mu^*(a \mid s)$ (see Eq. (9)) is given by

$$\mu^*(a \mid s) = \frac{\exp\left(\frac{1}{\lambda} \sum_{b' \in \mathcal{B}} \nu^*(b' \mid s) Q_\lambda^*(s, a, b')\right)}{\sum_{a' \in \mathcal{A}} \exp\left(\frac{1}{\lambda} \sum_{b' \in \mathcal{B}} \nu^*(b' \mid s) Q_\lambda^*(s, a', b')\right)}.$$

Recognizing that the denominator relates to the soft value function $V_\lambda^*(s) = \lambda \log \sum_{a \in \mathcal{A}} \exp\left(\frac{1}{\lambda} \sum_{b' \in \mathcal{B}} \nu^*(b' \mid s) Q_\lambda^*(s, a, b')\right)$, we can write

$$\mu^*(a \mid s) = \exp\left(\frac{1}{\lambda}\left(\sum_{b' \in \mathcal{B}} \nu^*(b' \mid s) Q_\lambda^*(s, a, b') - V_\lambda^*(s)\right)\right).$$

Using the definition of the Q-function, $Q_\lambda^*(s, a, b') = R(s, a, b') + \gamma \sum_{s' \in \mathcal{S}} P(s' \mid s, a, b') V_\lambda^*(s')$, we substitute this into the expression for $\mu^*(a \mid s)$:

$$\mu^*(a \mid s)$$
$$= \exp\left(\frac{1}{\lambda}\left(\sum_{b' \in \mathcal{B}} \nu^*(b' \mid s) R(s, a, b') + \gamma \sum_{b' \in \mathcal{B}} \nu^*(b' \mid s) \sum_{s' \in \mathcal{S}} P(s' \mid s, a, b') V_\lambda^*(s') - V_\lambda^*(s)\right)\right).$$

Taking the logarithm and rearranging terms yields

$$\sum_{b' \in \mathcal{B}} \nu^*(b' \mid s) R(s, a, b') = \lambda \log(\mu^*(a \mid s)) + V_\lambda^*(s) - \gamma \sum_{b' \in \mathcal{B}} \nu^*(b' \mid s) \sum_{s' \in \mathcal{S}} P(s' \mid s, a, b') V_\lambda^*(s').$$

Let $K_{V_\lambda^*}(s, a)$ denote the right-hand side of the equation above. Then, for any specific action $b \in \mathcal{B}$ such that $\nu^*(b \mid s) > 0$, we can express $R(s, a, b)$ as

$$R(s, a, b) = \frac{1}{\nu^*(b \mid s)}\left(K_{V_\lambda^*}(s, a) - \sum_{b' \in \mathcal{B} \setminus \{b\}} \nu^*(b' \mid s) R(s, a, b')\right).$$

This matches the form specified in the lemma, where $V(s)$ is taken as $V_\lambda^*(s)$, and the $|\mathcal{B}| - 1$ functions $R'$ correspond to the components $R(s, a, b')$ for $b' \neq b$ from the original feasible reward $R$.

For the opposing direction, assume there exists an arbitrary function $V \in \mathbb{R}^{\mathcal{S}}$ and a reward function $R$ (composed of $|\mathcal{B}| - 1$ given functions $R'(s, a, b')$ for $b' \neq b$ that are within $[-R_{\max}, R_{\max}]$, and the remaining component $R(s, a, b)$ defined by the formula) such that for all $(s, a, b)$ it holds that

$$R(s, a, b) = \frac{1}{\nu^*(b \mid s)}\left(\lambda \log(\mu^*(a \mid s)) + V(s) - \gamma \sum_{s' \in \mathcal{S}} \sum_{b'' \in \mathcal{B}} \nu^*(b'' \mid s) P(s' \mid s, a, b'') V(s')\right.$$
$$\left. - \sum_{b' \in \mathcal{B} \setminus \{b\}} \nu^*(b' \mid s) R(s, a, b')\right).$$

This structural definition implies that the expected reward for agent 1, $R^{\nu^*}(s, a) = \sum_{b' \in \mathcal{B}} \nu^*(b' \mid s) R(s, a, b')$, satisfies

$$R^{\nu^*}(s, a) = \lambda \log(\mu^*(a \mid s)) + V(s) - \gamma \sum_{s' \in \mathcal{S}} \sum_{b' \in \mathcal{B}} \nu^*(b' \mid s) P(s' \mid s, a, b') V(s'). \quad (11)$$

Let $P^{\nu^*}(s' \mid s, a) = \sum_{b' \in \mathcal{B}} \nu^*(b' \mid s) P(s' \mid s, a, b')$ be the expected transition probability from agent 1's perspective. Then (11) becomes $R^{\nu^*}(s, a) = \lambda \log(\mu^*(a \mid s)) + V(s) - \gamma \sum_{s' \in \mathcal{S}} P^{\nu^*}(s' \mid s, a) V(s')$. We now show that $R$ is feasible by demonstrating that $V(s)$ is the value function of policy $\mu^*$ for agent 1 (given agent 2 plays $\nu^*$) and that $\mu^*$ is the optimal policy.

First, let $V^{\mu^*}(s)$ be the value function for agent 1 when it follows policy $\mu^*$ and agent 2 follows $\nu^*$, with rewards $R(s, a, b)$. The Bellman equation for $V^{\mu^*}(s)$ is given by

$$V^{\mu^*}(s) = \sum_{a \in \mathcal{A}} \mu^*(a|s)\left(R^{\nu^*}(s, a) + \gamma \sum_{s' \in \mathcal{S}} P^{\nu^*}(s'|s, a) V^{\mu^*}(s') - \lambda \log \mu^*(a|s)\right).$$

Substituting the expression for $R^{\nu^*}(s,a)$ from (11):

$$V^{\mu^*}(s) = \sum_{a \in \mathcal{A}} \mu^*(a|s) \left( \left[ \lambda \log(\mu^*(a \mid s)) + V(s) - \gamma \sum_{s' \in \mathcal{S}} P^{\nu^*}(s' \mid s, a) V(s') \right] \right.$$

$$\left. + \gamma \sum_{s' \in \mathcal{S}} P^{\nu^*}(s'|s, a) V^{\mu^*}(s') - \lambda \log \mu^*(a|s) \right)$$

$$= \sum_{a \in \mathcal{A}} \mu^*(a|s) \left( V(s) - \gamma \sum_{s' \in \mathcal{S}} P^{\nu^*}(s' \mid s, a) V(s') + \gamma \sum_{s' \in \mathcal{S}} P^{\nu^*}(s'|s, a) V^{\mu^*}(s') \right).$$

Since $\sum_{a \in \mathcal{A}} \mu^*(a|s) = 1$, we have

$$V^{\mu^*}(s) = V(s) + \gamma \sum_{a \in \mathcal{A}} \mu^*(a|s) \sum_{s' \in \mathcal{S}} P^{\nu^*}(s' \mid s, a)(V^{\mu^*}(s') - V(s')).$$

Let $g(s) = V^{\mu^*}(s) - V(s)$. Then $g(s) = \gamma \sum_{a \in \mathcal{A}} \mu^*(a|s) \sum_{s' \in \mathcal{S}} P^{\nu^*}(s' \mid s, a) g(s')$. This equation, $g = \gamma \mathcal{P}_{\mu^*, \nu^*} g$, where $\mathcal{P}_{\mu^*, \nu^*}$ is the Bellman operator for policy evaluation, implies that $g(s) = 0$ is the unique solution since $\gamma \in [0, 1)$ ensures $\mathcal{P}_{\mu^*, \nu^*}$ is a contraction. Thus, $V^{\mu^*}(s) = V(s)$ for all $s \in \mathcal{S}$.

Next, we show that $\mu^*(a|s)$ is the entropy-regularized optimal policy for agent 1. The optimal policy, $\pi^{*,1}(a|s)$, is given by $\pi^{*,1}(a|s) \propto \exp\left( \frac{1}{\lambda} E_Q^{*,1}(s, a) \right)$, where $E_Q^{*,1}(s, a)$ is the expected Q-value using the optimal value function $V(s)$:

$$E_Q^{*,1}(s, a) = R^{\nu^*}(s, a) + \gamma \sum_{s' \in \mathcal{S}} P^{\nu^*}(s'|s, a) V(s').$$

Substituting the expression for $R^{\nu^*}(s,a)$ from (11):

$$E_Q^{*,1}(s, a) = \left[ \lambda \log(\mu^*(a \mid s)) + V(s) - \gamma \sum_{s' \in \mathcal{S}} P^{\nu^*}(s' \mid s, a) V(s') \right] + \gamma \sum_{s' \in \mathcal{S}} P^{\nu^*}(s'|s, a) V(s').$$

Rewriting this expression gives exactly the form of an optimal policy in the entropy regularized Markov game. Since $\mu^*(a|s)$ is the optimal policy for agent 1 under the reward $R$ (when agent 2 plays $\nu^*$), the reward function $R$ is feasible. This completes the proof. $\square$

In the following, we want to investigate how an error in estimating the expert policy pair and the transition function translates to the recovered reward function. Note that compared to the single-agent case it is required to estimate the policy of opponent accurately as well.

The next lemma is of great importance, to derive the error propagation. It states how estimating the induced transition probability is related to the joint transition probability.

**Lemma E.4.** *Let $V : \mathcal{S} \to \mathbb{R}$ be a function. Furthermore, let $P^\nu$ be the induced transition probability and $\widehat{P}^\nu$ the estimated one of an underlying Markov game with transition dynamic $P$. With out loss of generality, we assume that we are fixing the policy of agent 2, i.e. $\nu$. Then it holds true, that*

$$| \sum_{s'} P^\nu(s' \mid s, a) V(s') - \sum_{s'} \widehat{P}^\nu(s' \mid s, a) V(s')|$$

$$\leq \max_b | \sum_{s'} V(s') \left( P(s' \mid s, a, b) - \hat{P}(s' \mid s, a, b) \right) |$$

*Proof.*

$$|\sum_{s'} P^\nu(s' \mid s, a)V(s) - \sum_{s'} \widehat{P}^\nu(s' \mid s, a)V(s')|$$

$$= |\sum_{s'} \sum_{b \in \mathcal{B}} \nu(b \mid s)P(s' \mid s, a, b)V(s') - \sum_{s'} \sum_{b \in \mathcal{B}} \hat{\nu}(b \mid s)\widehat{P}(s' \mid s, a, b)V(s')|$$

$$\leq \sum_{b \in \mathcal{B}} \max(\nu(b \mid s), \hat{\nu}(b \mid s))|\sum_{s'} P(s' \mid s, a, b)V(s') - \sum_{s'} \widehat{P}(s' \mid s, a, b)V(s')|$$

$$\leq \max_b |\sum_{s'} P(s' \mid s, a, b)V(s') - \sum_{s'} \widehat{P}(s' \mid s, a, b)V(s')|$$

$$= \max_b |\sum_{s'} V(s') \left( P(s' \mid s, a, b) - \widehat{P}(s' \mid s, a, b) \right)|$$

$$\square$$

With the introduced Lemma, we can now introduce an error propagation theorem. The idea is that we use the explicit reward function from Lemma E.3 and bound the individual terms of the true underlying MAIRL problem and the estimated one.

**Theorem E.5** (Error propagation)**.** *Let the MAIRL problem be given by $(\mathcal{G}, (\mu^*, \nu^*))$ for a Markov game and let $(\hat{\mathcal{G}}, (\hat{\mu}^*, \hat{\nu}^*)$ be another MAIRL problem. Then, we have that*

$$|R(s, a, b) - \hat{R}(s, a, b)| \leq \frac{2}{\nu^*(b \mid s)\hat{\nu}^*(b \mid s)} \left( \lambda| \log \mu^*(a \mid s) - \log \hat{\mu}^*(a \mid s)| \right.$$

$$\left. + \gamma \max_b |\sum_{s'} V(s')P(s' \mid s, a, b) - \hat{P}(s' \mid s, a, b)| + R_{\max}\mathrm{TV}(\nu, \hat{\nu}) \right)$$

*Proof.* In the first step, we use the derived explicit form of the reward derived in Lemma E.3.

$$\hat{R}(s, a, b) = \frac{1}{\hat{\nu}^*}(b \mid s)\left( \lambda \log(\hat{\mu}^*(a \mid s)) + \hat{V}(s) \right.$$

$$\left. - \gamma \sum_s \sum_{b'} \hat{\nu}^*(b' \mid s)\hat{P}(s' \mid s, a, b')\hat{V}(s') - \sum_{b'} \nu^*(b' \mid s)\hat{R}(s, a, b') \right)$$

As pointed out in Metelli et al. [2023], the rewards $\hat{R}(s, a, b)$ do not have to be bounded by the same $R_{\max}$ as $R(s, a, b)$. To fix this issue the authors point out, that the reward needs to be rescaled such that the recovered feasible reward set is bounded by the same value. In our case we have to be a bit more careful with the choice of the scaling, as we did not assume that the reward is bounded by 1. As we proof the existence of such reward function, we can choose $\tilde{V}(s) = V(s)$ for every $s \in \mathcal{S}$ and $\tilde{R}(s, a, b') = R(s, a, b')\forall b' \neq b$ for every $(s, a, b) \in \mathcal{S} \times \mathcal{A} \times \mathcal{B}$, which results in rewards

$$\tilde{R}(s, a, b) =$$

$$\frac{\lambda \log(\hat{\mu}^*(a \mid s)) + V(s) - \gamma \sum_s \sum_{b'} \hat{\nu}^*(b \mid s)\hat{P}(s' \mid s, a, b')V(s') - \sum_{b' \neq b} \hat{\nu}^*(b' \mid s)R(s, a, b')}{\hat{\nu}^*(b \mid s)}.$$

Now we need to rescale the reward with $R_{\max} + |\varepsilon^i(s, a, b)|$ respectively,

$$\varepsilon(s, a, b)$$

$$= \frac{\lambda \log(a \mid s) + V(s) - \gamma \sum_{s'} \sum_{b' \in \mathcal{B}} \nu^*(b' \mid s)P(s' \mid s, a, b')V(s') - \sum_{b' \neq b} \nu^*(b' \mid s)R(s, a, b')}{\nu^*(b \mid s)}$$

$$- \frac{\left( \lambda \log(\hat{\mu}^*(a \mid s)) + \hat{V}(s) - \gamma \sum_s \sum_{b'} \hat{\nu}^*(b \mid s)\hat{P}(s' \mid s, a, b')\hat{V}(s') - \sum_{b'} \hat{\nu}^*(b' \mid s)\hat{R}(s, a, b') \right)}{\hat{\nu}^*(b \mid s)},$$

such that it remains bounded by $R_{\max}$, we receive $\hat{R}(s, a, b) = \tilde{R}(s, a, b)\frac{R_{\max}}{R_{\max} + |\varepsilon(s, a, b)|}$.

It then follows that:

$$
\begin{aligned}
|R(s,a,b) - \hat{R}(s,a,b)| &= |R(s,a,b) - \frac{R_{\max}\tilde{R}(s,a,b)}{R_{\max} + |\varepsilon(s,a,b)|}| \\
&\leq \frac{R_{\max}}{R_{\max} + |\varepsilon(s,a,b)|} \left| \left( \frac{R_{\max} + |\varepsilon(s,a,b)|}{R_{\max}} \right) R(s,a,b) - \tilde{R}(s,a,b) \right| \\
&\leq \frac{R_{\max}}{R_{\max} + |\varepsilon(s,a,b)|} \left( |R(s,a,b) - \tilde{R}(s,a,b)| + |\frac{\varepsilon(s,a,b)}{R_{\max}} R(s,a,b)| \right) \\
&\leq \frac{R_{\max}}{R_{\max} + |\varepsilon(s,a,b)|} \left( |\varepsilon(s,a,b)| + |\varepsilon(s,a,b)| \right) \\
&\leq \frac{R_{\max}}{R_{\max}} \left( |\varepsilon(s,a,b)| + |\varepsilon(s,a,b)| \right) \\
&= 2|\varepsilon(s,a,b)|
\end{aligned}
$$

In the next step, we bound the $|\varepsilon(s,a,b)|$

$$
\begin{aligned}
&|\varepsilon(s,a,b)| \\
&= \frac{\lambda \log(a \mid s) + V(s) - \gamma \sum_{s'} \sum_{b' \in \mathcal{B}} \nu^*(b' \mid s) P(s' \mid s,a,b') V(s') - \sum_{b' \neq b} \nu^*(b' \mid s) R(s,a,b')}{\nu^*(b \mid s)} \\
&\quad - \frac{\left( \lambda \log(\hat{\mu}^*(a \mid s)) + \hat{V}(s) - \gamma \sum_{s} \sum_{b'} \hat{\nu}^*(b' \mid s) \hat{P}(s' \mid s,a,b') \hat{V}(s') - \sum_{b'} \hat{\nu}^*(b' \mid s) \hat{R}(s,a,b') \right)}{\hat{\nu}^*(b \mid s)} \\
&\leq \frac{1}{\nu^*(b \mid s)\hat{\nu}^*(b \mid s)} \left( \lambda |\log \mu^*(a \mid s) - \log \hat{\mu}^*(a \mid s)| \right. \\
&\quad + \gamma \left| \sum_{s'} \sum_{b'} (\nu(b' \mid s) P(s' \mid s,a,b') - \hat{\nu}^*(b' \mid s) \hat{P}(s,a,b')) V(s') \right| \\
&\quad + \left. \left| \sum_{b' \neq b} R(s,a,b')(\nu^*(b' \mid s) - \hat{\nu}^*(b \mid s)) \right| \right) \\
&\overset{(i)}{\leq} \frac{1}{\nu^*(b \mid s)\hat{\nu}^*(b \mid s)} \left( \lambda |\log \mu^*(a \mid s) - \log \hat{\mu}^*(a \mid s)| \right. \\
&\quad + \gamma \left| \max_{b' \in \mathcal{B}} |\sum_{s'} P(s' \mid s,a,b') - \hat{P}(s,a,b')) V(s')| + \left| \sum_{b' \neq b} R(s,a,b')(\nu^*(b' \mid s) - \hat{\nu}^*(b \mid s)) \right| \right| \right) \\
&\overset{(ii)}{\leq} \frac{1}{\nu^*(b \mid s)\hat{\nu}^*(b \mid s)} \left( \lambda |\log \mu^*(a \mid s) - \log \hat{\mu}^*(a \mid s)| \right. \\
&\quad + \gamma \left| \max_{b' \in \mathcal{B}} \sum_{s'} P(s' \mid s,a,b') - \hat{P}(s,a,b')) V(s') \right| + R_{\max} \sum_{b' \neq b} |(\nu^*(b' \mid s) - \hat{\nu}^*(b \mid s))| \right) \\
&\leq \frac{1}{\nu^*(b \mid s)\hat{\nu}^*(b \mid s)} \left( \lambda |\log \mu^*(a \mid s) - \log \hat{\mu}^*(a \mid s)| \right. \\
&\quad + \gamma \left| \max_{b' \in \mathcal{B}} |\sum_{s'} P(s' \mid s,a,b') - \hat{P}(s,a,b')) V(s')| + R_{\max} \sum_{b' \neq b} \mathrm{TV}(\nu^*, \hat{\nu}^*) \right),
\end{aligned}
$$

where we used Lemma E.4 for $(i)$, then the assumption that $R(s,a,b)$ is bounded by $R_{\max}$ in $(ii)$ and last, we added $|\nu^*(b \mid s) - \hat{\nu}^*(b \mid s)|$ to obtain the definition of the total variation with the triangle inequality. $\square$

We now again, want to use the empirical estimators to do a sample complexity analysis. The part of the transition probability can be obtained similar to the case of the pure NE, while for the policy we

need to bound with high probability

$$|\log(\mu^*(a \mid s)) - \log(\hat{\mu}^*(a \mid s))|, \ |\log(\nu^*(a \mid s)) - \log(\hat{\nu}^*(a \mid s))|.$$

Let us first introduce the assumption, also common in single-agent IRL, that the lowest probability of an action taken from the expert is bounded away from zero by some constant.

**Assumption E.6.** Let $\mu^*, \nu^*$ be the QRE equilibrium expert policies. Then we assume that

$$\min_{a \in \mathcal{A}, b \in \mathcal{B}} (\mu^*(a \mid s), \nu^*(b \mid s)) \geq \Delta_{\min}.$$

Now we are introducing the empirical estimators, used for recovering the MAIRL problem.

For both estimation tasks, the expert policies and the transition probability, we employ empirical estimators. For each iteration $k \in [K]$, let $n_k(s, a, b, s') = \sum_{t=1}^{k} \mathbf{1}_{(s_t, a_t, b_t, s'_t)=(s,a,b,s')}$ denote the count of visits to the triplet $(s, a, b, s') \in \mathcal{S} \times (\mathcal{A} \times \mathcal{B}) \times \mathcal{S}$, and let $n_k(s, a, b) = \sum_{s' \in \mathcal{S}} n_k(s, a, b, s')$ denote the count of visits to the state-action pair $(s, a)$. Additionally, we introduce $n_k(s, a) = \sum_{t=1}^{k} \mathbf{1}_{(s_t, a_t)=(s,a)}$ and $n_k(s, b) = \sum_{t=1}^{k} \mathbf{1}_{(s_t, b_t)=(s,b)}$ as the count of times action $a$ and respectively $b$ was sampled in state $s \in \mathcal{S}$ for each agent $i$, and $n_k(s) = \sum_{a \in \mathcal{A}} n_k(s, a)$ as the count of visits to state $s$ for any agent.

It is important to note the distinction here: the count of actions must be done separately for each agent, whereas the count of state visits needs to be done for both of the agents.

The cumulative count of actions for each agent and the cumulative state visit count are given by:

$$N_k(s, a, b) = \sum_{j \in [k]} n_j(s, a, b) \quad N(s) = \sum_{j \in [k]} n_j(s).$$

After introducing the empirical counts, we can now state the empirical estimators for the transition model and the expert's policy:

$$\hat{P}_k(s' \mid s, a, b) = \begin{cases} \frac{N_k(s,a,b,s')}{N_k(s,a,b)} & \text{if} \quad N_k(s, a, b) > 0 \\ \frac{1}{S} & \text{otherwise} \end{cases} \tag{12}$$

$$\hat{\mu}_k(a \mid s) = \begin{cases} \frac{N_k(s,a)}{N_k(s)} & \text{if} \quad N_k(s) > 0 \\ \frac{1}{|\mathcal{A}|} & \text{otherwise.} \end{cases} \tag{13}$$

$$\hat{\nu}_k(b \mid s) = \begin{cases} \frac{N_k(s,b)}{N_k(s)} & \text{if} \quad N_k(s) > 0 \\ \frac{1}{|\mathcal{B}|} & \text{otherwise.} \end{cases} \tag{14}$$

Next, we state the lemma that derives the good event. Note that here we prove something stronger regarding the transition model, i.e. that the good event holds for all $s, a, b$, therefore also for $\max_{b \in \mathcal{B}}$.

**Lemma E.7** (Good Event Regularized Games). *Let $k$ be the number of iterations and $(\mu^*, \nu^*)$ be the QRE expert policies. Furthermore let $(\hat{\mu}, \hat{\nu})$ and $\hat{P}$ be the empirical estimates of the Nash expert and the transition probability after $k$ iterations as defined in Eq. (13) and Eq. (12) respectively. Then for $\delta \in (0, 1)$, define the good event $\mathcal{E}$ as the event such that the following inequalities hold*

*simultaneously for all $(s, a, b) \in \mathcal{S} \times \mathcal{A} \times \mathcal{B}$ and $k \geq 1$, which holds with probability at least $1 - \delta$:*

$$|\log(\mu(a \mid s)) - \log(\hat{\mu}(a \mid s))| \leq \frac{1}{\Delta_{\min}} \sqrt{\frac{2 \log(10|\mathcal{S}||\mathcal{A}|(N_k^+(s))^2/\delta)}{N_k^+(s)}},$$

$$\sum_{s'} \left| (P(s' \mid s, a, b) - \hat{P}_k(s' \mid s, a, b)) V(s') \right| \leq \frac{R_{\max}}{1 - \gamma} \sqrt{\frac{8 l_k(s, a, b)}{N_k^+(s, a, b)}},$$

$$\sum_{s'} \left| (P(s' \mid s, a, b) - \hat{P}_k(s' \mid s, a, b)) \hat{V}(s') \right| \leq \frac{R_{\max}}{1 - \gamma} \sqrt{\frac{8 l_k(s, a, b)}{N_k^+(s, a, b)}}$$

$$\sum_{b \in \mathcal{B}} |\nu(b \mid s) - \hat{\nu}(b \mid s)| \leq \sqrt{\frac{2|\mathcal{B}| \log(10|\mathcal{S}||\mathcal{B}|(N_k^+(s))^2/\delta)}{N_k^+(s)}}$$

$$\frac{1}{\hat{\nu}(b \mid s)\nu(b \mid s)} \leq \frac{1}{\Delta_{\min}^2} \sqrt{2 \frac{\log(10|\mathcal{S}||\mathcal{B}|(N_k^+(s))^2/\delta)}{N_k^+(s)}}$$

*where we introduced $l_k(s, a, b) := \log\left(\frac{30|\mathcal{S}||\mathcal{A}||\mathcal{B}|(N_k^+(s,a,b))^2}{\delta}\right)$.*

*Proof.* We start the proof by defining the good event for the transition model, which proceeds in a similar way as already seen in Lemma C.8. We start with bound of the transition dynamics. Note that here we prove something stronger, that the good event holds for all $s, a, b$, therefore also for $\max_{b \in \mathcal{B}}$. Therefore we define $l_k(s, a, b) := \log\left(\frac{30|\mathcal{S}||\mathcal{A}||\mathcal{B}|(N_k^+(s,a,b))^2}{\delta}\right)$ and additionally we denote $\beta_{N_k(s,a,b)} = \frac{\gamma R_{\max}}{1-\gamma} \sqrt{\frac{2 l_k(s,a,b)}{N_k^+(s,a,b)}}$. Now we define the set

$$\mathcal{E}^{\text{trans}} := \left\{ \forall k \in \mathbb{N}, \forall (s, a, b) \in \mathcal{S} \times \mathcal{A} \times \mathcal{B} : \sum_{s'} \left| P(s' \mid s, a, b) - \hat{P}(s' \mid s, a, b) \right| V(s') \leq \beta_{N_k(s,a,b)} \right\}.$$

Then we get for $V$ with probability of $1 - \delta$:

$$\mathbb{P}\left( (\mathcal{E}^{\text{trans}})^C \right)$$
$$= \mathbb{P}\left( \exists k \geq 1, \exists (s, a, b) \in \mathcal{S} \times \mathcal{A} \times \mathcal{B} : \right.$$
$$\left. \sum_{s'} \left| \left( P(s' \mid s, a, b) - \hat{P}_k(s' \mid s, a, b) \right) V(s') \right| > \beta_{N_k(s,a,b)}(s, a, b) \right)$$
$$\overset{(a)}{\leq} \sum_{(s,a,b) \in \mathcal{S} \times \mathcal{A} \times \mathcal{B}} \mathbb{P}\left( \exists k \geq 1 : \sum_{s'} \left| \left( P(s' \mid s, a, b) - \hat{P}_k(s' \mid s, a, b) \right) V(s') \right| > \beta_{N_k(s,a,b)} \right)$$
$$\overset{(b)}{\leq} \sum_{(s,a,b) \in \mathcal{S} \times \mathcal{A} \times \mathcal{B}} \mathbb{P}\left( \exists m \geq 0 : \sum_{s'} \left| \left( P(s' \mid s, a, b) - \hat{P}_k(s' \mid s, a, b) \right) V(s') \right| > \beta_{N_k(s,a,b)} \right)$$
$$\overset{(c)}{\leq} \sum_{(s,a,b) \in \mathcal{S} \times \mathcal{AB}} \sum_{m \geq 0} 2 \exp\left( \frac{\beta_{N_k(s,a,b)}^2 m^2 (1 - \gamma)^2}{4 m \gamma^2 R_{\max}^2} \right)$$
$$\leq \sum_{(s,a,b) \in \mathcal{S} \times \mathcal{A} \times \mathcal{B}} \sum_{m \geq 0} \frac{\delta}{15|\mathcal{S}||\mathcal{A}||\mathcal{B}|(m^+)^2}$$
$$\leq \frac{\delta}{15}\left(1 + \frac{\pi^2}{6}\right) \leq \frac{\delta}{5},$$

where (a) uses a union bound over the state and joint-action space, (b) uses that we only consider the $m$-times, where we updated the estimated transition model and (c) uses an union bound over the update times $m$ and an application of Hoeffding's inequality combined with the fact that we can bound the value function, i.e. $V^i(s') \leq \frac{\gamma R_{\max}}{1-\gamma}$ for every $s' \in \mathcal{S}$. Next, we will consider the

first equation regarding estimating the log probability of the expert policy. In a first step we define $\beta_2(s) := \frac{1}{\Delta_{\min}}\sqrt{\frac{\log(10|\mathcal{S}||\mathcal{A}|(N_k^+(s))^2/\delta)}{2N_k^+(s)}}$ the following set

$$\mathcal{E}^{\log} := \left\{\forall k \in \mathbb{N}, \forall(s,a) \in \mathcal{S}\times\mathcal{A} : |\log(\mu(a\mid s)) - \log(\hat{\mu}(a\mid s))| \leq \beta_2(s)\right\}.$$

$$\mathbb{P}\left((\mathcal{E}^{\log})^C\right) = \mathbb{P}\left(\exists k \geq 1, \exists(s,a) \in \mathcal{S}\times\mathcal{A} : |\log(\mu(a\mid s)) - \log(\hat{\mu}_k(a\mid s))| > \beta_2(s)\right)$$

$$\overset{(a)}{\leq} \sum_{(s,a)\in\mathcal{S}\times\mathcal{A}} \mathbb{P}\left(\exists k \geq 1 : |\log(\mu(a\mid s)) - \log(\hat{\mu}_k(a\mid s))| > \beta_2(s)\right)$$

$$\overset{(b)}{\leq} \sum_{(s,a)\in\mathcal{S}\times\mathcal{A}} \mathbb{P}\left(\exists m \geq 0 : |\log(\mu(a\mid s)) - \log(\hat{\mu}_m(a\mid s))| > \beta_2(s)\right)$$

$$\overset{(c)}{\leq} \sum_{(s,a)\in\mathcal{S}\times\mathcal{A}}\sum_{m\geq 0} \frac{\delta}{10|\mathcal{S}||\mathcal{A}|m^2}$$

$$\leq \frac{\delta}{10}\left(1 + \frac{\pi^2}{6}\right) \leq \frac{\delta}{5},$$

where (a) uses a union bound over the state and action space of player 1, (b) uses that we only consider the $m$-times, where we updated the estimated transition model and (c) we can applied Lemma H.3. To be precise, $(c)$ only holds if $N$ is large enough, however, we will late only consider this case, therefore we use it directly.

For the second last step we define the good event for the total variation

$$\mathcal{E}^{\mathrm{TV}} := \left\{\forall k \in \mathbb{N}, \forall(s,b) \in \mathcal{S}\times\mathcal{B} : \mathrm{TV}(\nu,\hat{\nu}) \leq \sqrt{\frac{|\mathcal{B}|\log(10|\mathcal{S}||\mathcal{B}|(N_k^+(s))^2/\delta)}{N_k^+(s)}}\right\}.$$

We will the bound the probability of the complement of this event by $\delta$ and can then take the intersection of both events to get the total result. We will skip some intermediate steps, as they are similar to the ones obtained above.

$$\mathbb{P}((\mathcal{E}^{\mathrm{TV}})^C) = \mathbb{P}\left(\exists k \in \mathbb{N}, \exists(s,b) \in \mathcal{S}\times\mathcal{B} : \mathrm{TV}(\nu,\hat{\nu}) > \sqrt{\frac{|\mathcal{B}|\log(10|\mathcal{S}||\mathcal{B}|(N_k^+(s))^2/\delta)}{N_k^+(s)}}\right)$$

$$\overset{(a'')}{\leq} \sum_{(s,a)\in\mathcal{S}\times\mathcal{A}}\sum_{m\geq 0} \mathbb{P}\left(\mathrm{TV}(\nu,\hat{\nu}) > \sqrt{\frac{|\mathcal{B}|\log(10|\mathcal{S}||\mathcal{B}|m^2/\delta)}{m}}\right)$$

$$\overset{(b'')}{\leq} \frac{\delta}{5},$$

where (a") uses a union bound over the state and action space, (b") uses Lemma H.5. Also here to be precise, $(b'')$ only holds if $N$ is large enough, however, we will late only consider this case, therefore we use it directly. To bound the second last event, we can apply Lemma H.4. To complete this proof, we first define $\beta_3(s) := \frac{1}{\Delta_{\min}^2}\sqrt{\frac{2\log(10|\mathcal{S}||\mathcal{B}|(N_k^+(s))^2/\delta)}{N_k^+(s)}}$ and once again apply the same argument for the good event,

$$\mathcal{E}^{\mathrm{invprop}} := \left\{\forall k \in \mathbb{N}, \forall(s,b) \in \mathcal{S}\times\mathcal{B} : \frac{1}{\nu(b\mid s)\hat{\nu}(b\mid s)} \leq \beta_3(s)\right\},$$

now combined with Lemma H.4. As all the good events holds with probability $\delta/5$, we get that

$$\mathbb{P}\left(\mathcal{E}^{\mathrm{TV}} \cap \mathcal{E}^{\log} \cap \mathcal{E}^{trans} \cap \mathcal{E}^{\mathrm{invprop}}\right) > 1 - \delta.$$

$\square$

Following, we present the reward uncertainty metric, which allows us to demonstrate that the difference between the recovered reward function and the true reward function is bounded.

**Definition E.8** (Reward Uncertainty). Let $k$ be the number of iterations. Then the reward uncertainty after $k$ iterations for any $(s, a, b) \in \mathcal{S} \times \mathcal{A} \times \mathcal{B}$ is defined as

$$
C_k(s, a, b) := \frac{4\gamma R_{\max}}{(1 - \gamma)\Delta_{\min}^2} \left( \sqrt{\frac{8|\mathcal{B}|\log(10|\mathcal{S}||\mathcal{B}|(N_k^+(s))^2/\delta)}{N_k^+(s)}} \right.
$$
$$
\left. + \sqrt{\frac{2\log(10|\mathcal{S}||\mathcal{A}|(N_k^+(s))^2/\delta)}{N_k^+(s)}} + \sqrt{\frac{2l_k(s, a, b)}{N_k^+(s, a, b)}} \right).
$$

**Theorem E.9.** *Let the reward uncertainty be defined as in C.9. Under the good event it holds for any $(s, a, b) \in \mathcal{S} \times \mathcal{A}$ that:*
$$
|R^i(s, a, b) - \hat{R}^i(s, a, b)| \leq C_k(s, a, b).
$$

*Proof.* The theorem is an application of the error propagation Theorem E.5, followed by Lemma E.7

$$
|R^i(s, a, b) - \hat{R}^i(s, a, b)|
$$
$$
\leq \frac{2}{\nu^*(b \mid s)\hat{\nu}^*(b \mid s)} \left( \lambda |\log \mu^*(a \mid s) - \log \hat{\mu}^*(a \mid s)| \right.
$$
$$
\left. + \gamma \max_b |\sum_{s'} V(s')P(s' \mid s, a, b) - \hat{P}(s' \mid s, a, b)| + R_{\max}\mathrm{TV}(\mu, \hat{\mu}) \right)
$$
$$
\leq \frac{4R_{\max}}{1 - \gamma} \left( \frac{1}{\Delta_{\min}^2}\sqrt{2\frac{\log(10|\mathcal{S}||\mathcal{B}|(N_k^+(s))^2/\delta)}{N_k^+(s)}} + \sqrt{\frac{2|\mathcal{B}|\log(10|\mathcal{S}||\mathcal{B}|(N_k^+(s))^2/\delta)}{N_k^+(s)}} \right.
$$
$$
\left. + \frac{1}{\Delta_{\min}}\sqrt{\frac{2\log(10|\mathcal{S}||\mathcal{A}|(N_k^+(s))^2/\delta)}{N_k^+(s)}} + \gamma\sqrt{\frac{2l_k(s, a, b)}{N_k^+(s, a, b)}} \right)
$$
$$
\leq \frac{4\gamma R_{\max}}{(1 - \gamma)\Delta_{\min}^2} \left( \sqrt{\frac{8|\mathcal{B}|\log(10|\mathcal{S}||\mathcal{B}|(N_k^+(s))^2/\delta)}{N_k^+(s)}} + \sqrt{\frac{2\log(10|\mathcal{S}||\mathcal{A}|(N_k^+(s))^2/\delta)}{N_k^+(s)}} \right.
$$
$$
\left. + \sqrt{\frac{2l_k(s, a, b)}{N_k^+(s, a, b)}} \right)
$$
$$
= C_k(s, a, b).
$$

$\square$

Before stating the correctness of the algorithm, we want to mention that the derivations of Lemma C.6 also hold for the regularized case. Therefore, we can continue with the the correctness result for the regularized case.

**Corollary E.10.** *Let $k$ be the number of iterations for any allocation of the samples over the state-action space $\mathcal{S} \times \mathcal{A}$. Furthermore, let $\mathcal{R}_{(\mathcal{G}, \boldsymbol{\pi}^{\mathrm{Nash}})}$ be the true feasible set and $\mathcal{R}_{(\hat{\mathcal{G}}, \hat{\boldsymbol{\pi}}^{\mathrm{Nash}})}$ the recovered one. Then the optimality criterion 3.3 holds true, if*

$$
\frac{1}{1 - \gamma} \max_{(s,a,b)} C_k(s, a, b) \leq \frac{\varepsilon}{2}.
$$

*Proof.* We complete the proof for the first case of the optimality criterion, the second one follows analogously.

$$
\sup_{R \in \mathcal{R}_{(\mathcal{G}, \boldsymbol{\pi}^{\mathrm{Nash}})}} \inf_{\hat{R} \in \mathcal{R}_{(\hat{\mathcal{G}}, \hat{\boldsymbol{\pi}}^{\mathrm{Nash}})}} \max\{\max_{\mu_\lambda} V_\lambda^1(\mu, \hat{\nu}) - V_\lambda^1(\hat{\mu}, \hat{\nu}), V_\lambda^1(\hat{\mu}, \hat{\nu}) - \min_{\nu_\lambda} V_\lambda^1(\hat{\mu}, \nu)\}
$$
$$
\leq \frac{2}{1 - \gamma} \max_{(s,a,b)} C_k(s, a, b) \leq \varepsilon,
$$

where we applied C.6 and we used the fact that $a + b \leq 2\max\{a, b\}$ followed by the error propagation Theorem E.5 and then we used C.10. $\square$

We can combine the derived results to now state the main theorem regarding the sample complexity of the Uniform Sampling

**Theorem E.11.** *Let Assumption 3.8 hold true. Then, allocating the samples uniformly over $\mathcal{S} \times \mathcal{A} \times \mathcal{B}$ and using the empirical estimators introduced in Eq. (13) and Eq. (12), we can stop the sampling procedure with a probability of at least $1 - \delta$ after iteration $\tau$ and satisfy the optimality criterion Definition E.2, where the sample complexity is of order*

$$\tilde{\mathcal{O}}\left(\frac{\gamma^2 R_{\max}^2 |\mathcal{S}||\mathcal{A}||\mathcal{B}|}{(1-\gamma)^4 \varepsilon^2 \Delta_{\min}^4}\right)$$

*Proof.* We know from C.11, that we need

$$\frac{1}{1-\gamma} \max_{(s,a,b)} C_k(s,a,b) \leq \frac{\varepsilon_k}{2}$$

By the definition of $C_k(s,a,b)$ this is satisfied if

$$\frac{4\gamma R_{\max}}{(1-\gamma)^2 \Delta_{\min}^2} \sqrt{\frac{2l_k(s,a,b)}{N_k^+(s,a,b)}} \leq \frac{\varepsilon_k}{6}$$

$$\frac{4\gamma R_{\max}}{(1-\gamma)^2 \Delta_{\min}^2} \sqrt{\frac{\log(10|\mathcal{S}||\mathcal{A}|(N_k^+(s))^2/\delta)}{N_k^+(s)}} \leq \frac{\varepsilon_k}{6},$$

$$\frac{4\gamma R_{\max}}{(1-\gamma)^2 \Delta_{\min}^2} \sqrt{\frac{8|\mathcal{B}|\log(|\mathcal{S}||\mathcal{B}|(N_k^+(s))^2/\delta)}{N_k^+(s)}} \leq \frac{\varepsilon_k}{6}.$$

To achieve the first condition, we get

$$N_k(s,a,b) \geq \frac{R_{\max}^2}{(1-\gamma)^4 \Delta_{\min}^4} \gamma^2 24^2 l_k(s,a,b) \frac{2}{\varepsilon_k^2}$$

$$= \frac{\gamma^2 1152 R_{\max}^2}{(1-\gamma)^4 \Delta_{\min}^4 \varepsilon_k^2} \log\left(\frac{12|\mathcal{S}||\mathcal{B}||\mathcal{A}|(N_k^+(s,a,b))^2}{\delta}\right)$$

Applying Lemma B.8 by Metelli et al. [2021] we get that

$$N_k(s,a,b) \leq \frac{4608 \gamma^2 R_{\max}^2}{(1-\gamma)^4 \varepsilon_k^2 \Delta_{\min}^4} \log\left(\frac{2304 \gamma^2 R_{\max}^2}{(1-\gamma)^4 \varepsilon_k^2} \sqrt{\frac{12|\mathcal{S}||\mathcal{B}||\mathcal{A}|}{\delta}}\right).$$

At each iteration we are allocating the samples uniformly over $\mathcal{S} \times \mathcal{A} \times \mathcal{B}$ and recalling that $\tau_{s,a,b} = |\mathcal{S}||\mathcal{B}||\mathcal{A}| N_k(s,a,b)$ therefore we get

$$\tau_{s,a,b} \leq \frac{4608 \gamma^2 |\mathcal{S}||\mathcal{B}||\mathcal{A}| R_{\max}^2}{(1-\gamma)^4 \Delta_{\min}^4 \varepsilon_k^2} \log\left(\frac{2304 \gamma^2 R_{\max}^2}{(1-\gamma)^4 \varepsilon_k^2 \Delta_{\min}^4} \sqrt{\frac{12|\mathcal{S}||\mathcal{B}||\mathcal{A}|}{\delta}}\right)$$

Now we have to achieve the second condition,

$$N_k(s) \geq \frac{24^2 \gamma^2 R_{\max}^2}{(1-\gamma)^4 \Delta_{\min}^4} \frac{\log(10|\mathcal{S}||\mathcal{A}|(N_k^+(s))^2/\delta)}{\varepsilon_k^2}$$

$$= \frac{\gamma^2 576 R_{\max}^2}{(1-\gamma)^4 \varepsilon_k^2 \Delta_{\min}^4} \log(10|\mathcal{S}||\mathcal{A}|(N_k^+(s))^2/\delta)$$

$$= \frac{\gamma^2 576 R_{\max}^2}{(1-\gamma)^4 \varepsilon_k^2 \Delta_{\min}^4} \left(\log(10|\mathcal{S}||\mathcal{A}|/\delta) + 2\log((N_k^+(s)))\right)$$

If we additionally force that $N_k(s) \geq 1$ and apply Lemma 15 of Kaufmann et al. [2021] with $1/\Delta^2 = \frac{\gamma^2 576 R_{\max}^2}{(1-\gamma)^4 \varepsilon_k^2 \Delta_{\min}^4}, a = \log(10|\mathcal{S}||\mathcal{A}|/\delta), b = 2, c = 0, d = 1$, we get that

$$N_k(s) \leq 1 + \frac{\gamma^2 576 R_{\max}^2}{(1-\gamma)^4 \varepsilon_k^2 \Delta_{\min}^4} \left(\log(10|\mathcal{S}||\mathcal{A}|/\delta)\right.$$

$$\left. + 2\log\left(\left(\frac{\gamma^2 576 R_{\max}^2}{(1-\gamma)^4 \varepsilon_k^2 \Delta_{\min}^4}\right)^2 (\log(10|\mathcal{S}||\mathcal{A}|/\delta) + 2)\right)\right)$$

As we here need to allocate samples uniformly over the state space only but for every agent separately and recalling that $\tau_{s_1} = SN_k(s)$, we get

$$\tau_{s_1} \leq |\mathcal{S}| + \frac{|\mathcal{S}|\gamma^2 576 R_{\max}^2}{(1-\gamma)^4 \varepsilon_k^2 \Delta_{\min}^4} \bigg( \log(10|\mathcal{S}||\mathcal{A}|/\delta)$$
$$+ 2\log\left( \left( \frac{\gamma^2 576 R_{\max}^2}{(1-\gamma)^4 \varepsilon_k^2 \Delta_{\min}^4} \right)^2 (\log(10|\mathcal{S}||\mathcal{A}|/\delta) + 2) \right) \bigg).$$

Lastly, we can proceed in a similar way compared to the last step,

$$N_k(s) \geq \frac{R_{\max}^2}{(1-\gamma)^4 \Delta_{\min}^4} \gamma^2 24^2 \frac{8|\mathcal{B}|\log(10|\mathcal{S}||\mathcal{B}|(N_k^+(s))^2/\delta)}{\varepsilon_k^2}$$
$$= \frac{\gamma^2 4608 R_{\max}^2 |\mathcal{B}|}{(1-\gamma)^4 \varepsilon_k^2 \Delta_{\min}^4} \log(10|\mathcal{S}||\mathcal{B}|(N_k^+(s))^2/\delta)$$

If we additionally force that $N_k(s) \geq 1$ and again apply Lemma 15 of Kaufmann et al. [2021] with $1/\Delta^2 = \frac{|\mathcal{B}|\gamma^2 4608 R_{\max}^2}{(1-\gamma)^4 \varepsilon_k^2 \Delta_{\min}^4}, a = \log(10|\mathcal{S}||\mathcal{A}|/\delta), b = 2, c = 0, d = 1$ we get that

$$N_k(s) \leq 1 + \frac{|\mathcal{B}|\gamma^2 4608 R_{\max}^2}{(1-\gamma)^4 \varepsilon_k^2 \Delta_{\min}^4} \bigg( \log(10|\mathcal{S}||\mathcal{B}|/\delta)$$
$$+ 2\log\left( \left( \frac{|\mathcal{B}|\gamma^2 4608 R_{\max}^2}{(1-\gamma)^4 \varepsilon_k^2 \Delta_{\min}^4} \right)^2 (\log(10|\mathcal{S}||\mathcal{B}|/\delta) + 2) \right) \bigg)$$

As we here need to allocate samples uniformly over the state space only but for every agent separately and recalling again that $\tau_{s_2} = SN_k(s)$, we get

$$\tau_{s_2} \leq |\mathcal{S}| + \frac{|\mathcal{S}||\mathcal{B}|\gamma^2 4608 R_{\max}^2}{(1-\gamma)^4 \varepsilon_k^2 \Delta_{\min}^4} \bigg( \log(10|\mathcal{S}||\mathcal{B}|/\delta)$$
$$+ 2\log\left( \left( \frac{|\mathcal{B}|\gamma^2 4608 R_{\max}^2}{(1-\gamma)^4 \varepsilon_k^2 \Delta_{\min}^4} \right)^2 (\log(10|\mathcal{S}||\mathcal{B}|/\delta) + 2) \right) \bigg).$$

Finally, as $\tau = \tau_{s,a} + \tau_{s_1} + \tau_{s_2}$ we get that this is exactly of order

$$\tilde{\mathcal{O}}\left( \frac{\gamma^2 R_{\max}^2 |\mathcal{S}||\mathcal{A}||\mathcal{B}|}{(1-\gamma)^4 \varepsilon^2 \Delta_{\min}^4} \right)$$

$\square$

One can see that the problem scales with $|\mathcal{A}||\mathcal{B}|$, which is due to the used union bound. Scaling this up to $n$-player games the union bound implies that the bound will scale exponentially in the number of players. Additionally, as we have to bound the inverse probability the term $\Delta_{\min}$ appears on the bound. A sufficiently large $\Delta_{\min}$ can e.g. be obtained if the regularization parameter $\lambda$ is small.

## F  Identifiability in multi-agent games?

This appendix provides supplementary details and proofs for the identifiability results discussed in Section 4. We follow the notation established in the main text, where applicable, using $R^1(s, a, b)$ to denote Player 1's reward, $R^\nu(s, a) = \sum_{b' \in \mathcal{B}} \nu(b' \mid s) R^1(s, a, b')$ for the average reward received by Player 1 when Player 2 uses policy $\nu$, and $\hat{P}_a^\nu$ for the induced transition matrix for Player 1 under $\nu$. $\lambda_1$ denotes Player 1's entropy regularization parameter.

**Average identifiability.**    We start with the case, where we try to identify the average reward function (up to constants) for any player. The theorem is a direct consequence of Theorem 3 by Rolland et al. [2022].

**Theorem F.1.** *Let a Markov game be given with two different opponents $\nu_1, \nu_2$ that induce different dynamics $P^{\nu_1}, P^{\nu_2}$ and discount factors $\gamma_1, \gamma_2$. Suppose that in both Games we observe QRE equilibrium policy pairs $(\mu_1, \nu_1)$ and a different $\nu_2$ with a best responding policy $\mu_2$ such that they have same average reward functions $R^{\nu_1} = R^{\nu_2}$. Additionally, define $P_a^{\nu_i} \in \mathbb{R}^{\mathcal{S} \times \mathcal{S}}$ the induced transition matrix of expert $i \in \{1, 2\}$. Then, the* average *reward player 1 receives can be recovered up to a constant if and only if*

$$
rank \begin{pmatrix} I - \gamma_1 P_{a_1}^{\nu_1} & I - \gamma_2 P_{a_1}^{\nu_2} \\ \vdots & \vdots \\ I - \gamma_1 P_{a_{|\mathcal{A}|}}^{\nu_1} & I - \gamma_2 P_{a_{|\mathcal{A}|}}^{\nu_2} \end{pmatrix} = 2|\mathcal{S}| - 1.
$$

*Analogously this holds for player 2.*

**Theorem F.2** (Sample Complexity for Induced Transitions)**.** *To estimate the induced transition model $P^{\nu^*}$ for Player 1 (where $\nu^*$ is Player 2's true policy) such that the maximum $L_1$ error over all $(s, a)$ rows is bounded by $\varepsilon$, i.e., $\max_{s,a} \|P^{\nu^*}(\cdot \mid s, a) - \widehat{P}^{\hat{\nu}}(\cdot \mid s, a)\|_1 \leq \varepsilon$, with probability at least $1 - \delta$, the total number of samples $N$ is of the order:*

$$
\mathcal{O}\left( \frac{|\mathcal{S}|^2 |\mathcal{A}||\mathcal{B}| \log(|\mathcal{S}||\mathcal{A}||\mathcal{B}|/\delta)}{\varepsilon^2} \right)
$$

*where $\widehat{P}^{\hat{\nu}}(s'|s, a) = \sum_{b \in \mathcal{B}} \hat{\nu}(b|s) \hat{P}(s'|s, a, b)$ uses empirical estimates $\hat{\nu}$ of $\nu^*$ and $\hat{P}$ of the true dynamics $P$.*

*Proof.* The estimated induced transition is $\widehat{P}^{\hat{\nu}}(\cdot|s, a) = \sum_b \hat{\nu}(b|s) \hat{P}(\cdot|s, a, b)$. The true induced transition is $P^{\nu^*}(\cdot|s, a) = \sum_b \nu^*(b|s) P(\cdot|s, a, b)$. We want to bound the $L_1$ error. We use the triangle inequality and properties of the $L_1$ norm:

$$
\|P^{\nu^*}(\cdot|s, a) - \widehat{P}^{\hat{\nu}}(\cdot|s, a)\|_1
$$

$$
= \left\| \sum_b \nu^*(b|s) P(\cdot|s, a, b) - \sum_b \hat{\nu}(b|s) \hat{P}(\cdot|s, a, b) \right\|_1
$$

$$
= \left\| \sum_b (\nu^*(b|s) - \hat{\nu}(b|s)) P(\cdot|s, a, b) + \sum_b \hat{\nu}(b|s)(P(\cdot|s, a, b) - \hat{P}(\cdot|s, a, b)) \right\|_1
$$

$$
\leq \sum_b |\nu^*(b|s) - \hat{\nu}(b|s)| \cdot \|P(\cdot|s, a, b)\|_1 + \sum_b \hat{\nu}(b|s) \|P(\cdot|s, a, b) - \hat{P}(\cdot|s, a, b)\|_1
$$

Since $\|P(\cdot|s, a, b)\|_1 = 1$ and $\sum_b \hat{\nu}(b|s) = 1$:

$$
\|P^{\nu^*}(\cdot|s, a) - \widehat{P}^{\hat{\nu}}(\cdot|s, a)\|_1 \leq \|\nu^*(\cdot|s) - \hat{\nu}(\cdot|s)\|_1 + \max_{b' \in \mathcal{B}} \|P(\cdot|s, a, b') - \hat{P}(\cdot|s, a, b')\|_1
$$

To ensure $\|P^{\nu^*}(\cdot|s, a) - \widehat{P}^{\hat{\nu}}(\cdot|s, a)\|_1 \leq \varepsilon$ with high probability, we need to ensure both terms on the right are sufficiently small, e.g., $\leq \varepsilon/2$.

Estimating the multinomial distribution $\nu^*(\cdot|s)$ over $|\mathcal{B}|$ actions requires $N_\nu(s)$ samples of Player 2's actions at state $s$. Applying Lemma H.5 gives us that to achieve $\|\nu^*(\cdot|s) - \hat{\nu}(\cdot|s)\|_1 \leq \varepsilon/2$ with probability $1 - \delta'$, requires that $N_\nu(s)$ is of the order $\mathcal{O}\left( \frac{|\mathcal{B}|}{\varepsilon^2} \right)$ samples.

Similarly, we can bound the transition dynamics. In particular, if we apply Lemma H.5, then we get that the amount of samples required to minimize it with high probability is of the order $\mathcal{O}\left( \frac{|\mathcal{S}||\mathcal{A}||\mathcal{B}|}{\varepsilon^2} \right)$. As the part of the transition dynamics dominates, the total number of samples is then of the order is then given by

$$
\mathcal{O}\left( \frac{|\mathcal{S}||\mathcal{A}||\mathcal{B}|}{\varepsilon^2} \right).
$$

$\square$

With this result we recover Theorem 8 by Rolland et al. [2022] for every player. For completeness we restate the result here.

**Theorem F.3.** *Suppose that we estimate the transition dynamics $P_a^{\nu_1}$ and $P_a^{\nu_1}$ by $P_a^{\tilde{\nu}_1}$ and $P_a^{\tilde{\nu}_2}$ such that $\|P_a^{\nu_1} - P_a^{\tilde{\nu}_1}\|_1 \leq \varepsilon$ and $\|P_a^{\nu_2} - P_a^{\tilde{\nu}_2}\|_1 \leq \varepsilon \quad \forall a \in \mathcal{A}$. Assume that the estimated transition dynamics satisfy Eq. (3), then the true transition dynamics satisfy Eq. (3), if for the second smallest eigenvalue $\sigma$ of the following matrix*

$$
\begin{pmatrix}
I - \gamma_1 P_{a_1}^{\tilde{\nu}_1} & I - \gamma_2 P_{a_1}^{\tilde{\nu}_2} \\
\vdots & \vdots \\
I - \gamma_1 P_{a_{|\mathcal{A}|}}^{\tilde{\nu}_1} & I - \gamma_2 P_{a_{|\mathcal{A}|}}^{\tilde{\nu}_2}
\end{pmatrix}
$$

*it holds true that*

$$
\sigma > \varepsilon \sqrt{2|\mathcal{A}|} \max\{\gamma_1, \gamma_2\}.
$$

**Reward identification in linear separable Markov games.** As the so far discussed theorems only work for the average reward case, we want to identify conditions that allow us to identify the rewards in the multi-agent case. As discussed in Section 4 one potential solution to achieve this is to assume linear separable rewards that naturally disentangle the rewards into a part that only depends on action $a$ and a part that only depends on action $b$.

We can immediately see that $R^{\nu^*}(s, a) = \sum_{b \in \mathcal{B}} \nu^*(b \mid s) R(s, a, b) = R_A(s, a) + \sum_{b \in \mathcal{B}} \nu^*(b \mid s) R_B(s, b)$. This means, that the average reward equation Eq. (2), can be rewritten. In particular, we get for every $(s, a) \in \mathcal{S} \times \mathcal{A}$

$$
R_A(s, a) = \lambda \log(\mu^*(a \mid s)) + V(s) - \gamma \sum_{s'} P^{\nu^*}(s' \mid, s, a) V(s') - \sum_{b \in \mathcal{B}} \nu^*(b \mid s) R_B(s, b).
$$

Next, we again consider the case, where we have two Markov games where we have the same reward function for player 1, in particular for action $a$. Using the explicit formulation of the reward, we get the following:

$$
R_A(s, a) = \lambda \log(\mu_1^*(a \mid s)) + V_1(s) - \gamma \sum_{s'} P_1^{\nu_1^*}(s' \mid, s, a) V_1(s') - \sum_{b \in \mathcal{B}} \nu_1^*(b \mid s) R_B(s, b)
$$

$$
= \lambda \log(\mu_2^*(a \mid s)) + V_2(s) - \gamma \sum_{s'} P_2^{\nu_2^*}(s' \mid, s, a) V_2(s') - \sum_{b \in \mathcal{B}} \nu_2^*(b \mid s) R_B(s, b)
$$

Let us consider two cases. The first case is that in both environments the opponent policy is the same, meaning that we have $\nu_1^* = \nu_2^*$. Then, we see immediately that $\sum_{b \in \mathcal{B}} \nu^*(b \mid s) R_B(s, b)$ cancels out and we get

$$
\lambda \log(\mu_1^*(a \mid s)) + V_1(s) - \gamma \sum_{s'} P_1^{\nu^*}(s' \mid s, a) V_1(s') = \lambda \log(\mu_2^*(a \mid s)) + V_2(s) - \gamma \sum_{s'} P_2^{\nu^*}(s' \mid s, a) V_2(s').
$$

Therefore, we reconstruct the single-agent case, as we obtain for every $(s, a) \in \mathcal{S} \times \mathcal{A}$ :

$$
V_1(s) - V_2(s) - \gamma \sum_{s'} P_1^{\nu^*}(s' \mid, s, a) V_1(s') + \gamma \sum_{s'} P_2^{\nu^*}(s' \mid, s, a) V_2(s') = \lambda(\log(\mu_2^*(a \mid s)) - \log(\mu_1^*(a \mid s))).
$$

Therefore, we can again write this as a system of equations but this time for the same $\nu^*$. We can summarize these findings in the following result.

**Proposition F.4.** *Let a Markov game with a QRE equilibrium policy pair $(\mu_1^*, \nu_1^*)$ be given. Additionally, let another Markov game with the same $\nu_1^*$ but a different transition Model $P_2$ and discount factor $\gamma_2$ and therefore also different best response $\mu_2^*$ be given such that $R_A^1$ is the same for both environments. Then, $R_A^1$ is identifiable if and only if*

$$
rank \begin{pmatrix}
I - \gamma_1 P_{1,a_1}^{\nu_1^*} & I - \gamma_2 P_{2,a_1}^{\nu_1^*} \\
\vdots & \vdots \\
I - \gamma_1 P_{1,a_{|\mathcal{A}|}}^{\nu_1} & I - \gamma_2 P_{2,a_{|\mathcal{A}|}}^{\nu_1^*}
\end{pmatrix} = 2|\mathcal{S}| - 1.
$$

*Analogously this holds for player 2.*

For the second case, we assume that the opponent policies are different in the two Markov games, meaning that $\nu_1^* \neq \nu_2^*$. In this case we need an additional assumption, namely $R_B(s, b_0) = 0$. his constraint fixes the baseline for $R_B(s, b)$, allowing us to determine how much player 2's different actions contribute to player 1's reward, relative to this baseline.

$$V_1(s) - V_2(s) - \gamma \sum_{s'} P_1^{\nu_1^*}(s' \mid s, a) V_1(s') + \gamma \sum_{s'} P_2^{\nu_2^*}(s' \mid s, a) V_2(s')$$

$$+ \sum_{b \neq b_0} R_B(s, b)(\nu_2^*(b \mid s) - \nu_1^*(b \mid s)) = \lambda \log(\mu_2^*(a \mid s)) - \lambda \log(\mu_1^*(a \mid s)).$$

This we can now again express as a system of equations as the above holds for every $(s, a) \in \mathcal{S} \times \mathcal{A}^1$ and we get:

$$\begin{pmatrix} M_{R_B} & I - \gamma_1 P_{a_1}^{\nu_1} & I - \gamma_2 P_{a_1}^{\nu_2} \\ \vdots & \vdots & \vdots \\ M_{R_B} & I - \gamma_1 P_{a_{|\mathcal{A}|}}^{\nu_1} & I - \gamma_2 P_{a_{|\mathcal{A}|}}^{\nu_2} \end{pmatrix} \begin{pmatrix} R_B \\ V_1 \\ V_2 \end{pmatrix} = \begin{pmatrix} \lambda(\log(\mu_2(\cdot \mid a_1)) - \log(\mu_1(\cdot \mid a_1))) \\ \vdots \\ \lambda(\log(\mu_2(\cdot \mid a_{|\mathcal{A}|})) - \log(\mu_1(\cdot \mid a_{|\mathcal{A}|}))) \end{pmatrix},$$

where $R_B$ is the reward vector for every $s \in \mathcal{S}$ and $b \in \mathcal{B} \setminus \{b_0\}$ and each $M_{R_B}$ is an $|\mathcal{S}| \times |\mathcal{S}||\mathcal{B}|$ block diagonal matrix with the following structure

$$M_{R_B} = \begin{pmatrix} \boldsymbol{\nu}(s_1) & \mathbf{0} & \dots & \mathbf{0} \\ \mathbf{0} & \boldsymbol{\nu}(s_2) & \dots & \mathbf{0} \\ \vdots & \vdots & \ddots & \vdots \\ \mathbf{0} & \mathbf{0} & \dots & \boldsymbol{\nu}(s_{|\mathcal{S}|}) \end{pmatrix},$$

where each $\boldsymbol{\nu}(s_i)$ for $s_i \in \mathcal{S}$ is a $1 \times (|\mathcal{B}| - 1)$ row vector where every element is the difference of the policies for the actions $\nu_2(b \mid s) - \nu_1(b \mid s)$ for $b \in \mathcal{B} \setminus \{b_0\}$.

This means that the matrix on the left has shape $|\mathcal{A}||\mathcal{S}| \times |\mathcal{S}|(|\mathcal{B}| + 1)$. As we require identification up to constants we need that the matrix has rank $|\mathcal{S}|(|\mathcal{B}| + 1) - 1$. We can conclude our findings in the following proposition.

**Proposition F.5.** *Let us assume that we have two-player general-sum Markov games with different transition functions $P_1, P_2$ and $\gamma_1, \gamma_2$ be given. Additionally, assume that the rewards for player 1 are the same under the QRE $(\mu_1^*, \nu_1^*)$ and the other observed best responding policy $\mu_2^*$ to $\nu_2^*$. Additionally, suppose player 1's reward function is linearly separable: $R_1(s, a, b) = R_A(s, a) + R_B(s, b)$. To ensure unique decomposition between $R_A$ and $R_B$, the normalization $R_B(s, b_0) = 0$ is imposed for some fixed $b_0 \in \mathcal{B}$ and for all $s \in \mathcal{S}$. Then, player 1's reward function $R_1(s, a, b)$ (and its components $R_A(s, a)$ and $R_B(s, b)$ under the given normalization) can be recovered up to a single additive constant if and only if:*

$$\text{rank} \begin{pmatrix} M_{R_B} & I - \gamma_1 P_{a_1}^{\nu_1} & I - \gamma_2 P_{a_1}^{\nu_2} \\ \vdots & \vdots & \vdots \\ M_{R_B} & I - \gamma_1 P_{a_{|\mathcal{A}|}}^{\nu_1} & I - \gamma_2 P_{a_{|\mathcal{A}|}}^{\nu_2} \end{pmatrix} = |\mathcal{S}|(|\mathcal{B}| + 1) - 1.$$

# G   Experimental evaluation

In this section, we first give the details for the figures in Fig. 2 and Fig. 3. Then, we aim to demonstrate the advantages of IRL in the multi-agent setting compared to Behavior Cloning.

It is important to emphasize that the primary goal of this paper is to address the IRL problem from a theoretical perspective by defining a new objective and presenting the first algorithm to characterize the feasible set of rewards. In particular, we will demonstrate the case of pure NE strategies in a simple environment. Although, we have shown that in the general case one needs to observe multiple equilibria to infer a meaningful reward function, we will demonstrate that one can still obtain a good performance in simpler environments. We motivate this simple example given that calculating the NE is PPAD-hard. The idea is to emphasize the need for MAIRL framework and motivate future research on computationally more feasible equilibrium solutions.

### G.1 Numerical verifications for Nash and QRE equilibrium observations.

In this section, we give the details for the numerical examples provided in Fig. 2 and Fig. 3 respectively. The idea of both experiments is the same. The considered environment is the one used in Proposition 3.4 and illustrated in Fig. 2.

**Expert observations.** In the case of Nash equilibrium experts, we can simply take any Nash equilibrium solver to calculate the equilibrium for state $s_0$. This holds true because in the following states the Nash equilibrium actions of the Normal Form Game in $s_0$ are rewarded ($s_1$ and $s_3$), while the other actions are not $s_2$. Regarding the equilibrium observations for the QRE, we again only consider state $s_0$ and run a simple algorithm that iteratively computes the expected reward of a player keeping the other players strategy fixed and then updates the policy. This is repeated until the strategies of both players are not changing anymore.

**Calculating new equilibria and exploitability.** For both expert observations, we then use any IRL algorithm that picks a reward function, such that the observed equilibrium is feasible under this reward. Then, we compute again the new equilbria and compare the list of original Nash equilibria with the ones under the recovered reward function. From this list, we then randomly select an equilibrium and calculate the exploitability of the picked strategy in the original Markov game. This we repeat for $10000$ iterations and compute the average exploitability and the average correlations.

### G.2 MAIRL vs. Behavior Cloning.

In this section, we empirically evaluate the benefits of MAIRL compared to BC and describe the details on the environments in the following paragraphs. One of the advantages of IRL over BC lies in its ability to transfer the recovered reward function to new environments with different transition probabilities. This is particularly significant in a multi-agent setting, where even minor changes in transition probabilities can alter not only the individual behavior of agents but also the interactions between them.

For our experiments, we utilize the $3 \times 3$ Gridworld example, also considered in Hu and Wellman [2003]. To recover the feasible reward set and learn the expert policy with BC, we consider a scenario where the transition probabilities are deterministic. The Nash experts are learned via NashQ-Learning as proposed in Hu and Wellman [2003]. The resulting Nash Experts and more details on the environments can be found in G.2.

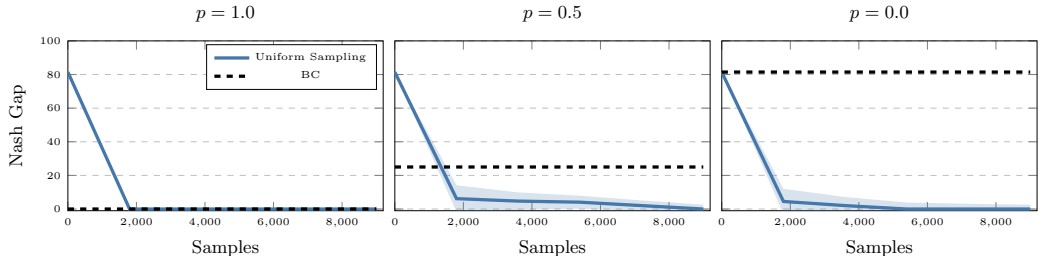

Figure 4: Nash Gap in Grid Games for different transition probabilities.

Using the Uniform Sampling algorithm to recover the entire feasible set, we then apply a Random Max Gap MAIRL algorithm to extract a reward function from the feasible set, similar to the approach introduced in Appendix C of Metelli et al. [2021]. A detailed description can be found in Appendix G.2. To test the transferability of the recovered reward function, we alter the transition probabilities so that in states $(0, \text{any})$ and $(\text{any}, 2)$, taking action "up" is only successful with a probability of $0.5$; otherwise, the agent remains in the same state (as in Grid Game 2 in Hu and Wellman [2003]). In a second scenario, we introduce an obstacle into the environment, that prohibits the agent from passing through. While in the first scenario, the BC strategy still performs reasonably, the second altered environment leads to a failure to reach the goal state for agent 1, resulting in the maximal Nash Gap.

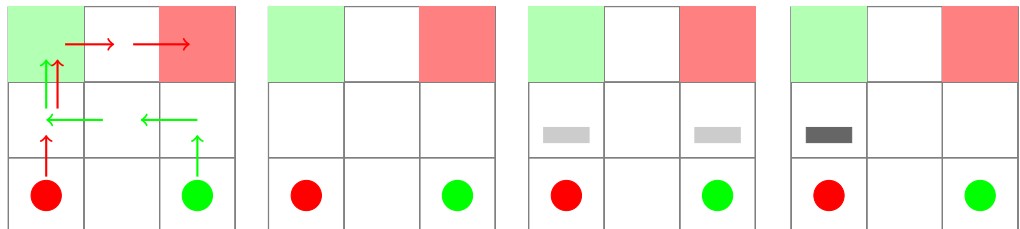

Figure 5: Multi-agent grid world environments with different transition probabilities and learned NE path

We observe that while BC may perform better in the original environment, for the first iterations, the Uniform Sampling Algorithm proves superior when transferring the reward function, especially as the number of samples increases and the environment changes.

**Multi-agent Gridworld.** In this section, we describe the environments used for the experiments. The environments are similar to the ones used in Hu and Wellman [2003]. We adjust them in such that for the random transition probabilities in the states $(0, \text{any})$ and $(2, \text{any})$ the environment still has different goals for each agent. Additionally, we introduce the scenario, where an obstacle is added in the middle of the environment, that bounces the agent back with probability 1. This results in a failure of reaching the goal for agent 1 in the BC case.
In the left column, we draw the learned Nash path for both agents in the deterministic environment, when applying the NashQ Learning algorithm to retrieve an expert policy.

**Max Gap MAIRL.** In this section, we describe the Max Gap MAIRL algorithm, an extension of the approach presented by Metelli et al. [2021] (see Appendix C in Metelli et al. [2021]) to the multi-agent setting. This algorithm is chosen due to the limited number of existing works that address the selection of feasible reward functions in general-sum Markov games with a Nash expert, particularly without imposing additional assumptions on the reward structure. Given the simplicity of the chosen environments, the Max Gap MAIRL procedure is a suitable choice.

The algorithm operates as follows: for each agent $i \in [n]$, a random reward function $\tilde{R}^i$ is selected such that $\|\tilde{R}^i\| \leq R_{\max}$. The next step involves finding a reward function $R^i$ that minimizes the squared 2-norm distance to the randomly chosen reward $\tilde{R}^i$, subject to two constraints: (1) $R^i$ must belong to the recovered feasible set, and (2) it must maximize the maximal reward gap, thereby enforcing the feasible reward condition as introduced in C.4. This results in the following constrained quadratic optimization problem:

$$\max_{R^i \in \mathbb{R}^S, A^{i,\boldsymbol{\pi}}} A^{i,\boldsymbol{\pi}}$$
$$\text{s.t.} \quad (\boldsymbol{\pi}^{\text{Nash}} - \tilde{\boldsymbol{\pi}})(I - \gamma \boldsymbol{P}\boldsymbol{\pi}^{\text{Nash}})^{-1}\boldsymbol{R}^i \geq A^{i,\boldsymbol{\pi}} \mathbf{1}_{\{\pi^{i,\text{Nash}}=0\}} \mathbf{1}_{\{\pi^{-i,\text{Nash}}>0\}} \mathbf{1}_{\mathcal{S}\times\mathcal{A}},$$
$$\|R^i\|_{\infty} \leq R_{\max}.$$

# H    Technical Results

In this section, we present results that were used throughout this work.

**Theorem H.1** (compare Theorem 1 in Cao et al. [2021]). *For a fixed policy $\bar{\pi}(a|s) > 0$, discount factor $\gamma \in [0, 1)$, and an arbitrary choice of function $v : \mathcal{S} \to \mathbb{R}$, there is a unique corresponding reward function*

$$r(s, a) = \lambda \log \bar{\pi}(a|s) - \gamma \sum_{s' \in \mathcal{S}} P(s'|s, a)v(s') + v(s)$$

*such that the MDP with reward $r$ yields a value function $V_\lambda^* = v$ and entropy-regularized optimal policy $\pi_\lambda^* = \bar{\pi}$.*

*Proof.* Fix $r$ as in the statement of the theorem. Then the corresponding value function is given by

$$V_\lambda^*(s) = \lambda \log \sum_{a\in\mathcal{A}} \exp\left(\frac{1}{\lambda}\left(r(s,a) + \gamma \sum_{s'\in\mathcal{S}} P(s'|s,a)V_\lambda^*(s')\right)\right)$$

$$= v(s) + \lambda \log \sum_{a\in\mathcal{A}} \bar\pi(a|s) \exp\left(\frac{\gamma}{\lambda} \sum_{s'\in\mathcal{S}} P(s'|s,a)(V_\lambda^*(s') - v(s'))\right),$$

which rearranges to give

$$\exp(g(s)) = \sum_{a\in\mathcal{A}} \bar\pi(a|s) \exp\left(\gamma \sum_{s'\in\mathcal{S}} P(s'|s,a)g(s')\right) \tag{15}$$

with $g(s) = (V_\lambda^*(s) - v(s))/\lambda$. Applying Jensen's inequality, we can see that, for $\underline{s} \in \arg\min_{s\in\mathcal{S}} g(s)$,

$$\exp\left(\min_s g(s)\right) = \exp\left(g(\underline{s})\right) \geq \exp\left(\gamma \sum_{a\in\mathcal{A},s'\in\mathcal{S}} \bar\pi(a|\underline{s})P(s'|\underline{s},a)g(s')\right).$$

However, the sum on the right is a weighted average of the values of $g$, so

$$\sum_{a\in\mathcal{A},s'\in\mathcal{S}} \bar\pi(a|\underline{s})P(s'|\underline{s},a)g(s') \geq \min_s g(s).$$

Combining these inequalities, along with the fact $\gamma < 1$, we conclude that $g(s) \geq 0$ for all $s \in \mathcal{S}$.

Again applying Jensen's inequality to Eq. (15), for $\bar{s} \in \arg\max_{s\in\mathcal{S}} g(s)$ we have

$$\max_s \{\exp\left(g(s)\right)\} = \exp\left(g(\bar{s})\right) \leq \sum_{a\in\mathcal{A},s'\in\mathcal{S}} \bar\pi(a|\bar{s})P(s'|\bar{s},a) \exp\left(\gamma g(s')\right).$$

As the sum on the right is a weighted average, we know

$$\sum_{a\in\mathcal{A},s'\in\mathcal{S}} \bar\pi(a|\bar{s})P(s'|\bar{s},a) \exp\left(\gamma g(s')\right) \leq \max_s \{\exp\left(\gamma g(s)\right)\}.$$

Hence, as $\gamma < 1$, we conclude that $g(s) \leq 0$ for all $s \in \mathcal{S}$.

Combining these results, we conclude that $g \equiv 0$, that is, $V_\lambda^* = v$. Finally, we substitute the definition of $r$ and the value function $v$ into (6) to see that the entropy-regularized optimal policy is $\pi_\lambda^* = \bar\pi$. $\quad\square$

The next lemma is an extension of Lemma 3 from Zanette et al. [2019] for the multi-agent setting, accounting that different Nash equilibria can have different values.

**Lemma H.2** (Simulation Lemma). *Let $i \in [n]$ be any agent. Then it holds true that*

$$\hat{V}^{i,\boldsymbol{\pi}}(s) - V^{i,\boldsymbol{\pi}}(s)$$

$$= \sum_{s,\boldsymbol{a}} \overline{w}_{s,\boldsymbol{a}}^{\boldsymbol{\pi}} \left(\hat{R}^i(s,\boldsymbol{a}) - R^i(s,\boldsymbol{a}) + \gamma\left(\sum_{s'} \left(\hat{P}(s' \mid s,\boldsymbol{a}) - P(s' \mid s,\boldsymbol{a})\right) V^{i,\boldsymbol{\pi}}(s')\right)\right)$$

*Proof.* Let the starting distribution be a dirac measure on some $s \in \mathcal{S}$. It then holds that

$$\hat{V}^{i,\boldsymbol{\pi}}(s) - V^{i,\boldsymbol{\pi}}(s)$$

$$= \hat{R}^i(s,\mathbf{a}) - R^i(s,\mathbf{a}) + \gamma\left(\sum_{s'} \hat{P}(s' \mid s,\mathbf{a})\hat{V}^{i,\boldsymbol{\pi}}(s') - P(s' \mid s,\mathbf{a})V^{i,\boldsymbol{\pi}}(s')\right)$$

$$= \hat{R}^i(s,\mathbf{a}) - R^i(s,\mathbf{a}) + \gamma\left(\hat{P}(s' \mid s,\mathbf{a}) - P(s' \mid s,\mathbf{a})\right) V^{i,\boldsymbol{\pi}}(s')$$

$$+ \gamma \sum_{s'} \hat{P}(s' \mid s,\mathbf{a})(\hat{V}^{i,\boldsymbol{\pi}}(s') - V^{i,\boldsymbol{\pi}}(s'))$$

The proof follows by induction. $\quad\square$

**Lemma H.3.** *Let $\mu^*, \nu^*$ be the QRE equilibrium expert policies, such that Assumption E.6 holds true and $\hat{\mu}^*, \hat{\nu}^*$ the respective empirical estimators with samples $N$. Then for $\delta \in (0,1)$ it holds true with probability $1 - \delta$ if $\sqrt{\frac{\log(2/\delta)}{2N_k^+(s)}} \leq \frac{\Delta_{\min}}{2}$, that for $a \in \mathcal{A}, b \in \mathcal{B}$*

$$|\log(\mu^*(a \mid s)) - \log(\hat{\mu}^*(a \mid s))| \leq \frac{2}{\Delta_{\min}}\sqrt{\frac{\log(2/\delta)}{2N}}$$

$$|\log(\nu^*(a \mid s)) - \log(\hat{\nu}^*(a \mid s))| \leq \frac{2}{\Delta_{\min}}\sqrt{\frac{\log(2/\delta)}{2N}}$$

*Proof.* The difference in logarithms can be bounded using the inequality:

$$|\log(\mu(a \mid s)) - \log(\hat{\mu}(a \mid s))| \leq \frac{|\mu(a \mid s) - \hat{\mu}(a \mid s)|}{\min(\mu(a \mid s), \hat{\mu}(a \mid s))}.$$

This follows from the fact that the derivative of $\log(x)$ is $1/x$, so the difference in logarithms is controlled by the relative difference in probabilities.

By Hoeffding's inequality, for any $(s, a) \in \mathcal{S} \times \mathcal{A}$, with probability at least $1 - \delta/(SA)$, we have:

$$|\mu(a \mid s) - \hat{\mu}(a \mid s)| \leq \sqrt{\frac{\log(2/\delta)}{2N_k^+(s)}}.$$

Since $\mu(a \mid s) \geq \Delta_{\min}$, and assuming $\hat{\mu}(a \mid s)$ is close to $\mu(a \mid s)$, we have:

$$\hat{\mu}(a \mid s) \geq \mu(a \mid s) - \sqrt{\frac{\log(2/\delta)}{2N_k^+(s)}} \geq \Delta_{\min} - \sqrt{\frac{\log(2SA/\delta)}{2N_k^+(s)}}.$$

To ensure $\hat{\mu}(a \mid s) > 0$, we require:

$$\sqrt{\frac{\log(2/\delta)}{2N_k^+(s)}} < \Delta_{\min}.$$

Under this condition, the denominator satisfies:

$$\min(\mu(a \mid s), \hat{\mu}(a \mid s)) \geq \Delta_{\min} - \sqrt{\frac{\log(2/\delta)}{2N_k^+(s)}}.$$

Substitute the bounds for $|\mu(a \mid s) - \hat{\mu}(a \mid s)|$ and $\min(\mu(a \mid s), \hat{\mu}(a \mid s))$ into the inequality for the difference in logarithms:

$$|\log(\mu(a \mid s)) - \log(\hat{\mu}(a \mid s))| \leq \frac{\sqrt{\frac{\log(2/\delta)}{2N_k^+(s)}}}{\Delta_{\min} - \sqrt{\frac{\log(2/\delta)}{2N_k^+(s)}}}.$$

If $\sqrt{\frac{\log(2/\delta)}{2N_k^+(s)}} \leq \frac{\Delta_{\min}}{2}$, then:

$$|\log(\mu(a \mid s)) - \log(\hat{\mu}(a \mid s))| \leq \frac{2\sqrt{\frac{\log(2/\delta)}{2N_k^+(s)}}}{\Delta_{\min}}.$$

$\square$

**Lemma H.4.** *Let $p_i$ be a probability such that $p_i \geq \Delta_{\min} > 0$, and let $\hat{p}_i$ be the empirical estimate of $p_i$ based on $n$ independent samples. Then, for any $\delta \in (0,1)$, with probability at least $1 - \delta$, the following bound holds:*

$$\left|\frac{1}{p_i} - \frac{1}{\hat{p}_i}\right| \leq \frac{\sqrt{\frac{\log(2/\delta)}{2n}}}{\Delta_{\min}\left(\Delta_{\min} - \sqrt{\frac{\log(2/\delta)}{2n}}\right)}.$$

*Furthermore, if* $\sqrt{\frac{\log(2/\delta)}{2n}} \leq \frac{\Delta_{\min}}{2}$, *the bound simplifies to:*

$$\left| \frac{1}{p_i} - \frac{1}{\hat{p}_i} \right| \leq \frac{2\sqrt{\frac{\log(2/\delta)}{2n}}}{\Delta_{\min}^2}.$$

*Proof.* The difference can be rewritten as:

$$\left| \frac{1}{p_i} - \frac{1}{\hat{p}_i} \right| = \frac{|\hat{p}_i - p_i|}{p_i \hat{p}_i}.$$

By Hoeffding's inequality, for any $\delta \in (0, 1)$, with probability at least $1 - \delta$, we have:

$$|\hat{p}_i - p_i| \leq \sqrt{\frac{\log(2/\delta)}{2n}}.$$

Let $\varepsilon = \sqrt{\frac{\log(2/\delta)}{2n}}$. Then, with high probability:

$$|\hat{p}_i - p_i| \leq \varepsilon.$$

Since $p_i \geq \Delta_{\min}$ and $|\hat{p}_i - p_i| \leq \varepsilon$, we have:

$$\hat{p}_i \geq p_i - \varepsilon \geq \Delta_{\min} - \varepsilon.$$

To ensure $\hat{p}_i > 0$, we require $\varepsilon < \Delta_{\min}$.

Using $p_i \geq \Delta_{\min}$ and $\hat{p}_i \geq \Delta_{\min} - \varepsilon$, the denominator satisfies:

$$p_i \hat{p}_i \geq \Delta_{\min}(\Delta_{\min} - \varepsilon).$$

Substitute the bounds for $|\hat{p}_i - p_i|$ and $p_i \hat{p}_i$ into the expression for the difference:

$$\left| \frac{1}{p_i} - \frac{1}{\hat{p}_i} \right| = \frac{|\hat{p}_i - p_i|}{p_i \hat{p}_i} \leq \frac{\varepsilon}{\Delta_{\min}(\Delta_{\min} - \varepsilon)}.$$

This gives the first part of the lemma.

If $\varepsilon \leq \frac{\Delta_{\min}}{2}$, then $\Delta_{\min} - \varepsilon \geq \frac{\Delta_{\min}}{2}$, and the bound simplifies to:

$$\left| \frac{1}{p_i} - \frac{1}{\hat{p}_i} \right| \leq \frac{\varepsilon}{\Delta_{\min} \cdot \frac{\Delta_{\min}}{2}} = \frac{2\varepsilon}{\Delta_{\min}^2}.$$

Substituting $\varepsilon = \sqrt{\frac{\log(2/\delta)}{2n}}$, we obtain:

$$\left| \frac{1}{p_i} - \frac{1}{\hat{p}_i} \right| \leq \frac{2\sqrt{\frac{\log(2/\delta)}{2n}}}{\Delta_{\min}^2}.$$

This completes the proof of the lemma. $\qquad\square$

**Lemma H.5** (Concentration Inequality for Total Variation Distance, see e.g. Thm 2.1 by Berend and Kontorovich [2012]). *Let* $\mathcal{X} = \{1, 2, \cdots, |\mathcal{X}|\}$ *be a finite set. Let $P$ be a distribution on $\mathcal{X}$. Furthermore, let $\widehat{P}$ be the empirical distribution given $m$ i.i.d. samples $x_1, x_2, \cdots, x_n$ from P, i.e.,*

$$\widehat{P}(j) = \frac{1}{n} \sum_{i=1}^{n} \mathbb{I}\{x_i = j\}.$$

*Then, with probability at least $1 - \delta$, we have that*

$$\|P - \widehat{P}\|_1 := \sum_{x \in \mathcal{X}} |P(x) - \widehat{P}(x)| \leq \sqrt{\frac{2|\mathcal{X}| \log(1/\delta)}{n}}.$$

*Proof.* Define the function $f(x_1, \ldots, x_n) = \sum_{x \in \mathcal{X}} |\widehat{P}(x) - P(x)|$, where $\widehat{P}$ is the empirical distribution. Replacing one sample $x_i$ can change $f$ by at most $2/n$, since the empirical frequencies change by at most $1/n$ per coordinate and total variation sums these differences.

By McDiarmid's inequality, we have for any $\varepsilon > 0$,

$$\Pr\left(f - \mathbb{E}[f] \geq \varepsilon\right) \leq \exp\left(-\frac{n\varepsilon^2}{2}\right).$$

Berend and Kontorovich (2013) show that $\mathbb{E}[f] \leq \sqrt{\frac{|\mathcal{X}|}{n}}$. Setting the failure probability to $\delta$, we solve

$$\exp\left(-\frac{n\varepsilon^2}{2}\right) = \delta \quad \Longrightarrow \quad \varepsilon = \sqrt{\frac{2\log(1/\delta)}{n}}.$$

Therefore, with probability at least $1 - \delta$,

$$\|P - \widehat{P}\|_1 \leq \sqrt{\frac{|\mathcal{X}|}{n}} + \sqrt{\frac{2\log(1/\delta)}{n}} \leq \sqrt{\frac{2|\mathcal{X}|\log(1/\delta)}{n}},$$

$\square$

