# OpenReview forum: "On Feasible Rewards in Multi-Agent Inverse Reinforcement Learning"
_NeurIPS.cc/2025/Conference — NeurIPS 2025 spotlight_

### Official Review · Reviewer_4wfF · 2025-07-01

**Clarity:** 3
**Significance:** 3
**Originality:** 3
**Rating:** 4
**Confidence:** 3

**Summary:**

This paper presents a theoretical analysis of reward recovery feasibility in multi-agent inverse reinforcement learning (MAIRL) within Markov games, using two behavioral game theory equilibrium concepts: (i) Nash equilibrium and (ii) Quantal Response Equilibrium (QRE). It demonstrates that under the Nash setting, access to a single equilibrium policy may be insufficient for accurate reward recovery, as the resulting policy can exhibit a large optimality gap (Proposition 3.4). In contrast, under the QRE setting, the paper establishes sample complexity bounds for estimating transition dynamics and agents' QRE policies, ensuring reward recovery meets optimality guarantees (Theorem 3.9). Additionally, Theorem 4.1 identifies conditions under which a player’s average reward can be recovered from observations of two distinct opponent policies.

**Questions:**

1. Can you please justify the assumption of the same average reward function in Theorem 4.1?

2. Can you elaborate on how your analysis on QRE can be extended to more than 2 players?

**Ethical Concerns:**

["NO or VERY MINOR ethics concerns only"]

**Final Justification:**

Based on our post-rebuttal discussion, I will keep my rating. Thanks!

**Limitations:**

Yes.

**Paper Formatting Concerns:**

None.

**Quality:**

3

**Strengths And Weaknesses:**

Strengths:
+ The paper offers a thorough theoretical analysis of the feasibility, ambiguity, and complexity of reward recovery in Markov games. The analysis is grounded in well-established equilibrium concepts—Nash and Quantal Response Equilibrium (QRE)—from competitive game theory.
+ The writing is clear and well-organized.

Weaknesses:
− The connection between the Nash and QRE analyses lacks consistency. While the Nash analysis applies to general Markov games with two or more players, the QRE results are restricted to the two-player setting.

− In Theorem 4.1, it is unclear under what conditions the assumption—that two distinct opponent policies yield the same average reward for the player—can be realistically satisfied.

− In Theorem 3.9, all equations and definitions referred should be included in the main paper, not in the appendix.

---

> ### Author Rebuttal · Authors · 2025-07-31
>
> We thank the reviewer for their careful reading and valuable feedback. We are glad the reviewer found the paper *“clear and well-organized”* and appreciated its *”thorough theoretical analysis”*.
>
> Below, we address the reviewer’s concerns in a detailed way.
>
> **Extension to n-player setting**
>
> While our QRE-based analysis is presented in the two-player setting for clarity, the framework can be naturally extended to the general $n$-player case. Specifically:
>
>
> - In Theorem 3.8 we would then have the sum of the total variation of the variations of the other players' policies.
> - In Theorem 3.9, as stated below the theorem, we would get an exponential dependency in the number of agents $\mathcal{A}^n$.
>
> We will clearly state in the introduction that we focus on the two-player case for readability, but the framework extends to $n$ players.
>
> **Average reward assumption**
>
> - The assumption that two distinct opponent policies yield the same average reward for a player can hold in various realistic settings. One example is a symmetric zero-sum game where the agent’s reward depends only on the final outcome (e.g., win/loss).
> - This assumption is also consistent with identifiability conditions in the single-agent inverse RL literature (e.g., [1]), where one needs to observe the same reward function across two MDPs.
>
> - Compared to the single-agent case, this assumption can be less restrictive as in the single-agent case, it is required to observe two different MDPs with different transition functions as well. On the other hand, in the multi-agent case, a different transition function can be induced by a different expert in the same Markov Game. Coming back to the example, one can consider a Tic-Tac-Toe game where one faces two different winning strategies; then the average reward function is the same, and the transition dynamics can be altered for the same Tic-Tac-Toe game.
>
>
> **Notations in the appendix**
>
> We thank the reviewer for noting this and agree with the reviewer that all the notations should be included in the main paper instead of the appendix. As the final version allows an additional page, we include the necessary notation in the main section of the camera-ready version.
>
> Once again,  we would like to thank the reviewer for their valuable feedback that helps us to improve the paper, and we are happy to answer any further questions that the reviewer might have.
>
> [1] Rolland, Paul, et al. "Identifiability and generalizability from multiple experts in inverse reinforcement learning." Advances in Neural Information Processing Systems 35 (2022): 550-564.

---

> > ### Comment · Reviewer_4wfF · 2025-08-05
> >
> > Thank you for the authors' response. My Q2 has been addressed.
> >
> > However, regarding Q1, I remain unconvinced by the additional clarifications provided. The setting in [1] differs in that it assumes a fixed reward function across multiple MDPs and uses diverse expert policies to recover this common reward. On the other hand, under the average reward assumption, the “average” reward for each (state, action) pair is defined as an expectation with respect to the opponent’s policy. The assumption that two different opponent policies can yield the same average reward effectively translates into a set of linear constraints that these policies must satisfy. It remains unclear how restrictive  this assumption is in practice.

---

> > > ### Author Response · Authors · 2025-08-06
> > >
> > > Thank you for the interesting question. While the assumption can be more restrictive, the freedom of choosing different opponent strategies also gives us more freedom. A simple, concrete setting where two distinct opponent policies induce exactly the same average reward is a *simplified turn-based poker*:
> > >
> > > **Setup.**
> > >  - Public state $s$: the board cards and pot size.
> > >  - Your action $a \in \\{\mathrm{bet}, \mathrm{check}\\}$.
> > >  - Opponent action $b \in \\{\mathrm{fold}, \mathrm{call}, \mathrm{raise}\\}$.
> > >  - Stage payoff $r(s,a,b)$: chip change resulting from the resolution of $(a,b)$.
> > > - Two opponents with different strategies that result in the same value for player 1:
> > >  1. $\nu_1$ is a standard Nash equilibrium that mixes calls and folds in certain proportions on draw boards.
> > >  2. $\nu_2$ is a different equilibrium—identical on all _reachable_ infosets but with alternative mixing in _off-path_ nodes you almost never visit.
> > > - This yields the same average reward but different transition dynamics.
> > >
> > > In general, in many zero-sum games there are typically *infinitely many* strategies that are value-equivalent, i.e.they induce the same counterfactual payoffs, even though their action distributions differ in irrelevant or low-probability branches. Collapsing to the MDP over public states gives exactly two different transitions but the same average reward.
> > > This poker example shows that the "same average reward under two different opponents" can be both natural and easy to satisfy in practice.

---

### Official Review · Reviewer_14AJ · 2025-07-02

**Clarity:** 3
**Significance:** 3
**Originality:** 2
**Rating:** 5
**Confidence:** 3

**Summary:**

The paper investigates the Inverse RL problem in the multi-agent setting (MAS) where they formalize the problem of learning the joint reward function (feasible reward set) from the expert demonstrations (at some equilibrium). They formalize the added complexities that arise due to the multi-agent setting, in particularly, they present an important result that states a observations from a singe Nash equilibrium can be insufficient for a identifying feasible reward set. This is due to the nature of multi-agent settings where there can be multiple Nash-equilibria and it can result in a Nash of order $(1-\gamma)^-1$ (Prop 3.4).

The paper then presents a method to mitigate this issue via building on the recent work in single-agent entropy-regularized IRL in the multi-agent IRL setting and Quantile Response Equilibria (QRE). For the QRE, they provide the analysis in terms of error and sample-complexity bounds for finding feasible rewards (Theorem 3.7, Thm 3.9).

Then the authors ask if the reward identifiability even feasible in MAS and propose that for general-sum Markov Games, it is only possible for the average reward case where the reward structure has a particular form, i.e. when they are linearly separable (Prop 4.3).

**Questions:**

- My suggestion will be to bringing the QRE and average reward definitions in technical background, and then work with consistent setting throughout the paper. If that seems quite a big delta from the current version, a more detailed section describe the need of switching the setting will help the reader and make paper more accessible.

**Ethical Concerns:**

["NO or VERY MINOR ethics concerns only"]

**Limitations:**

yes

**Quality:**

3

**Strengths And Weaknesses:**

# Strengths
- The paper is clearly written and does the important task of characterizing the MAIRL problem which is important given the agentic trends.
- The paper is rigorous - assumptions are laid clearly, and the results and analysis does a good job of characterizing the findings.
- The main result shows the hardness of the MAIRL problem and lays open the ground for future opportunities and problems in this domain.

# Weakness
- The main weakness of the paper is Sec 3.2 where the paper switches the setting to Entropy-regularized MGs and QRE. While the authors provide some motivation of working with QRE, it is unclear why wasn't this introduced in the beginning (instead of NE). That would make the paper more consistent. For instance, how does QRE changes the optimality gape (Definition 3.3) that was defined earlier?
- A similar concern also lies with switching from regular rewards to average rewards based criterions (Eq 2). Why we need this? How does it impact the general setting? Any other implications?

---

> ### Author Rebuttal · Authors · 2025-07-31
>
> We thank the reviewer for their thoughtful and encouraging feedback. We are glad that the reviewer found the paper  *” is clearly written”* and *”rigorous”*.
>
> Below we address the reviewer’s questions and suggestions in detail.
>
> **Introduction of QRE**
>
> We appreciate the reviewer’s suggestion to introduce QRE earlier in the paper. Our current presentation begins with NE because it is the most commonly studied solution concept in game-theoretic analysis. This allows us to clearly highlight the limitations of single-equilibrium inference. That said, we agree that introducing QRE earlier would provide a more consistent conceptual framework throughout. In the camera-ready version, we make the following adjustments:
>
> - Introduce QRE in the Preliminaries (in the Entropy Regularized Game section).
> - Clarify the transition from NE to QRE in Section 3.2, emphasizing the practical and analytical motivations for using QRE.
>
> **Average reward**
>
> We introduced the average-reward setting (Equation 2) in the context of our second research question, on when rewards are identifiable in multi-agent games. This was not necessary before, as we were mainly considering the feasible reward set, meaning all reward functions that are feasible under an observed expert equilibrium. To clarify this, we are happy to explicitly state this transition when Equation 2 is introduced.
>
> Once again, we thank the reviewer for their constructive and positive feedback, helpful suggestions, and we hope that we have clarified the reviewer's concerns.

---

> > ### Comment · Reviewer_14AJ · 2025-08-06
> > **Acknowledgment**
> >
> > Thanks for responding to my comments and incorporating my suggestions. I will maintain my original rating of accept.

---

### Official Review · Reviewer_7b2B · 2025-07-02

**Clarity:** 2
**Significance:** 4
**Originality:** 3
**Rating:** 5
**Confidence:** 4

**Summary:**

The paper studies feasible‐reward sets in Multi‐Agent Inverse Reinforcement Learning (MAIRL), showing that inferring rewards from a single observed Nash equilibrium is fundamentally ambiguous, different reward assignments can admit novel, exploitable equilibria. To address this, the authors first define a multi‐equilibrium feasible‐reward set (intractable in general) and then introduce an entropy‐regularized framework yielding a unique quantal‐response equilibrium (QRE). They propose a new optimality criterion, the Nash Imitation Gap, to mirror the value‐gap objective in single‐agent IRL, and under the QRE they derive an explicit characterization of all reward functions consistent with the expert policy. For linearly decomposable rewards, identification is possible up to additive constants (as in single‐agent IRL), and they prove polynomial sample‐complexity bounds for recovering these rewards from finite data. Finally, they also establish a negative result: without entropy regularization, reward identifiability is not possible in the general MAIRL setting.

**Questions:**

### Remarks and Questions
1. Content-Sensitive Typos
	- In the "Mathematical Background" paragraph, the definition of the Hausdorff (pre)metric contains two $\inf_{\sup_{y\in\mathcal{Y}}}$ which are supposed to be $\inf_{y\in\mathcal{Y}}$.
	- In line 180 "...player 2 plays action $a_{2}$ ..." should be "action $b$" instead to align with the following expressions.
	- In lines 109 and 181 the $\mathcal{A}_{i}$ should be $\mathcal{A}^{i}$ to be consistent with the notation introduced in the Preliminaries/Markov Games paragraph.
	- In line 110 the expectation is missing the condition on the initial state $s$.
	- In line 192 the closing parenthesis on the superscript of the last $V$ is aligned with the $Nash$ instead of the $\pi$.
	- In line 362 the cross-reference to "Section 4"  should refer to the appendix?
2. Some elements would benefit from some notational clarity. To give some examples:
	 - On line 88, the probability $\mathbb{P}$ should include a time index (e.g. $\mathbb{P}^t$) to indicate that it transitions from the initial state to $(s,\mathbf{a})$ by executing the policy for $t$ steps.
	 - E.g. in lines 98 and 110 it would increase clarity to add a time index to variables where applicable ($s$, $a$, $b$, etc.) or a separate notation for random variables and realizations.
	 - The paragraph "Entropy Regularized Markov Games" would benefit from adding that it is described in the setting of a two-player game.
	 - The action set of player 2 is sometimes referred to as $\mathcal{A}^{2}$ and sometimes as $\mathcal{B}$, while both refer to the same set, I would suggest using the same notation as synonyms might spark a first thought of -- something is different here? -- when its in fact the same object.
	 While most of these issues can be understood in the context and it is mostly clear from the context what is meant by the expressions or symbols, I think that it would improve clarity and ease of access.
3. Figure 2 needs a more thorough description to give context to the values shown as well as the "Stag Hunt MG". Another option would be to move this figure to E.1.
4. There are some consistency related issues. The two things that stand out the most are the mixed usage of "Markov Games" (capitalized) and "Markov games" (lowercase) as well as the inconsistent capitalization in section and paragraph titles. Lastly, the use of acronyms versus their expanded counterparts is inconsistent. Fixing these little issues would further improve the otherwise well written text.
5. In Theorem 3.7, is $TV(\nu, \hat{\nu})$ the total variation distance?
6. The insights from the future-work and limitations sections should be integrated into Section 5 of the main paper, as they offer valuable and relevant information on the topic.

##### Typos:
- Description of Figure 1: Plural "..sets of Nash equilibira..."
- Line 120: considers -> consider
- Line 128: Typo in equilibrium
- Line 309: agent' 1 -> agent 1's ?
- Line 367: showed -> shown

**Ethical Concerns:**

["NO or VERY MINOR ethics concerns only"]

**Final Justification:**

My concerns centered on notation, clarity, and consistency, as I worried that the interesting contribution might not be as accessible as befits the importance of the topic; the authors agree with these points and intend to improve the paper accordingly. I have read the other reviews and their rebuttals and do not see any major points that would contradict my decision. Because these changes are easily done within the scope of producing the camera-ready version, I propose to accept this work.

**Limitations:**

yes

**Quality:**

3

**Strengths And Weaknesses:**

Besides some room for improved clarity, as noted above, the paper is well written and pleasant to read. The authors explain the topic with attention to detail and make room to elaborate on the underlying intuition. They do a great job capturing the essence of the MAIRL problem and explaining if, when, and how feasible-reward identification is possible -- delivering what a reader would expect from a paper with this title. They also introduce valuable tools, such as their optimality criterion, that future work can build upon. I believe this work makes a valuable contribution by strengthening the foundations of MAIRL and will be a relevant read for anyone entering the field. That is why I have emphasized clarity and the placement of key information in the main text: clear, unambiguous presentation is particularly important when building a theoretic foundation, and especially helpful for newcomers. However, few of the issues I have raised detract from the paper’s contributions; if the content-sensitive typos are corrected or clarified, I consider this work a valuable contribution.

---

> ### Author Rebuttal · Authors · 2025-07-31
>
> We thank the reviewer for their detailed and constructive feedback. We are pleased that the reviewer found the paper *”the paper is well written and pleasant to read”* and *”makes a valuable contribution by strengthening the foundations of MAIRL and will be a relevant read for anyone entering the field.”* Below, we address all comments and suggestions in detail.
>
> **Content-Sensitive Typos**
>
> - We corrected the definition of the Hausdorff metric to properly reflect the intended subscript and structure:
>
> $\mathcal{H}\_d(\mathcal{Y}, \mathcal{Y}') := \max\left\\{ \sup\_{y \in \mathcal{Y}} \inf\_{y' \in \mathcal{Y}'} d(y,y'), \sup\_{y' \in \mathcal{Y}'} \inf\_{y \in \mathcal{Y}} d(y,y')\right\\}$
>
> - Changed action $a_2$ to action $b$.
> - We changed the subscript in the action space to a super script  $\mathcal{A}^i$ to be consistent.
> - Added the missing condition $s \sim \rho$ in the expectation on Line 110.
> - We changed the alignment in line 192: $V^{i, (\pi^{\mathrm{Nash}}\_i, \pi\_{-i}^{\mathrm{Nash}}) }\_{\mathcal{G} \cup R}(s) $.
> - We changed the reference in line 362 to Appendix G.
>
> **Clarity**
>
> - We added time indices where appropriate.
> - Adopted a convention where random variables are capitalized and their realizations use lowercase (e.g., $S_t$, $s_t$).
> - In the paragraph on entropy-regularized Markov games, we now clarify that the exposition focuses on the two-player setting, but the framework generalizes to $n$-player games.
> - While we defined $\mathcal{A}^2 := \mathcal{B}$ on Line 109, we acknowledge the potential confusion and now consistently use one notation throughout the paper for clarity.
>
> **Figure 2**
>
> We move the illustration of the Markov Game to section E.1 and refer in the main text to this section for a detailed description of the setup.
>
> **Consistency and Terminology**
>
> - To align with common usage in the literature, we now consistently use *Markov game*
> - We standardized capitalization across section and paragraph titles.
> - We ensured consistent usage of acronyms and their expanded forms throughout the paper.
>
> **Total Variation**
>
> Yes, the reviewer is correct. We now introduce this explicitly in the Preliminaries (Mathematical background).
>
> **Limitations and future work section**
>
> As the camera-ready version allows us to include an additional page, we are happy to extend section 5 by including the future work and limitation setting which used to be in Appendix C.
>
> **Typos**
>
> We thank the reviewer for pointing out these typos, we have fixed them in the current version of the paper (equilibira -> equilibria, considers -> consider, agent’1 -> agent 1’s, showed -> “shown”).
>
>
> Once again,we would like to thank the reviewer for their careful reading and valuable suggestions. We believe these revisions improve the clarity and quality of the paper, and we hope it now better reflects the significance of the contributions.

---

> > ### Comment · Reviewer_7b2B · 2025-08-01
> >
> > Thank you for incorporating my feedback and improving the notation. The clearer notation and the relocation of the limitations to the main body make the contribution more accessible. This directly addresses my primary concern; accordingly, I am comfortable increasing my score.

---

> > > ### Author Response · Authors · 2025-08-06
> > >
> > > Dear Reviewer 7b2B,
> > >
> > > Thank you very much for increasing your score. We are glad your concerns have been addressed.
> > >
> > > Best,
> > > Authors

---

### Official Review · Reviewer_qBYJ · 2025-07-03

**Clarity:** 4
**Significance:** 4
**Originality:** 3
**Rating:** 5
**Confidence:** 3

**Summary:**

The paper explores the problem of inverse reinforcement learning (IRL) in Markov games (MG). They start with defining a feasible reward set for a given MG as the set of reward functions under which a single observed Nash equilibrium (NE) policy has higher value than the policies that can be obtained by unilateral deviations. The authors observe the existence of degenerate MGs for which a NE policies of a non-constant reward function in the feasible reward set might have Nash Imitation Gap of order $(1 - \gamma)^{-1}$. This stems from the fact that the feasible reward set captures many reward functions, some of which allows undesirable scenarios where the nature of the game is changed. The authors argue that, although one can address this issue by requiring a feasible reward function to align with every NE of the original game, this would lead to tractability issues. In order to mitigate these issues, they investigate the entropy-regularized 2-player MGs. In this setting, they show that the sample complexity of the problem is of order $\mathcal{O}\left(\gamma^2 (1 - \gamma)^{-4} |\mathcal{S}| |\mathcal{A}| |\mathcal{B}| \Delta_{\min}^{-4} \right)$, where $\mathcal{A}$ and $\mathcal{B}$ are the action spaces of agents, $\mathcal{S}$ is the state space and $\gamma$ is the discount factor, and $\Delta_{\min}$ is a lower bound on the lowest probability of an action. While this shows an exponential dependence on the number of players, the authors point out that this matches the complexity of learning an NE, which is known to have such exponential dependence in the worst case. Finally, the authors focus on the identifiability problem and note that when the other agent's policy is fixed, the 2-agent game reduces to an MDP, which allows the single-agent IRL theory to be used. The authors thus provide a condition under which the average reward can be identified up to a constant. Finally, the authors show that reward identification is possible in linearly separable Markov games.

**Questions:**

1. As of Section 3.2, the authors focuses two-player MGs for better readability (line 234). Is it possible to straightforwardly extend the analysis to n-agent case?
2. Theorem 3.7 provides an upper-bound on the estimated rewards for two MAIRL problems. This allows the quantification of the difference between the original and recovered MAIRL problems. Could your analysis lead to a lower bound on the difference of estimated rewards as well to understand the fundamental limitations of the problem?
3. Does your results require knowing $\lambda$ for entropy-regularized Markov games? If so, could you discuss how restrictive is this assumption? Is it possible to obtain a bound similar to Theorem 3.7 for when there is a mismatch between the true $\lambda$ and the one used for solving the MAIRL problem?

**Ethical Concerns:**

["NO or VERY MINOR ethics concerns only"]

**Final Justification:**

The authors identify the core aspects of the problem and elaborate on their methodology with relevant discussions. The paper is well-written and is enjoyable to read. The authors' rebuttal addressed the concerns. They will add further improvements on clarity that will enhance readability.

**Limitations:**

yes

**Quality:**

4

**Strengths And Weaknesses:**

**Quality and Clarity:** The paper presents a strong and methodologically sound contribution. The results and the ideas are clearly explained. Some experimental results were presented; as an addition, it would be nice to see an experiment designed to observe how identifiability differs in general games and linearly separable games.

**Significance and Originality:** This paper extends the theoretical line of work on multi-agent inverse reinforcement learning (MAIRL) under varied game-structure assumptions. It demonstrates the inherent reward ambiguity that arises when only a single equilibrium is observed, and analyses the identifiability of reward functions in entropy-regularized games, both with and without the assumption of linearly separable rewards.

**Minor issues:**
1. typo in Definition 3.1: equilirbium
2. the dependence of $\mathcal{E}$ on $R$ is rather unclear at first sight, e.g. in the inequalities in Definition 3.3. I would consider making the dependence explicit.
3. There are some related works that were not cited in the work, including: [1], [2]

[1] X. Lin, S. C. Adams, and P. A. Beling, ‘Multi-agent Inverse Reinforcement Learning for Certain General-sum Stochastic Games’, _Journal of Artificial Intelligence Research_, vol. 66, pp. 473–502, Oct. 2019.
[2] J. Liao, Z. Zhu, E. X. Fang, Z. Yang, and V. Tarokh, ‘Decoding Rewards in Competitive Games: Inverse Game Theory with Entropy Regularization’, in _Forty-second International Conference on Machine Learning_, 2025.

---

> ### Author Rebuttal · Authors · 2025-07-31
>
> We thank the reviewer for their time to read our paper, providing this kind and insightful review that helps us to improve it. In particular, we are grateful that the reviewer acknowledges that our paper makes a  *“strong and methodologically sound contribution.”*
>
> Below, we would like to address the concerns raised by the reviewer and answer the questions.
>
> ### Experiment on Identifiability
>
> We thank the reviewer for the excellent suggestion to include an experiment validating our claims on identifiability. We will add the following to the appendix.
>
> **Experimental Setup**: We test identifiability by observing the QRE policies from two environments with different, randomly generated transition dynamics satisfying the identifiability. The Markov Game has 15 states, and both agents have 5 actions for both setups. We only consider the results for the reward of the first player; it follows analogously for the second one.
> - *Averagel Case*: We first use a ground-truth reward $R\_1​(s,a\_1​,a\_2​)$. The algorithm perfectly identifies the *average reward* (max difference: 0.000000). However, the full reward is not identifiable, achieving a correlation of only 0.9097 with the ground truth.
> - Linearly Separable Case: We repeat the experiment with a linearly separable reward. Then, the algorithm perfectly recovers the full reward component R\_1​ (max difference: 0.000000), meaning the full reward is successfully identified.
> We summarize the results of the correlation in the following table
>
> | Game Structure           | Correlation with True Full Reward |
> |--------------------------|:---------------------------------:|
> | Average Case             | 0.9097                        |
> | Linearly Separable Case  | 1.0000                  |
>
> ### Minor issues
>
> We thank the reviewer for pointing out these minor issues, and we have addressed them separately to improve the quality of our paper:
>
> - Fixed typo, changed equilirbium to equilibrium
> - To clarify the dependence on $R$, we addressed the dependency directly in Definition 3.2 and changed it to $\mathcal{E}(\hat{\boldsymbol{\pi}};R) := \max\_{i \in [n]} \max\_{\pi^i \in \Pi^i} V\_{\mathcal{G}\cup R}^{i}(\pi^i,\hat{\pi}^{-i}) - V\_{\mathcal{G}\cup R}^{i}(\boldsymbol{\hat{\pi}}).$ Then, we can include $\mathcal{E}(\hat{\boldsymbol{\pi}};R)$ also in Definition 3.3 to emphasize the dependence.
> - We thank the reviewer for pointing out the missing citations. We have already cited [1] in our related work section (line 984). Regarding [2] we are happy to include the concurrent work(it was released after the NeurIPS submission deadline), in our related work section of the camera-ready version. In particular, we note that [2] introduces identifiability in the context of linear payoff functions, where the essential goal is to estimate the parameter $\theta$ if the experts are playing a QRE. Next, the authors extend their results to Zero-Sum Markov Games with a linear reward function. In contrast, we consider the general-sum Markov Game case, and additionally, we provide insights on the ambiguity of the feasible reward set.
>
> ### Extension to n-player case
>
> - We would like to affirm this question from the reviewer. The results can easily be generalized to the n-person case:
>        - If we denote the policy of the other player as $\pi\_{-i}$, all policies but the policy of agent i, we can simply replace $\nu$ by this joint policy.
>        - In Theorem 3.8 we would then have the sum of the total variations of the other players' policies.
>        - In Theorem 3.9, as stated below the theorem, we would get an exponential dependency in the number of agents $\mathcal{A}^n$.
> We will add these comments as a footnote in the camera-ready version.
>
> ### Lower bound
>
> This is an interesting question raised by the reviewer, and it remains an open question for future works. Constructing a lower bound is, in general, not easy and as we have shown in this work, MAIRL is hard, and therefore it is unclear if techniques used in the single-agent case [3] also apply to the multi-agent setting. Next, we lay out our conjectures:
>
> - In the single-agent case, it was shown that Inverse RL suffers from the same lower bound as RL, except for an additional $(1-\gamma)$ factor, when given access to a generative model [3]. This could indicate that, also in Multi-agent Inverse RL, the lower bound will be of the same order as learning equilibria in the first place. This would mean that a lower bound would also suffer from the curse of multi-agents.
> - In [5] the authors provide a lower bound for Zero-Sum Games for the reward agnostic case, meaning that the learner has no access to the reward function during learning. The lower bound here scales with $\vert \mathcal{A} \vert \vert \mathcal{B}\vert$, which is worse than the reward-aware case in zero-sum games, scaling linearly in the number of players.
>
> To summarize, we conjecture that Multi-Agent Inverse RL will probably suffer from the curse of multi-agents, meaning a bound that will scale exponentially in the number of players. We will add a comment about this future direction in the appendix.
>
> ### Knowledge of entropy coefficient
>
> We thank the reviewer for this relevant question. We answer it below:
>
> - The knowledge of the entropy coefficient is also required in the single-agent case [4].
> - To check the identifiability conditions, knowledge of $\lambda$ is not required.
> - If the goal is to recover a specific reward function, it is necessary to know the coefficient, as this scales the recovered reward. Therefore, a mismatch in this coefficient would also result in a mismatch of this difference for the recovered reward and true identifiablity would not be possible.
> - Regarding the feasibility, one can see this as input given by the expert, as the entropy coefficient reflects the bounded rationality of the agent in a multi-agent setting. This means how much the expert believes in the rationality of the other agents.
>
> We are happy to include a discussion regarding this in the camera-ready version of the paper.
>
> [3] Metelli, Alberto Maria, Filippo Lazzati, and Marcello Restelli. "Towards theoretical understanding of inverse reinforcement learning." International Conference on Machine Learning. PMLR, 2023.
>
> [4] Rolland, Paul, et al. "Identifiability and generalizability from multiple experts in inverse reinforcement learning." Advances in Neural Information Processing Systems 35 (2022): 550-564.
>
> [5] Zhang, Kaiqing, et al. "Model-Based Multi-Agent RL in Zero-Sum Markov Games with Near-Optimal Sample Complexity." Advances in Neural Information Processing Systems 33 (2020): 1166-1178.

---

> > ### Comment · Reviewer_qBYJ · 2025-08-04
> >
> > I thank the authors for the response. Apologies for the oversight about the missing citations. I have read the discussions and agree with other reviewers that the paper is a enjoyable to read. I will increase my score.

---

> > > ### Author Response · Authors · 2025-08-06
> > >
> > > Dear Reviewer qBYJ,
> > >
> > > Thank you very much for increasing your score. We are glad that you find the paper enjoyable to read and that your concerns have been addressed.
> > >
> > > Best,
> > > Authors

---

### Note · Authors · 2025-08-12

We would like to thank the reviewers again for taking the time to read our paper. Additionally, we would like to thank the reviewers for providing insightful comments and engaging in a discussion with us after the rebuttal that helped us to improve the quality as well as the clarity of the camera-ready version.

In general, we are happy to hear that the reviewers think *”that the paper is enjoyable to read”* and that our *”results and the ideas are clearly explained.”* Additionally, we appreciate that the reviewers acknowledge the contributions of our work to the field of Multi Agent Inverse Reinforcement Learning: *”makes a valuable contribution by strengthening the foundations of MAIRL and will be a relevant read for anyone entering the field”* which required us to provide a *”thorough theoretical analysis”.*

We agree with the reviewers and thanks to their help to improve our work, we believe that it serves as a foundation for future works in Mult-Agent Inverse Reinforcement Learning.

---

### Decision · Program_Chairs · 2025-09-17

**Decision:**

Accept (spotlight)

**Comment:**

This paper makes nice progress on core challenges of understanding the theoretical underpinnings of MAIRL. At this stage I don't see any major outstanding concerns.  There was a lingering concern that the assumption in Theorem 4.1 may be on the strong side, but I am satisfied with the author's justification.  There are a number of constructive comments regarding the presentation that I encourage the authors to incorporate.